

**Aerosol characteristics and particle production in the upper troposphere**
**over the Amazon Basin**
Meinrat O. Andreae[1,12], Armin Afchine[2], Rachel Albrecht[3], Bruna Amorim Holanda[1], Paulo Ar-
taxo[4], Henrique M. J. Barbosa[4], Stephan Borrmann[1], Micael A. Cecchini[5,3], Anja Costa[2], Maxi-
milian Dollner[6,9], Daniel Fütterer[6], Emma Järvinen[10], Tina Jurkat[6], Thomas Klimach[1], Tobias
Konemann[1], Christoph Knote[9], Martina Krämer[2], Trismono Krisna[8], Luiz A. T. Machado[5],
Stephan Mertes[7], Andreas Minikin[6,16], Christopher Pöhlker[1], Mira L. Pöhlker[1], Ulrich Pöschl[1],
Daniel Rosenfeld[14], Daniel Sauer[6], Hans Schlager[6], Martin Schnaiter[10], Johannes Schneider[1],
Christiane Schulz[1], Antonio Spanu[6], Vicinius B. Sperling[5], Christine Voigt[6,15], Adrian Walser[9,6],
Jian Wang[1,11], Bernadett Weinzierl[6,13], Manfred Wendisch[8], and Helmut Ziereis[6]
[1]Biogeochemistry, Multiphase Chemistry, and Particle Chemistry Departments, Max Planck Institute for Chemistry,
Mainz, Germany
[2]Forschungszentrum Jülich, Jülich, Germany
[3]Instituto de Astronomia, Geofísica e Ciências Atmosféricas, Universidade de São Paulo, São Paulo, Brazil
[4]Institute of Physics, University of São Paulo, São Paulo, Brazil
[5]National Institute for Space Research (INPE), São José dos Campos, Brazil
[6]German Aerospace Center (DLR), Institute of Atmospheric Physics (IPA), Weßling, Germany
[7]Leibniz Institute for Tropospheric Research, 04318 Leipzig, Germany
[8]Leipzig Institute for Meteorology, Leipzig University, Leipzig, Germany
[9]Meteorological Institute, Ludwig Maximilian University, Munich, Germany
[10]Institute for Meteorology and Climate Research, Karlsruhe Institute of Technology, Karlsruhe, Germany
[11]Brookhaven National Laboratory, Upton, New York, USA
[12]Scripps Institution of Oceanography, University of California San Diego, La Jolla, California, USA
[13]University of Vienna, Aerosol Physics and Environmental Physics, Wien, Austria
[14]Institute of Earth Sciences, The Hebrew University of Jerusalem, Israel
[15]Institute of Atmospheric Physics (IPA), Johannes Gutenberg University, Mainz, Germany
[16]German Aerospace Center (DLR), Flight Experiments, Oberpfaffenhofen, Germany

**Abstract**
Airborne observations over the Amazon Basin showed high aerosol particle concentra-
tions in the upper troposphere (UT) between 8 and 15 km altitude, with number densities (nor-
malized to standard temperature and pressure) often exceeding those in the planetary boundary
layer (PBL) by one or two orders of magnitude. The measurements were made during the Ger-
man-Brazilian cooperative aircraft campaign ACRIDICON-CHUVA on the German High Alti-
tude and Long Range Research Aircraft (HALO). The campaign took place in September/Octo-
ber 2014, with the objective of studying tropical deep convective clouds over the Amazon rain-
forest and their interactions with atmospheric trace gases, aerosol particles, and atmospheric radi-
ation.



Aerosol enhancements were observed consistently on all flights during which the UT was
probed, using several aerosol metrics, including condensation nuclei (CN) and cloud condensa-
tion nuclei (CCN) number concentrations and chemical species mass concentrations. The UT
particles differed in their chemical composition and size distribution from those in the PBL, rul-
ing out convective transport of combustion-derived particles from the BL as a source. The air in
the immediate outflow of deep convective clouds was depleted in aerosol particles, whereas
strongly enhanced number concentrations of small particles (<90 nm diameter) were found in UT
regions that had experienced outflow from deep convection in the preceding 5-72 hours. We also
found elevated concentrations of larger (>90 nm) particles in the UT, which consisted mostly of
organic matter and nitrate and were very effective CCN.
Our findings suggest a conceptual model, where production of new aerosol particles takes
place in the UT from volatile material brought up by deep convection, which is converted to con-
densable species in the UT. Subsequently, downward mixing and transport of upper tropospheric
aerosol can be a source of particles to the PBL, where they increase in size by the condensation
of biogenic volatile organic carbon (BVOC) oxidation products. This may be an important
source of aerosol particles in the Amazonian PBL, where aerosol nucleation and new particle
formation has not been observed. We propose that this may have been the dominant process sup-
plying secondary aerosol particles in the pristine atmosphere, making clouds the dominant con-
trol of both removal and production of atmospheric particles.

## 62    1. Introduction

Aircraft measurements in the upper troposphere (UT) have consistently shown large re-
gions with very high aerosol particle number concentrations, typically in the tens of thousands of
particles per $cm^3$, with the strongest enhancements reported in tropical and subtropical regions
(Clarke et al., 1999; Andreae et al., 2001; de Reus et al., 2001; Twohy et al., 2002; Krejci et al.,
2003; Lee et al., 2003; Young et al., 2007; Ekman et al., 2008; Yu et al., 2008; Froyd et al.,
2009; Weigelt et al., 2009; Borrmann et al., 2010; Clarke and Kapustin, 2010; Mirme et al.,
2010; Weigel et al., 2011; Waddicor et al., 2012; Reddington et al., 2016; Rose et al., 2017). In
most cases, these elevated aerosol concentrations were in the nucleation and Aitken mode size
ranges, i.e., at particle diameters smaller than about 90 nm, with maxima typically between 20
and 60 nm (e.g., de Reus et al., 2001; Lee et al., 2003; Weigel et al., 2011; Waddicor et al.,



2012). They generally occur as layers of a few hundred to thousand meters in thickness, often ex-
tending over large horizontal distances, and are found over continents as well as over the most
remote oceanic regions. The high concentrations of these aerosols in the UT are of great signifi-
cance for the climate system, because they make this region an important reservoir of particles
for the transport either downward into the planetary boundary layer (PBL) (Clarke et al., 1999;
Clarke et al., 2013; Wang et al., 2016a) or upward into the Tropical Transition Layer (TTL) and
the lower stratosphere (Weigel et al., 2011; Randel and Jensen, 2013), where they can grow into
the optically and cloud-microphysically active size range.

Where enhanced particle concentrations in the accumulation mode (larger than about 90

nm) have been observed, the enrichment was frequently attributed to sources of sulfur dioxide
($SO_2$) and other combustion emissions, especially biomass burning, based on correlations with
combustion tracers, such as carbon monoxide (CO), and airmass trajectories (e.g., Andreae et al.,
2001; Clarke and Kapustin, 2010; Weigel et al., 2011; Clarke et al., 2013). After having been
lofted to the UT by deep convection, particles in this size range can be transported over hemi-
spheric distances, because removal processes are very inefficient at these altitudes (Andreae et
al., 2001; Clarke and Kapustin, 2010).

The enhanced particle concentrations in the ultrafine (UF) size range (here defined as par-

ticles smaller than 90 nm), on the other hand, cannot be explained by transport from the lower
troposphere, since they far exceed typical concentrations in the PBL and generally are too short-
lived to survive deep convection and long-range transport. Therefore, nucleation and new parti-
cle formation (NPF) from gas phase precursors brought into the UT by the outflow from deep
convection have been proposed as the source of these enhanced particle concentrations (Clarke et
al., 1999; Twohy et al., 2002; Krejci et al., 2003; Lee et al., 2003; Young et al., 2007; Froyd et
al., 2009; Merikanto et al., 2009; Weigel et al., 2011; Waddicor et al., 2012; Carslaw et al.,
2017). High actinic flux, low preexisting aerosol surface area, and low temperatures make the
UT an environment that is highly conducive to nucleation and NPF.

The nature of the gaseous species involved in particle nucleation and growth has been the

subject of some debate (Kulmala et al., 2006). Most of the earlier papers attributed the nucleation
to $H_2SO_4$ in combination with $H_2O$ and $NH_3$, especially in marine and anthropogenically influ-
enced regions, where a sufficient supply of sulfur gases from either DMS oxidation or pollution



sources is available (e.g., Clarke et al., 1999; Twohy et al., 2002; Lee et al., 2003; Merikanto et
al., 2009). However, there is growing evidence that, in most cases, there is not enough $H_2SO_4$
available to explain the observed rates of growth. Therefore, the condensation of organics has
been proposed to dominate particle growth after nucleation, especially over unpolluted vegetated
areas such as the Amazon Basin (Ekman et al., 2008; Weigel et al., 2011; Waddicor et al., 2012;
Murphy et al., 2015).
In fact, $H_2SO_4$ may not always be required even to be the initially nucleating species. Re-
cent studies conducted as part of the Cosmics Leaving OUtdoor Droplets (CLOUD) project have
shown that organic vapors alone can produce particle nucleation (Kirkby et al., 2016) and that
nearly all nucleation throughout the present-day atmosphere involves ammonia or biogenic or-
ganic compounds (Dunne et al., 2016). Highly oxygenated multifunctional organic compounds
(HOMs) formed by ozonolysis of α-pinene were found to nucleate aerosol particles, especially
when aided by ions. Extremely low volatility organic compounds (ELVOCs, which may be at
least in part identical to HOMs) are also produced from the $O_3$- or OH-initiated oxidation of bio-
genic volatile organic compounds (BVOCs) (Jokinen et al., 2015). Following nucleation by the
lowest-volatility species, with increasing particle size the condensation of progressively more
volatile compounds is facilitated by the decrease in the Kelvin effect (Tröstl et al., 2016). These
laboratory studies were confirmed by field observations at a mountain site in the free tropo-
sphere, where NPF was found to take place through condensation of HOMs, albeit from anthro-
pogenic precursor VOCs, within 1-2 days after being lofted from the PBL (Bianchi et al., 2016).
The production of particles in the UT may be a key component of the atmospheric budget
of optically and cloud-microphysically active aerosols, especially in pristine or relatively unpol-
luted regions, as was suggested in a modeling study by Merikanto et al. (2009). Studies in the
Amazon have shown that NPF almost never takes place under clean conditions in the PBL over
the Amazon Forest (Zhou et al., 2001; Martin et al., 2010; Andreae et al., 2015) and rarely oc-
curs over the taiga forest in remote Siberia (Heintzenberg et al., 2011). Over the Amazon, down-
ward transport of aerosols from the free troposphere (FT) has been identified as an important, if
not dominant, source of particles to the lower troposphere (LT) (Zhou et al., 2001; Roberts and
Andreae, 2003; Wang et al., 2016a). In turn, the concentrations of aerosols in the PBL have a
pronounced influence on the characteristics of convection and thereby influence cloud radiative





forcing and atmospheric dynamics (Sherwood, 2002; Rosenfeld et al., 2008; Fan et al., 2012;
Rosenfeld et al., 2014; Stolz et al., 2015; Cecchini et al., 2017).

Understanding the processes that control the aerosol burden in the pristine atmosphere is

an essential prerequisite for assessing the magnitude of the climate forcing by anthropogenic aer-
osols, since it forms the baseline from which anthropogenic forcing is derived. Because of the
strong non-linearity of the relationship between particle number concentration and cloud-medi-
ated aerosol effects, the uncertainty regarding the aerosol burden of the pristine atmosphere is the
largest contributor to the uncertainty in estimates of anthropogenic aerosol climate forcing
(Carslaw et al., 2013; Carslaw et al., 2017). For example, model calculations suggest that the in-
clusion of ion-induced particle formation from biogenic HOMs in the natural atmosphere reduces
the cloud-albedo radiative forcing by about one-third because of the higher albedo calculated for
the clouds in the pre-industrial atmosphere (Gordon et al., 2016).

In this paper, we present the results of aerosol measurements made in the upper tropo-

sphere across the Amazon Basin during the ACRIDICON-CHUVA campaign on the German
HALO aircraft during September and October 2014 (Wendisch et al., 2016). ACRIDICON
stands for "Aerosol, Cloud, Precipitation, and Radiation Interactions and Dynamics of Convec-
tive Cloud Systems"; CHUVA is the acronym for "Cloud Processes of the Main Precipitation
Systems in Brazil: A Contribution to Cloud Resolving Modeling and to the GPM (Global Precip-
itation Measurement)". We characterize these UT aerosol particles in terms of their microphysi-
cal and chemical properties, and contrast them with the LT aerosols. From their spatial distribu-
tion and their relationship to deep convection and convective outflow, we derive hypotheses
about their mode of formation. Finally, we discuss the role of upper tropospheric aerosol for-
mation in the life cycle of the atmospheric aerosol.

**2. Methods**

The observations discussed in this paper were collected aboard the HALO aircraft

(http://www.halo.dlr.de/), a modified Ultra Long Range Business Jet G 550 (manufactured by
Gulfstream, Savannah, USA). Because of its high ceiling altitude (up to 15 km) and long endur-
ance (up to eight hours with a scientific payload), HALO is capable of collecting airborne meas-
urements of cloud microphysical and radiative properties, aerosol characteristics, and chemical





tracer compounds in the upper troposphere, in and around tropical deep convective clouds. The
aircraft and its instrumentation are described in the ACRIDICON-CHUVA overview paper by
Wendisch et al. (2016).

In-situ meteorological and avionics data were obtained at 1 Hz from the BAsic HALO

Measurement And Sensor System (BAHAMAS). This data set includes pressure, temperature,
wind direction and speed, humidity, water vapor mixing ratio, aircraft position, and altitude. All
concentration data have been normalized to standard temperature and pressure (T = 273.15 K
and p =1000 hPa).

### 171    2.1. The HALO aerosol submicrometer inlet (HASI)

All aerosol sampling was conducted using the HALO aerosol submicrometer inlet

(HASI), designed for HALO by the German Aerospace Center (DLR) in collaboration with en-
viscope GmbH (Frankfurt, Germany) with the aim of providing up to 30 l min$^{-1}$ sample air flow
(divided over four sample lines) to aerosol instruments mounted inside the aircraft cabin. HASI
samples the air on top of the fuselage outside of the aircraft boundary layer. The air stream is
aligned in the inlet using a front shroud and decelerated by a factor of approximately 15. Four
sample tubes with 6.2 mm outer diameter and frontal diffusors protrude into the decelerated air
stream. The design goal is to allow regulating the sample airflow in each of the four sample lines
to achieve isokinetic sampling conditions according to the actual speed of the aircraft. Since the
automatic adjustment had not been implemented at the time of the field experiment, the flow was
fixed to values providing near-isokinetic sampling for typical flight conditions based on geomet-
ric considerations and preliminary flow simulations for the initial design of the inlet. The geo-
metric design should prevent large cloud droplets and ice crystals from entering the sample lines
directly. The inlet position is located in the shadow zone for larger ice crystals, which precludes
artifacts by shattering and break-up of larger ice particles at the inlet tip (Witte, 2008). Judging
from the first measurements with HASI, it appears that measurements of interstitial aerosol in
liquid clouds are affected by artifacts, while in ice clouds there is no indication for such artifacts.
The data selection procedures to exclude artifacts are discussed in section 2.2.





## 2.2. Condensation nuclei

Condensation nuclei (CN) number concentrations ($N_{CN}$) were measured using the Aerosol Measurement System (AMETYST). This system was designed to provide an instrument package for HALO to measure basic microphysical properties of the ambient atmospheric aerosol (integral number concentration, sub-micrometer size distribution, fraction of non-volatile particles, and particle absorption coefficient). AMETYST includes four butanol-based condensation particle counters (CPCs, modified Grimm CPC 5.410 by Grimm Aerosol Technik, Ainring, Germany) with flow rates of 0.6 and 0.3 l min$^{-1}$, configured with different nominal lower cutoff diameters at 4 nm and 10 nm (set via the temperature difference between saturator and condenser). In addition, two differential mobility analyzers (Grimm M-DMA) with a nominal size range between 5.5 and 350 nm using $^{241}$Am radioactive sources as aerosol neutralizers are part of the system.

Two of the four CPCs are generally set to measure the integral particle concentrations, while for the two other CPCs the configuration is selectable depending on measurement priorities. They can be used either as detectors for the DMAs or for additional integral concentration measurements. The DMAs can either be set to select specific diameters or operated as a DMPS (differential mobility particle sizer) system scanning the size distribution at predefined diameter steps. The integration times at each step have to be chosen such that meaningful statistics can be achieved depending on the measurement strategy. AMETYST also includes an optional thermodenuder, which heats a section of the sample line to 250°C for the measurement of the non-volatile particle fraction.

The raw CPC data are corrected using an empirical, pressure-dependent flow correction to account for changes in the volume flow at different flight altitudes (D. Fütterer, PhD thesis, in preparation). Particle losses in the sampling lines have been estimated with the particle loss calculator by von der Weiden et al. (2009). Accounting for these effects leads to an increase of the effective cutoff diameter for all CPCs. The effective cutoffs are calculated as a convolution of the pressure-dependent CPC counting efficiency and the size-dependent transmission efficiency of the sample lines. The data reported here were taken by the CPC operated at 0.6 l min$^{-1}$, with a nominal cutoff of 4 nm. Due to inlet losses, the effective cutoff diameter increases to 9.2 nm at 1000 hPa, 11.2 nm at 500 hPa, and 18.5 nm at 150 hPa. This implies that the present setup of





AMETYST essentially does not detect nucleation mode particles below 10 nm at low altitudes
and below 20 nm in the UT. Typical uncertainties of CPC number concentration measurements
are estimated to be of the order of 5 to 10% (Petzold et al., 2011).
To eliminate artifacts from cloud hydrometeors and bias from local pollution, we ex-
cluded measurements using the following criteria: (1) All cloud passages below 6 km were re-
moved. During passages through water clouds, the CPCs showed erratic, unreasonably high
number concentrations that are probably caused by droplet shattering at the probe tip. Cloud pas-
sages were identified from the observation of elevated concentrations of particles >3 μm using
the hydrometeor probes (see below). (2) In the mixed phase and ice phase regimes, all cloud pas-
sages were inspected for possible shattering artifacts, and suspect data were rejected. Cloud pas-
sages through pure ice clouds did not show evidence of hydrometeor shattering. (3) The flight
segments during departure and approach to Manaus airport were removed to avoid pollution
from the airport and its surroundings. (4) Flights segments through the Manaus urban plume,
which was sampled during joint flight experiments with the DOE G1 aircraft and in the course of
tracer studies in the PBL, were excluded in order to provide a sampling representative of the dry
season atmosphere over the Amazon Basin away from local pollution. (5) Fire plumes that were
sampled deliberately to study fresh emissions were not analyzed for this paper. (6) Segments
where the aircraft passed through its own exhaust were also excluded from the data analysis.
**2.3. Aitken mode aerosol size spectra**
To obtain aerosol size spectra for particle sizes up to 300 nm, the DMAs within
AMETYST were connected to two of the CPCs and operated in scanning mode for selected
flight sequences (especially during longer flight legs, where relatively homogeneous conditions
can be assumed). The size range covered by the scans was typically between 20 and 300 nm di-
ameter in nine steps. To improve the time resolution, the two DMPS were usually set to scan the
same sequence in opposite direction. The DMPS data are then analyzed taking into account a
correction for multiple charges following Wiedensohler (1988) after correcting the measured
concentrations to standard atmospheric conditions. To derive modal parameters for the particle
size distribution, a bi-modal log-normal fit to the data points was computed.



## 2.4. Accumulation mode aerosol particles


For the purposes of this paper, we define the accumulation mode as the particle size range

from 90 nm to 600 nm and the total number concentration in this size class as the accumulation
mode number concentration, $N_{acc}$. The particle concentrations in this range were measured with
an optical particle counter (OPC), the Ultra High Sensitivity Aerosol Spectrometer (UHSAS;
Droplet Measurement Technologies, Inc., Longmont, CO) (Cai et al., 2008; Brock et al., 2011).
The UHSAS combines a high-power infrared laser ($\lambda$=1054 nm) and a large solid angle range in
sideways direction for the detection of light scattered by individual particles. Due to the resulting
almost monotonic increase of instrument response with particle size, the UHSAS enables high-
resolution measurements (100 selectable channels). The high laser intensity enables the detection
of particle diameters down to about 60 nm, with the upper limit being approximately 1 μm. Parti-
cle concentrations of up to 3000 cm$^{-3}$ are recorded without significant counting coincidence
losses (Cai et al., 2008). The airborne instrument version is mounted in an under-wing canister
and equipped with a forward facing diffusor inlet. The slowed airflow is subsampled by a second
inlet at approximately isokinetic conditions. The sample is not actively dried before the measure-
ment, but due to combined heating effects the measured diameters can be assumed to be close to
their dry diameters (Chubb et al., 2016). The UHSAS was calibrated with monodisperse polysty-
rene latex (PSL) spheres of known refractive index and size. The evaluation of the OPC calibra-
tion results and the derivation of realistic uncertainty estimates for the OPC size distributions is
outlined in a recent study by Walser et al. (2017). Due to changes in the laser and instrument pa-
rameter settings during the campaign, only the size range from ~90 nm to ~600 nm is considered
here.

## 2.5. Cloud condensation nuclei


The number concentration of CCN ($N_{CCN}$) was measured with a continuous-flow stream-

wise thermal gradient CCN counter (CCNC, model CCN-200, DMT, Longmont, CO, USA)
(Roberts and Nenes, 2005; Rose et al., 2008). The CCN-200 consists of two columns, in which
particles with critical supersaturations (S) above a preselected value are activated and form water
droplets. Droplets with diameters $\geq$ 1 μm are detected by an optical particle counter (OPC) at the
exit of the column. The inlet flow rate of the column used was 0.5 L min$^{-1}$ with a sheath-to-aero-
sol flow ratio of 10. The water pump was operated at the CCNC setting of "high" liquid flow.





Variations in ambient pressure have a strong influence on the S inside the CCNC. For this pur-
pose, a novel constant pressure inlet without significant particle losses was deployed on HALO.
The instrument was calibrated before, during, and after the campaign at different pressures and
flow rates according to Rose et al. (2008). For the data used in this study, we sampled from the
HASI inlet and measured at S = 0.52±0.05% and a time resolution of 1 Hz.

Since the flow in the instrument was kept constant for the data used here, the error in $S$

resulted from the calibration uncertainty, as described by M. Pöhlker et al. (2016); it is estimated
to be in the range of 10%. According to Krüger et al. (2014), the error in $N_{CCN}$ is based on the
counting error of the measured particle number and is 10% of $N_{CCN}$ for large concentrations; given
that mostly low concentrations prevailed, the mean error was about 20% of $N_{CCN}$.
**2.6. Cloud droplet and ice particle measurements**

While measurements of liquid water and ice hydrometeor concentrations are not a subject

of this paper, they were used to determine whether the aircraft was sampling inside clouds and if
so, whether the cloud particles were liquid or frozen. For this purpose, we used data from the
Cloud Droplet Probe (CDP) and the Cloud and Aerosol Spectrometer (CAS-DPOL), both of
which are based on the principle of forward scattering detection. The CDP detects particles with
sizes from 3 µm to 50 µm, and classifies them into size histograms of bin widths between 1 and
2 µm. The CAS-DPOL covers the size range of 0.6-50 µm in 17 bins of varying width. The
probes are described in Voigt et al. (2017) and probes and data correction techniques in Weigel
et al. (2016).

Information regarding the ice particle properties was obtained from the Particle Habit Im-

aging and Polar Scattering Probe (PHIPS-HALO), a single-particle cloud probe that measures
microphysical and angular light scattering properties of individual particles (Abdelmonem et al.,
2016). The instrument is composed of a stereoscopic imager that takes two brightfield images
from the particles under a viewing angle difference of 120°. Simultaneously to collecting the im-
ages, the scattering part of the instrument measures the angular scattering function of the parti-
cles from 18° to 170° with an angular resolution of 8°. The optical resolution of the imager is
about 2.5 µm.



### 2.7. Aerosol mass spectrometer

For in-situ chemical analysis of submicrometer aerosol particles a compact time-of-flight
aerosol mass spectrometer (C-ToF-AMS) (Drewnick et al., 2005; Schmale et al., 2010) was op-
erated onboard HALO. The C-ToF-AMS was sampling through the HASI inlet for ambient aero-
sol measurements. The aerosol particles enter the instrument via a pressure-controlled inlet and
are focused into a narrow beam by an aerodynamic lens. In the vacuum chamber, the particles
are flash-vaporized and the resulting gas-phase molecules are ionized by electron impact. The
ions are guided into the Time-of-Flight mass spectrometer, separated by their mass-to-charge ra-
tio, and detected by a microchannel plate detector. The C-ToF-AMS was operated with a time
resolution of 30 seconds, providing mass concentrations of particulate organics, nitrate, sulfate,
chloride, and ammonium.

### 2.8. Refractory black carbon

An eight-channel Single Particle Soot Photometer (SP2) was used to detect and quantify
refractory black carbon (rBC) particles using laser-induced incandescence (Stephens et al., 2003;
Schwarz et al., 2006). The instrument measures the time-dependent scattering and incandescence
signals produced by individual aerosol particles when crossing a Gaussian-shaped laser beam
(Nd:YAG; $\lambda = 1064$ nm). The particles containing rBC cores absorb the laser light and evaporate
within the optical chamber emitting thermal radiation (incandescence). The peak intensity of the
incandescence signal, recorded by two photomultiplier tubes over two different wavelength inter-
vals, is linearly proportional to the mass of the rBC in the particle (Laborde et al., 2013). The in-
strument is sensitive to rBC cores in the nominal size range of 70 - 500 nm mass-equivalent di-
ameter, assuming a density of 1.8 g cm$^{-3}$. The SP2 also detects the intensity of the light scattered
by the particles using an avalanche photo-detector in order to determine the optical size of purely
scattering particles in the diameter range of 200 - 400 nm.
The SP2 incandescence signal was calibrated several times (at the beginning, during, and
at the end of the campaign) using size-selected fullerene soot particles. The scattering signal was
calibrated using either spherical polystyrene latex size standards (208, 244, and 288 nm) or am-
monium sulfate particles of different diameters selected by a differential mobility analyzer
(DMA).



## 2.9. Trace gases

Ozone ($O_3$) was measured by a dual-cell ultraviolet (UV) absorption detector (TE49C, Thermo Scientific) operating at a wavelength of 254 nm. Signal differences from a cell with the sample air and a parallel cell with ozone-scrubbed air are used to infer the concentration of $O_3$. Sample air was drawn into the instruments through the standard HALO gas inlet via a Teflon PFA line using an external pump at a nominal flow rate of 1 l min$^{-1}$. The calibration of the instrument is traceable to the $O_3$ standard of the Global Atmosphere Watch station at Hohenpeißenberg, Germany. The data output of the instrument is corrected for the temperature and pressure in the absorption cells. The precision of the $O_3$ measurements is 2% or 1 ppb, whichever is larger, the accuracy is 5%. Details on the use of this instrument can be found in Huntrieser et al. (2016).

Carbon monoxide (CO) was detected with a fast-response fluorescence instrument (AL5002, Aerolaser, Garmisch, Germany) (Gerbig et al., 1999). The detection of CO is based on the excitation of CO at 150 nm using a $CO_2$ resonance UV lamp. The fluorescence light is detected by a UV-sensitive photomultiplier. The CO detector was calibrated in-flight using onboard calibration and zero gas sources. Data are recorded at 1 Hz. The precision and accuracy are 3 ppb and 5%, respectively.

Nitrogen monoxide (NO) and total reactive nitrogen ($NO_y$) were measured by a dual-channel chemiluminescence detector (CLD-SR, Eco Physics). For the $NO_y$ channel, the chemiluminescence detector is combined with a custom-built Au converter which reduces all oxidized reactive nitrogen species to NO (Ziereis et al., 2000). Detection of ambient NO is performed via reaction with $O_3$ in a chamber and the luminescence signal of the excited $NO_2$ produced by this reaction. Both detector channels are equipped with a pre-reaction chamber for determination of cross-reactions of $O_3$ with interfering species. Sampling of ambient air is conducted via a standard HALO gas inlet using a Teflon line. The precision and accuracy of the measurements depend on the ambient concentrations, typical values are 5% and 7% (NO) and 10% and 15% ($NO_y$), respectively.

## 2.10. Trajectories and air mass history analysis

Backtrajectories were calculated for each minute, starting at the location of the HALO aircraft and using the FLEXPART ("FLEXible PARTicle") Lagrangian Particle Dispersion



Model version 9.02 (Stohl et al., 1998; Stohl and Thomson, 1999; Seibert and Frank, 2004; Stohl
et al., 2005). Trajectories were driven by six-hourly analyses, interlaced with the three-hour fore-
casts, from the Global Forecast System (GFS) of the National Centers for Environmental Predic-
tion (NCEP), provided on a 0.5 x 0.5 degree horizontal grid
(http://www.nco.ncep.noaa.gov/pmb/products/gfs/, last accessed 8 Sep 2016). For each trajec-
tory, 10,000 'particles' (infinitesimally small air parcels) are released and followed back in time
for 10 days. Sub-grid-scale processes like convection and turbulence act stochastically on each
'particle', resulting in a trajectory location probability distribution at each point in time. For con-
venience, the location probability distribution is simplified using a clustering algorithm, calculat-
ing five cluster centers of most probable trajectory locations (Stohl et al., 2002). Additional tra-
jectory calculations were performed using the HYSPLIT model (Stein et al., 2015) with NCEP
GDAS1 data and model vertical velocities.

We examined the history of the sampled airmasses for interactions with deep convection

using the FLEXPART trajectories and GOES (Geostationary Operational Environmental Satel-
lite) imagery. Every one-minute flight position was traced back in time in one-hour steps up to
120 hours. Each position was then matched in time to the closest GOES-13 (Geostationary Oper-
ational Environmental Satellite) infrared brightness temperature ($T_b$). As a proxy for deep con-
vection, we searched for cloud top $T_b$ below –30 ºC and looked up the minimum $T_b$ in a 1ºx1º
box around the center of the back-traced parcel. An example of this procedure is available in the
Supplement (Figs. S1-S3). From these data, we recorded the time difference between the mo-
ment that HALO was sampling the airmass and its encounter with deep convection, possibly in-
cluding multiple contacts with deep convection. We also noted the "deepest convection" (mini-
mum $T_b$) encountered by the parcels and their height at the time of the encounter, as well as the
number of hours that the parcel was within boxes with deep convection ($T_b < $ –30 ºC).
**3. Results and Discussion**
**3.1. The ACRIDICON-CHUVA campaign**

The ACRIDICON-CHUVA flights covered most of the Amazon Basin, reaching from the

Atlantic coastal waters in the east to near the Colombian border in the west, and from the Guy-



anas border in the north to the arc of deforestation in the south. The flight tracks of the flights an-
alyzed in this paper are shown in Fig. 1, where the flight segments at altitudes >8 km are shown
as heavier lines.

### 3.2. Synoptic situation and chemical context

3.2.1. Meteorological overview
During boreal summer, the Intertropical Convergence Zone (ITCZ) undergoes a seasonal
northward shift towards the northernmost part of South America, so that almost all of the Ama-
zon Basin is in the meteorological Southern Hemisphere. Examination of cloud top height and
precipitation images showed that the ITCZ was located between about 4 and 12 °N during the
campaign (6 Sep to 1 Oct 2014), but was often not very well defined over South America
(worldview.earthdata.nasa.gov, accessed 13 Jan 2017). This seasonal shift establishes the large-
scale thermodynamic conditions that define the dry season over the Amazon Basin, characterized
by synoptic-scale subsidence, a relatively dry planetary boundary layer (PBL) and mid-tropo-
sphere, and warm temperatures at the top of the PBL, resulting in elevated convective inhibition
energy (CINE) (Fu et al., 1999; Wang and Fu, 2007; Collow et al., 2016). During the dry season,
there is less shallow convection, cloud cover, and rainfall than in the wet season, but the convec-
tion that does occur is more organized with pronounced vertical development because of the sim-
ultaneous presence of high convective available potential energy (CAPE) and high CINE
(Machado et al., 2004; Collow et al., 2016; Giangrande et al., 2017; Zhuang et al., 2017). The
deep convective cloud fraction peaks in the late afternoon and evening (1600LT to 2400LT) with
a cloud fraction maximum between 9 and 13 km altitude and a minimum near and above the
freezing level between 4 and 7 km (Collow et al., 2016; Zhuang et al., 2017).
During the ACRIDICON-CHUVA campaign, the intense warm sea-surface temperature
(SST) anomaly that had earlier prevailed in the southern South Atlantic and a less intense cold
SST anomaly in the northern South Atlantic and near the Equator were strongly reduced, and a
warm SST anomaly in the equatorial Pacific was building to form the 2015 El Nino (see also
Martin et al., 2016). Consequently, the pattern of wind and omega (vertical motion) field anoma-
lies decreased to nearly normal conditions. However, during the campaign there was a clear
northeast-southwest contrast with drier conditions in the northeast and wetter ones in the south-
west, as seen in the columnar precipitable water anomaly data from the NCEP Climate Forecast



System Version 2 Reanalysis (Fig. 2) (Saha et al., accessed 20 March 2017). The majority of
HALO flights were over the drier anomaly or the neutral region. As a consequence of this drier
anomaly, these regions presented warmer temperatures and lower relative humidity than the nor-
mal climatology. The synoptic pattern during the campaign resulted in a spatial rainfall distribu-
tion with a meridional pattern, with more intense rainfall in the west, around 300 mm in Septem-
ber, and less than 100 mm in the eastern Amazon (Fig. 3). Nine cold fronts penetrated into Brazil
during September, however, only two moved northward and they had little interaction with Ama-
zon convection. Only the cold front on 20 to 23 September was able to organize convection in
the south of the Amazon Basin.
Figures 4a and 4b show the low (850 hPa) and high (200 hPa) level wind fields during
September 2014. The mean low-level flow at 850 hPa shows the typical easterly winds through-
out the Amazon Basin (Fig. 4a), decelerating near the Andes and curving to the subtropics. At
high levels (Fig. 4b), there is a weak anticyclonic circulation over the southern basin, featuring
the initial increased deep convection in the transition from the dry to the wet season (September)
and the development of the Bolivian High during the onset of the wet season (December to
March) (Virji, 1981; Zhou and Lau, 1998).
During the research flights, HALO reached maximum altitudes of 12.6 to 14.4 km a.s.l.,
corresponding to potential temperatures between 352 and 360 K (Fig. 5), i.e., the bottom of the
tropical tropopause layer (TTL). The vertical profiles of temperature and potential temperature
were remarkably consistent between the flights, showing a fairly stable stratification up to about
8 km and a slightly weaker gradient in potential temperature above this altitude. Relative humid-
ity shows a broad minimum in the region between 6 and 10 km. For comparison, the data from
radiosonde soundings at Manacapuru (a site southwest of Manaus) are provided in the supple-
ment (Fig. S4).
Based on the soundings, the mean height of the thermal tropopause during the campaign
was 16.9±0.6 km, corresponding to a potential temperature of about 380 K. During September
2014, the mean CAPE was 1536 J kg$^{-1}$ and the mean CINE value was 37 J kg$^{-1}$, the precipitable
water was 42 mm, the lifting condensation level 919 hPa, and the bulk shear 4.8 m s$^{-1}$ (difference
between the mean wind speed in the first 6 km and 500 meters). These values give a clear idea



about the typical cloud base expected, the high instability, the need of a forcing due to the CINE,
the high shear, and the amount of integrated water vapor.
In this paper, we use the following terminology to describe the different layers of the
tropical atmosphere: The region from the surface to the convective cloud base (typically about
1.2 to 1.7 km during mid-day) is the planetary boundary layer (PBL), above which is the convec-
tive cloud layer (CCL), which typically reached to altitudes of about 4-5 km during our cam-
paign. The region between the CCL and the TTL is the free troposphere (FT), which we subdi-
vide into the middle troposphere (MT) between about 5 km and 9 km and the and the upper trop-
osphere (UT) above ca. 9 km.
3.2.2. Airmass origins and history
For an overview of airmass movement in the UT over the central Amazon during the
campaign, we obtained trajectory frequency statistics for airmasses arriving at altitudes between
10 and 14 km over the central Amazon Basin. The frequency analysis indicated that airmass
movement in the upper troposphere was generally relatively slow and tended to follow anticy-
clonic patterns (Fig. 6), consistent with the 200 hPa streamlines shown in Fig. 4b. The frequency
diagram for the 72-h trajectories initialized at 12 km altitude (Fig. 6a) shows that most airmasses
had remained over the basin for the preceding three days (only about 1% of the endpoints fall
outside of the basin). The 10 and 14 km statistics show essentially the same patterns (Supplement
Figs. S5-S6), as do the individual trajectories calculated from the aircraft positions along the
flight tracks (not shown).
The 120-h trajectory statistics (Fig. 6b) and the examination of the individual trajectories
along the flight tracks indicate that the air sampled in the UT had followed a number of different
general flow patterns before being sampled by HALO: 1) flow from the Pacific with an anticy-
clonic loop of variable extent over the basin, ranging from almost zonal west-to-east flow to a
huge loop going as far south as Argentina and as far east as the Atlantic, and then returning to the
basin (types A and B in Table 1), 2) flow from the Atlantic, often almost zonal (type C), 3) inter-
nal circulation within the basin, usually along anticyclonic loops, but sometimes erratic (type D),
and 4) flow from the Caribbean, often following an anticyclonic pattern (type E). The flow pat-
terns of the UT airmasses that were enriched in aerosol particles are given in Table 1.



3.2.3. Atmospheric chemical environment

The atmospheric chemical environment over the Amazon Basin shows a pronounced sea-

sonal variation (Talbot et al., 1988; Andreae et al., 1990b; Talbot et al., 1990; Andreae et al.,
2002; Artaxo et al., 2002; Martin et al., 2010; Andreae et al., 2012; Artaxo et al., 2013; Andreae
et al., 2015). During the rainy season, regional biomass burning is at a minimum and biological
sources dominate trace gas and aerosol emissions in the basin, resulting in often near-pristine
conditions. The most significant pollution input during this season is long-range transport from
North and West Africa, which brings in a mixture of mineral dust and emissions from biomass
and fossil fuel burning (Talbot et al., 1990; Wang et al., 2016b). In contrast, ACRIDICON-
CHUVA took place during the dry season, when the Amazon Basin is impacted by a mixture of
pollution from regional and remote sources (Andreae et al., 1988; Talbot et al., 1988; Artaxo et
al., 2013). Deforestation and pasture-maintenance burning occurs throughout the basin, with the
highest intensity along the southern periphery, the so called "arc of deforestation". This creates a
steep gradient of pollutant concentrations from the relatively moist and less densely developed
northern and western basin to the drier and highly deforested and developed southern basin
(Andreae et al., 2012).

Long-range transport from Africa affects pollution levels over the Amazon, in addition to

regional sources. In the northern part of the basin, part of the 10-day backtrajectories arriving at
the aircraft positions in the lower troposphere reach West Africa, where biomass burning and
fossil-fuel emissions are prevalent, while other trajectories follow the northeastern coast of Bra-
zil, which is densely populated. As one moves south, the influence of long-range transport from
Southern Africa becomes more prevalent. This was clearly observed during flight AC19, which
extended over the Atlantic east of the Brazilian coast. On this flight, an extended, 300-m thick
layer of pollution at 4 km altitude was identified over the Atlantic with elevated rBC concentra-
tions up to 2 µg m$^{-3}$ (see section 3.4.4). The backtrajectories from the Amazon south of the Equa-
tor very frequently end in the central and eastern tropical Atlantic (see Fig. 3 in Andreae et al.,
2015), where high levels of ozone, aerosols, and other pollutants from biomass burning have
been documented by in-situ and satellite observations starting in the 1980s (Watson et al., 1990;
Fishman et al., 1991; Andreae et al., 1994; Browell et al., 1996; Fishman et al., 1996).



### 3.3. Vertical distribution of aerosol particle number concentrations over the Amazon Basin


Figure 7a shows a statistical summary of all CN number concentrations ($N_{CN}$) observed

during the campaign. Data affected by local pollution and cloud artifacts have been removed as
discussed in section 2.2. (Additional information about the flight segments on which elevated
$N_{CN}$ were encountered is provided in Table 1.) In the PBL, which typically reached heights of 1.4
to 1.8 km during the afternoon, mean $N_{CN}$ ranged from ~750 cm$^{-3}$ on the least polluted flights to
~4500 cm$^{-3}$ in the most polluted regions over the southern part of the basin. Above the PBL, CN
concentrations typically remained relatively high within the lower troposphere up to about 3-4
km, and then declined with altitude. $N_{CN}$ reached a minimum of ~700 cm$^{-3}$ at about 4-5 km alti-
tude everywhere over the basin. This aerosol minimum coincides with the minimum in cloud
cover that has been observed at and above the freezing level, which has been suggested to be as-
sociated with rain development by the Wegener-Findeisen-Bergeron process at this level
(Collow et al., 2016).

Above this level, we found a general increase in particle concentrations, such that above 8

km, $N_{CN}$ were typically in the range of 2000 to 10,000 cm$^{-3}$. This altitude corresponds approxi-
mately to the 340 K potential temperature level, above which elevated CN concentrations had
also been found in previous studies (Borrmann et al., 2010; Weigel et al., 2011).

While the statistical plot in Fig. 7a shows a general particle enrichment in the UT, indi-

vidual vertical profiles show more complex structures (Fig. 7b). The highest $N_{CN}$, sometimes
reaching up to 65,000 cm$^{-3}$, were encountered in thin layers often only a few hundreds of meters
thick. A distinct example for such a layer is seen in the descent profile (segment A2) from flight
AC09 (Fig. 4b), with peak CN concentrations of ca. 35,000 cm$^{-3}$. Other profiles, e.g., the descent
profile from flight AC07 (segment G), show enhancements over a layer about 3 km thick, with
$N_{CN}$ of 10,000 – 20,000 cm$^{-3}$.

The CN enrichments in the UT consist predominantly of ultrafine particles in the size

range below 90 nm. In contrast to $N_{CN}$, the enhancement of accumulation mode particles ($N_{acc}$,
defined here as the particles in the size range 90 to 600 nm) in the UT is much less pronounced.
The concentration of accumulation mode particles in the LT typically ranged from ~500 to
~3000 cm$^{-3}$, depending on the level of pollution (Fig. 8a). Like the vertical profile of $N_{CN}$, the
profile of $N_{acc}$ also shows a decrease above the LT to a minimum around 4-5 km, followed by an



increase towards the upper troposphere. Over the more polluted regions in the southern basin,
$N_{acc}$ in the UT was often considerably lower than in the LT.

Figure 8b illustrates the different behavior of CN and accumulation mode particle number

concentrations at the example of a sounding in the central Amazon Basin from flight AC19. In
the LT, $N_{CN}$ and $N_{acc}$ have similar values and decline to a minimum at about 4.7 km. Above this
altitude, $N_{CN}$ shows several sharp concentration peaks, with one at about 7.4 km reaching con-
centrations around 65,000 cm$^{-3}$. These peaks are only weakly, if at all, reflected in $N_{acc}$, which
shows a broad enhancement in the UT to values around 1000 cm$^{-3}$. Consequently, we find two
types of aerosol enrichments in the UT: at one extreme, thin layers with extremely high $N_{CN}$ val-
ues but no significant increase in particles larger than 90 nm, at the other, broad overall particle
enrichments with modest values of both $N_{CN}$ and $N_{acc}$.

**3.4. Differences between UT and LT aerosols**

The high concentrations of particles in the UT over the Amazon Basin beg the question of

their origin. Three different mechanisms can be considered: vertical transport of particles from
the PBL by deep convection, horizontal long-range transport from remote source regions, and in-
situ new particle formation. To assess these possibilities, we discuss in the following sections the
chemical and physical properties of the UT aerosols and contrast them with the LT aerosol.

A first argument against vertical transport as the dominant source mechanism for the

large particle concentrations in the UT comes simply from the observed CN concentrations.
Since we are using concentrations normalized to standard temperature and pressure, $N_{CN}$ should
not change with vertical transport alone, and the values measured in the UT should not exceed
those measured in the PBL. The fact that CN concentrations in the UT across the entire Amazon
Basin are higher than the PBL values we measured anywhere in the basin, often by very large
factors, rules out vertical transport of particles from the Amazon PBL as the dominant source of
UT particles.
3.4.1. Particle size

The particles in the UT have a very different size distribution from those in the LT, which

also shows that they could not have originated from upward transport of PBL aerosols by deep
convection. Unfortunately, a detailed analysis of the size distribution of the particles in the UT is



hampered by the significant losses of small particles in our inlet system. As discussed in section
2.2, the particle losses increase with altitude such that in the UT most of the particles below ca.
20 nm are lost in the inlet system before reaching the CPC. Because of a longer inlet tubing con-
nection and lower sample flow, the losses were even more significant for the DMPS, and as a re-
sult of this and other operational limitations, valid particle size distributions are only available
from the LT.
The DMPS measurements in the LT showed that the aerosol size distribution was domi-
nated by an accumulation mode centered at about 190 nm, flanked by an Aitken mode with a
maximum at about 80 nm (Fig. 9), in good agreement with the size distributions measured previ-
ously at ground level in the Amazon (Zhou et al., 2002; Rissler et al., 2006; Andreae et al., 2015;
Pöhlker et al., 2016) and those obtained over the Amazon on the G1 aircraft during the GoAma-
zon 2014 campaign (Martin et al., 2016; Wang et al., 2016a). For comparison, we show size
spectra from GoAmazon 2014 from Wang et al. (2016a), the only published size spectra from the
FT over central Amazonia. Unfortunately, even these data reach only up to 5.8 km, the ceiling
altitude of the G1 aircraft. In the PBL, these spectra were similar to our measurements from the
LT. With increasing altitude, total particle concentrations increased and the size spectrum be-
came dominated by an Aitken mode at ca. 50 nm (Wang et al., 2016a). A previous study over the
northern Amazon in Suriname had also found a decrease in the modal diameter of the Aitken
mode from ~70 nm in the LT to ~ 30 nm in the UT above 10 km (Krejci et al., 2003). Assuming
that similar size distributions prevailed in the UT during ACRIDICON-CHUVA and given the
fact that inlet losses limited our measurements to particle diameters >20-30 nm, it seems justified
to conclude that our $N_{CN}$ concentrations in the UT are actually lower limits and that the true con-
centrations might have been significantly higher.
In the absence of full size spectra, we use the ultrafine fraction [UFF, defined as the frac-
tion of particles with diameters between 90 nm (the lower cutoff of the UHSAS) and ~20 nm
(the lower cutoff of the CPC), i.e., UFF = $(N_{CN}-N_{acc})/N_{CN}$] as a metric for the contribution of the
Aitken and nucleation modes to the total observed particle concentration. The summary profile
plot (Fig. 10a) shows the dramatic difference between the UFF in the LT and UT: In the LT, the
mean UFF is about 0.05 to 0.2, showing the dominance of the accumulation mode. The share of
ultrafine particles increases throughout the middle troposphere, and in the UT they account for
the vast majority of particles, with UFF values around 0.7 in regions where both $N_{acc}$ and $N_{CN}$ are



moderately enriched, and values approaching 1.0 in the layers with very high $N_{CN}$. This shows
up more clearly in individual profiles, e.g., the soundings in Fig 10b from flight AC18. The
highly enriched layers are represented by UFF peaks in the range of 0.7 to 1.0, whereas the back-
ground UT enrichment exhibits UFF values of 0.5 to 0.8. The highest UFF values were measured
in the very young aerosol layer in segment E2 at 13.5 km (Fig. 10b), with an estimated particle
age of about 1-5 hours (more on this layer in section 3.5.2).
### 3.4.2. Cloud nucleating properties
The cloud nucleating ability of aerosol particles depends both on their size and their
chemical composition. Here we focus on CCN concentrations at 0.52% supersaturation ($N_{CCN0.5}$),
which are dominated by the particles in the accumulation mode size range, but also include a
fraction of the Aitken mode. A full discussion of the CCN measurements during ACRIDICON-
CHUVA will be presented elsewhere (M. Pöhlker et al., 2017, in preparation).
Figure 11a shows the vertical distribution of CCN for the entire campaign, indicating
strong variability in the LT, a minimum at ca. 5 km, and elevated concentrations in the UT. The
$N_{CCN0.5}$ variability in the LT was related to the variable level of pollution, mostly from biomass
burning, which was much higher in the southern part of the basin than in the north. In contrast,
there was no systematic difference between the CCN concentrations in the UT above polluted
and relatively clean regions. Therefore, depending on the level of pollution in the lower tropo-
sphere, the $N_{CCN0.5}$ in the UT during our campaign were higher or lower than those in the LT.
This is illustrated at the example of $N_{CCN0.5}$ profiles from Flights AC09 and AC12+13, from a
clean region (AC09) and one polluted by biomass burning emissions (AC12+13), respectively
(Fig. 11b). While there was a large difference in the CCN concentrations in the LT, the values in
the UT were very similar between these flights, indicating that the CCN enrichments in the UT
are independent of the pollution levels in the LT.
The CCN concentrations at a supersaturation S=0.52% in the UT were consistently
greater than the corresponding values of accumulation particle number concentrations, $N_{acc}$, re-
sulting in a median $N_{CCN0.5}/N_{acc}$ ratio of 1.66 (quartile range 1.32 – 2.32, N=53,382) above 8 km.
This implies that some of the particles smaller than 90 nm are also able to nucleate cloud drops at
S=0.52%. Because size-selective CCN measurements were not performed during ACRIDICON-
CHUVA, it was not possible to derive the actual critical diameters and hygroscopicity factors ($\kappa$,



Petters and Kreidenweis, 2007) for the CCN on this campaign. However, a consistency check
can be made using the measured chemical composition. As will be discussed in detail in section
3.4.4, the UT particles consist predominantly of organic material, with minor amounts of nitrate
and very small fractions of sulfate. The hygroscopicity of particles consisting completely of or-
ganic matter can vary greatly, with $\kappa$ between near 0 and about 0.3 (Jimenez et al., 2009). Our
AMS measurements (see section 3.4.4) showed that the UT secondary organic aerosol (SOA)
contains a substantial fraction of organics derived from the oxidation of isoprene (IEPOX-SOA)
(Schulz et al., 2017), which has relatively high hygroscopicity ($\kappa \geq 0.1$) (Engelhart et al., 2011;
Thalman et al., 2017). Assuming a conservative value of $\kappa_{org} \cong 0.1$, which had been found previ-
ously for the Amazon PBL (Gunthe et al., 2009; Pöhlker et al., 2016), pure SOA particles would
have to have diameters of $\geq 80$ nm to act as CCN at 0.52% supersaturation, whereas for pure am-
monium sulfate particles ($\kappa \cong 0.6$), the critical diameter would be ca. 45 nm (Petters and
Kreidenweis, 2007). At a typical organic mass fraction of 0.8 for the UT aerosol (see section
3.4.4), an effective $\kappa$ of ca. 0.2, corresponding to a critical diameter of ~65 nm, is likely. Given
the expected steep increase in particle concentration between the $N_{acc}$ cutoff of 90 nm and the es-
timated critical diameter of 65 nm, a $N_{CCN0.5}/N_{acc}$ ratio of the observed magnitude appears thus
quite reasonable.
The vertical distribution of the CCN fraction, i.e., the ratio $N_{CCN0.5}/N_{CN}$, shows a pro-
nounced decrease with altitude (Fig. 12a), reflecting the smaller particle size in the UT. It also
exhibits a strong inverse relation to the total particle concentration, $N_{CN}$. This is illustrated at the
example of flight AC18 (Fig. 12b), where the data from different flight segments are plotted.
Segments A and F (yellow and orange) are from soundings in the somewhat more polluted cen-
tral part of the Amazon Basin, while B and C (green) are from the cleaner westernmost part and
show the lowest CCN concentrations and the highest CCN fractions. Both soundings have high-
CN layers at altitudes between 7 and 13 km, with $N_{CN}$ up to almost 23,000 cm$^{-3}$, and correspond-
ingly low $N_{CCN0.5}/N_{CN}$. Segment E2 (red) is from a layer that was intercepted downwind of a
massive convective complex, with a transport time of 1-5 hours between the anvil and the air-
craft (see section 3.5.2). This layer had $N_{CN}$ values up to 45,000 cm$^{-3}$, CCN fractions down to
0.01, and UFF $\cong 0.98$, suggesting that these recently formed particles were too small to act as
CCN. This layer was embedded in a region of moderately elevated CN (segment E1 at 13-14 km;
lilac), which had much higher $N_{CCN0.5}/N_{CN}$ (0.2-0.5) and lower UFF (0.6-0.8), indicating larger



particle sizes and likely a more aged aerosol. Segment D (blue), at 11-12 km altitude, had similar
properties to E1. These observations point to the presence of two distinct aerosol populations in
the UT. At one extreme are aerosols with very high $N_{CN}$ and ultrafine fractions and low CCN
fractions (e.g., E2), presumably representing newly formed particles with sizes too small to act as
CCN. At the other extreme are populations with modest $N_{CN}$, but high UFF and CCN fractions,
indicating a more aged aerosol with larger particles (e.g., E1 and D).

The existence of these two populations is confirmed in plots of $N_{CCN0.5}$ and $N_{CCN0.5}/N_{CN}$

against supersaturation. Examples are shown in Fig. 13a and 13b, with AC18-DD representing a
segment dominated by larger and aged particles, AC07-F a region with high concentrations of
small and younger particles, and AC09-AA a mixed case with short periods of very high $N_{CN}$
over a background of moderately elevated particle concentrations. Even though the mean CN
concentration exceeds 8900 cm$^{-3}$ in AC07-F, the mean $N_{CCN0.5}$ in the same region is only 13 cm$^{-3}$
and therefore the $N_{CCN0.5}/N_{CN}$ vs. S plot falls essentially on the baseline. In contrast, AC18-DD
presents a fairly "classical" supersaturation spectrum, and AC09-AA is a mixed case with the
measurements made during the $N_{CN}$ peaks showing very low $N_{CCN0.5}/N_{CN}$.

In Fig. 13c and 13d, we compare the mean supersaturation spectra from the lower, middle

and upper troposphere obtained on flights AC12 and AC13, which were taken on successive
days over the same region and where the LT was influenced by biomass burning pollution. In the
LT, the CCN fraction is in the range observed at ground level at the Amazon Tall Tower Obser-
vatory (ATTO) site (Pöhlker et al., 2016) and in close agreement with measurements in the
southern Amazon during the biomass burning season (Vestin et al., 2007). In the UT, we ob-
served low CCN fractions representing the regions with high $N_{CN}$ and UFF, mostly at altitudes of
10-11 km, and higher CCN fractions at 12 km and above corresponding to a region with elevated
CCN (cf. Fig. 11b, which shows the CCN concentrations from these flights). In the middle tropo-
sphere (5-8 km) we found intermediate CCN fractions, consistent with a mixture of LT and UT
aerosols.
3.4.3. Volatility

On several flights (AC16, 18, 19, and 20), a second CPC was operated behind a ther-

modenuder at a temperature of 250 °C, in parallel to the regular CPC. The results of these meas-
urements are shown in Fig. 14a in the form of the volatile fraction (VF), i.e., $(N_{CN} - N_{nonvol})/N_{CN}$,





plotted against altitude. In the LT, most particles are nonvolatile and the VF is typically between
10 and 20%. This is consistent with the behavior described by Clarke and Kapustin (2010) and
Thornberry et al. (2010), who found that aged combustion aerosols (from biomass of fossil-fuel
burning) are non-volatile and mostly in the accumulation mode size fraction. With increasing al-
titude, the VF increases, closely resembling the profile of the UFF. In the UT, the mean VF
reaches about 80%, and approaches 100% in the most highly enriched layers (e.g., segment E2).
In previous campaigns, high volatile fractions had also been observed in the tropical UT and
TTL, with the highest VF in the region between 340 and 360 K potential temperature, corre-
sponding to about 9-15 km (Borrmann et al., 2010; Weigel et al., 2011).
More detail can be seen when looking at data from an individual flight. In Fig. 14b we
show the profiles from AC18, which we had already discussed in the context of CCN concentra-
tions in the previous section. The profiles (segments A, B, C, and F) show the overall increase in
VF with height, with peak values at embedded high-CN layers. The freshest layer (E2), which
had the highest UFF, also has the highest VF. In contrast, segments D and E1, representing larger
UT regions with moderate CN enrichments, larger particles, and higher CCN fraction also have
lower VFs, between 0.4 and 0.7. A contribution from aged combustion aerosols can be ruled out
as source for the non-volatile particles in these layers, because the rBC concentrations are close
to zero (see below). As we will show in the next section, it appears that these low-volatility parti-
cles represent a more aged organic aerosol.
3.4.4. Chemical composition
As discussed above, the LT aerosol over the Amazon during the dry season is dominated
by the products of biomass burning, with increasing concentrations from north to south. This is
clearly reflected in its chemical composition, which is dominated by carbonaceous matter (or-
ganic and elemental carbon) and only contains minor fractions of inorganic species, such as po-
tassium, sulfate, and nitrate. Elemental or black carbon is a unique tracer of combustion emis-
sions and was measured on HALO in the form of refractory black carbon (rBC).
The vertical profile of rBC shows a sharp separation between LT and FT (Fig. 15). The
average rBC concentration in the region below 5 km was $0.31\pm0.29$ µg m$^{-3}$, whereas in the FT
above 6 km it was $0.0026\pm0.0069$ µg m$^{-3}$ in terms of mass concentrations, and $99\pm92$ cm$^{-3}$ vs.
$1.5\pm2.5$ cm$^{-3}$ in number concentrations of rBC particles. Interestingly, these concentrations over



the Amazon Basin are only slightly higher than the values measured over the tropical Western
Atlantic during the Saharan Aerosol Long-range Transport and Aerosol-Cloud-Interaction Ex-
periment (SALTRACE), June/July 2013: ca. 0.2 μg m$^{-3}$ in the LT and ca. 0.001 μg m$^{-3}$ in the FT
(Schwarz et al., 2017), which suggests that a significant fraction of the rBC is entering the basin
by long-range transport from Africa.
In 14 instances, elevated rBC concentrations were seen for short durations (usually less
than 30 sec) in the UT. Most of the time, they occurred during cloud penetrations in the course of
vertical cloud microphysics profiling. In the case of the flights over the northern half of the Ama-
zon Basin, they could likely be attributed to sampling of HALO's own exhaust, based on the
flight track and the presence of associated NO enhancements in the absence of strong enhance-
ments of CO and other aerosol species (CCN, $N_{acc}$, $N_{CN}$). On flights over the southern Amazon
(AC07, AC12, AC13, and AC20), where the PBL was more polluted and active fires were pre-
sent, there were a few instances when elevated rBC coincided with peaks in CO and accumula-
tion mode particles, which suggests upward transport of biomass smoke aerosols. In view of the
scarcity of such events during our campaign and their modest rBC concentrations, it is clear that
they do not represent a major source of combustion aerosol for the UT during our campaign. No
elevated rBC concentrations were observed during the extensive outflow sampling legs on any of
the flights. A detailed discussion of the rBC measurements during the campaign will be pre-
sented in a companion paper (Holanda et al., 2017).
The drop in rBC concentration by two orders of magnitude between LT and FT implies
that rBC, and by extension other aerosols (which are likely even more prone to being removed
by nucleation scavenging), are efficiently removed during deep convection and consequently that
there is little transport of LT aerosols into the FT. Consequently, enrichments in $N_{CN}$ and $N_{acc}$ in
the FT cannot be explained by vertical transport of particles from the FT.
The AMS measurements also show pronounced differences in the composition of the LT
and UT aerosols (Fig. 16). In Table 2 we present a detailed analysis of the results from three
flights, AC07 from a polluted region in the southern Amazon, and AC09 and AC18 from rela-
tively clean regions in the northern and northwestern parts of the Basin, respectively. Organic
aerosol (OA) is the dominant aerosol species in all three regions at all altitudes, as expected in an





area where biomass burning and secondary organic aerosol (SOA) production are the dominant
sources.
In the LT, (ammonium) sulfates (SO4) are together with rBC the next most important
species. Here, we see a clear difference between the BB-dominated region in the south (with
high OA, ammonium [NH4], and rBC, and relatively low SO4) versus the northern basin, where
SO4, likely from long-range transport, plays a more important role. The ratio OA/rBC in the LT
is in the range 3-11, consistent with values from BB aerosols. The biomass burning marker, $f_{60}$
(Schneider et al., 2006; Alfarra et al., 2007), is present in all the measurements from the LT, but
always mixed with oxidized secondary organics. It should also be noted that the $f_{60}$ marker is not
an inert tracer but decays with time, and an observed background level of the $f_{60}$ tracer is 0.3% of
OA (Cubison et al., 2011).
In the UT, SO4 shows lower concentrations than in the LT, with the most pronounced
difference on flights AC07 and AC18. The latter flights also show a large difference in the
OA/SO4 ratio, which is around 10 in the UT and around 2 in the LT. Because of the high BB
component in flight AC07, this ratio is also high in the LT on this flight. The most pronounced
differences between UT and LT are seen in the nitrogen species. Ammonium is usually present
in the BL, sometimes at considerable levels (e.g., on AC07), but always below the detection limit
in the UT. In contrast, nitrate (NO3) is a minor species in the LT, whereas in the UT it is compa-
rable or greater than SO4, so that the ratio NO3/SO4 is about an order of magnitude higher in the
UT than in the LT. High concentrations of organics, especially oxidized organics, and nitrate had
been seen previously in the UT by Froyd et al. (2009).
The nature of the nitrate signal in the UT cannot be definitely identified from our data.
The absence of NH4 and the ratio of the peaks associated with ammonium nitrate make it un-
likely that the NO3 signal represents ammonium nitrate (Fry et al., 2009; Bruns et al., 2010). It
may be, at least to a large part, indicative of organonitrates, which have been shown to account
for 15-40% of SOA mass in laboratory experiments (Berkemeier et al., 2016) and whose for-
mation is enhanced at low temperatures (Lee et al., 2014).
A closer look at the aerosol-enriched layers in the UT from these flights reinforces these
conclusions (Table 2). In these layers, the ratios OA/SO4 and NO3/SO4 can reach very high val-
ues, especially in the SO4-poor UT of flight AC07. On flights AC09 and AC18, we encountered



extended periods when $N_{acc}$ and $N_{CCN0.5}$ were elevated, while $N_{CN}$ did not show extremely high
values (AC09-AA, AC18-AA, and AC18-DD). The AMS data from these segments were gener-
ally similar to the UT averages, suggesting that they are representative of the ambient UT aero-
sols. The layers with very high $N_{CN}$ on these flights (AC09-BB, AC09-EE, AC09-A1+A2, and
AC18-A1, AC18-A2, AC18-E2, AC18-F) also did not show significant differences from the UT
means on these flights, likely because the numerous, but very small CN in these layers did not
contain enough mass to influence the measurements in a detectable way.

We attempted to examine this hypothesis further by investigating the size dependence of

the AMS signals, but because of the small aerosol mass concentrations in the UT, size infor-
mation from the AMS data required extended integration periods, which precluded obtaining size
data from the relatively short segments with very high $N_{CN}$. The most robust size data were from
the segments where relative high $N_{acc}$ concentrations prevailed over extended periods of time,
e.g., segment DD (Table 2) on flight AC18. Here, the organic aerosol (OA) showed a broad
mode between 80 and 250 nm, with a modal diameter at 150 nm. This confirms that the AMS
compositional data are dominated by the accumulation mode, while the particles that make up
most of the UF fraction in the UT do not have enough mass to provide a clear AMS signal. An
exception may be some segments on AC09 (BB and EE), where OA and NO3 data suggest a
mass mode between 60 and 120 nm. Here, the UFF is quite high (0.85 and 0.92, compared to
segment DD on flight AC18, where it was 0.61) suggesting a smaller and therefore younger aer-
osol population.

More detailed information on the origin of the organics in the UT aerosol can be obtained

from specific markers. In the UT, the BB marker $f_{60}$ is typically not detectable, which in combi-
nation with the fact that the ratio OA/rBC is of the order of 1000, precludes a significant contri-
bution of aerosols from biomass burning or other primary combustion aerosols to the OA in the
UT. In contrast, the marker $f_{82}$, which is indicative of IEPOX-SOA formed by the photooxidation
of isoprene (Robinson et al., 2011; Hu et al., 2015), is found in the aerosol-enriched layers in the
UT, suggesting oxidation of isoprene and other biogenic volatile organic compounds (BVOC) as
source of the OA (Schulz et al., 2017). The plot $f_{43}$ vs. $f_{44}$ is frequently used to represent the ag-
ing of organic aerosols (Ng et al., 2011). In Fig. 17, we show the median locations of the LT and
UT aerosol in this plot, which indicates that both are fairly well aged and oxidized, with the UT
data plotting slightly towards less oxidized and younger values. This may reflect an overall



younger aerosol, or the admixture of recent material either by condensation on the accumulation
mode particles or in the form of an external mixture of larger aged particles with small younger
ones. The individual segments from flight AC18, which had the lowest OA/SO4 and NO3/SO4
ratios, also plot in this region, showing that they are dominated by a relatively well-aged aerosol.
In contrast, segments AC09-AA, and AC07-AA1, AC07-AA2, and AC07–GG, which have the
highest OA/SO4 and NO3/SO4 ratios and much higher $N_{CN}$, plot much further to the lower right
indicating a less oxidized, fresher aerosol. On this flight, the concentrations of accumulation
mode aerosols in the UT were relatively low, so that freshly formed aerosol could be more evi-
dent because of a lower background of aged aerosol.
In summary, the chemical composition data show that, while both LT and UT aerosols
are dominated by aged organics, their sources must be different because the UT aerosol is essen-
tially devoid of the combustion tracers, rBC and $f_{60}$, whereas the OA/rBC ratios in the LT are
consistent with combustion aerosols. Nitrate is strongly elevated in the UT, and may consist to a
large extent of organonitrates. NH4 is a significant component in the LT, whereas it is below the
detection limit in the UT. Size-selective chemical analysis is difficult because of the low aerosol
mass concentrations, but the available data suggest that the AMS measurements are dominated
by the accumulation mode, and the strong $N_{CN}$ enhancements are not distinctly seen in the AMS
data. Chemical marker analysis shows the general absence of BB tracers in the UT, while the
marker $f_{82}$ indicates production of IEPOX-SOA from isoprene. Most of the UT organics are aged
and oxidized, but in some of the CN-enriched layers, younger and less oxidized OA was evi-
denced by much lower $f_{44}/f_{43}$ ratios. A detailed discussion of the AMS measurements during
ACRIDICON-CHUVA will be presented in Schulz et al. (2017).
**3.5. Relationship to Deep Convection**
In the preceding section, we have documented the differences between the aerosols in the
LT and the UT, which rule out the possibility that convective transport of PBL aerosols can be an
important source for the UT aerosols. This opens the question about the source of these particles:
are they the result of long-range transport from remote sources or do they originate over the Am-
azon Basin? In the latter case, are they directly released in the outflow from the convective
clouds or are they produced by subsequent nucleation and growth in the UT?





For the larger particles in the accumulation mode, represented by elevated $N_{acc}$ and
$N_{CCN0.5}$ in the UT, long-range transport cannot be excluded, because such particles can have long
lifetimes in the upper troposphere (Williams et al., 2002). While the absence of detectable rBC
still rules out an origin from pollution aerosols lofted from the LT, they may have formed days or
weeks ago by gas-to-particle formation mechanisms anywhere in the free troposphere. In con-
trast, the high concentrations of small UF particles that we observed with high frequency in the
UT cannot come from distant sources, as they persist only for hours to a few days before grow-
ing to larger sizes and decreasing in concentration due to coagulation and dilution processes
(Williams et al., 2002; Krejci et al., 2003; Ekman et al., 2006).
3.5.1. Aerosols in cloud tops, anvils and outflows
First, we consider the possibility of these particles having been produced already inside
the clouds and released by outflow into the UT. In earlier studies, NPF had been shown to occur
in ice clouds in the tropical/subtropical UT, especially in conditions where the available surface
area of ice particles was relatively low (e.g., Lee et al., 2004; Frey et al., 2011). To look for this
phenomenon, we examined the particle concentrations during passages through the upper levels
of deep convective clouds and in the anvils directly attached to active cumulonimbus clouds
(Cb). Our measurements during these passages consistently show lower CN and CCN concentra-
tions than in the surrounding UT air, as exemplified in Fig. 18a by data from flight AC18. Dur-
ing this flight segment we performed multiple penetrations of the tops of growing Cb at altitudes
between 10.7 and 12.0 km and temperatures in the range of 225 to 236 K. During each cloud
passage (indicated in Fig. 18a by the ice particle concentrations) the aerosol concentrations de-
creased sharply, to values of $N_{CN}$ around 800 cm$^{-3}$ and $N_{CCN0.5}$ around 250 cm$^{-3}$ during the longer
cloud passages. (Here, we use $N_{CCN0.5}$ as proxy for the accumulation mode particles, since the
$N_{acc}$ measurements in clouds were perturbed by shattering at the probe tip, whereas the $N_{CN}$ and
$N_{CCN0.5}$ measurements showed no artifacts in ice clouds.) In the case of $N_{CN}$, the values in the
cloud tops are about the same as the PBL concentrations measured in the same region, while for
$N_{CCN0.5}$ they are significantly lower than the PBL values of around 400 cm$^{-3}$.
The same behavior was found for all cloud penetrations in the UT during the campaign.
In particular, extensive cloud top and outflow sampling on AC09, AC15, and AC16 also showed



$N_{CCN0.5}$ values down to 160-250 $cm^{-3}$ and $N_{CN}$ values down to 600-1000 $cm^{-3}$. The lowest parti-
cle concentrations were seen in a large outflow sampled on AC13 (20:08-20:30 UTC), when
both $N_{CN}$ and $N_{CCN0.5}$ reached values below 50 $cm^{-3}$ (Fig. 18b). In this airmass, NO and $NO_y$
were strongly elevated indicating recent NO production by lightning in the large Cb from which
this outflow originated.
Given that the air sampled during the cloud passages had already mixed in by lateral en-
trainment some of the surrounding air with much higher particle concentrations (Bertram et al.,
2007; Yang et al., 2015), these low particle concentrations in the cloud tops and outflows are
clear evidence that in-cloud processes were a sink and not a source of particles in the size class
measureable with our instrumentation. A rough estimate of the scavenging efficiency of the con-
vective process can be gained by using CO as a conservative tracer. For example, on flight AC18
the PBL concentrations of CO and $N_{CN}$ averaged ~120 ppb and 780 $cm^{-3}$, and the UT during the
cloud penetrations around 1900 UTC had CO ~95 ppb and $N_{CN}$ ~1500 $cm^{-3}$. In the cloud, CO
rose to 108 ppb and $N_{CN}$ dropped to 750 $cm^{-3}$. Following the approach of Bertram et al. (2007),
we can estimate that the fraction of PBL air in the center of the cloud was ca. 0.52, and that with-
out scavenging, $N_{CCN0.5}$ would be ca. 1130 $cm^{-3}$. From these values, a scavenging loss of 90% or
more of CCN can be estimated, in good agreement with previous studies (e.g., Andreae et al.,
2001; Yang et al., 2015), and with the absence of detectable rBC.
Flight AC20 was the only exception to this behavior. Here, CN were strongly enhanced
during cloud passages and even CCN were slightly elevated in some passages. The cloud that
was sampled on this flight appears to have been a pyrocumulus that had been ingesting fresh bio-
mass smoke, as suggested by the strongly elevated CO during the cloud passages. This flight will
be discussed as a separate case study below (section 3.6.).
While these results show that the high particle concentrations we observed in the UT
were not directly released from the cloud tops, they do not rule out the possibility that new parti-
cle formation had already started in the clouds or anvils. This is because the newly formed parti-
cles observed in the earlier studies were almost exclusively in the size range below 20 nm (Lee et
al., 2004; Frey et al., 2011). Since our measurements are limited to particle sizes >20 nm, we
would not have been able to detect such freshly nucleated particles, and therefore the earliest
phases of particle nucleation and NPF over Amazonia will have to be addressed in future studies.



Our data do show, however, that release of particles by hydrometeor evaporation following deep
convection is not a net source of particles to the UT over Amazonia, in contrast to what was ob-
served over the Indian Ocean region by Engström et al. (2008). Because the $N_{CN}$ and $N_{CCN0.5}$
concentrations in the ambient air in the UT are actually higher than in the air detrained by the Cb
clouds, the detrainment leads at least initially to a reduction in UT particle concentrations in the
size class >20 nm. Only through subsequent NPF can this be reversed and deep convection then
become a net source of UT aerosols.
3.5.2. Relationship between aerosol enhancements and airmass history
Connections between the presence of aerosol enhancements and the outflow from con-
vective systems had been observed in some previous studies (de Reus et al., 2001; Twohy et al.,
2002; Benson et al., 2008; Weigelt et al., 2009). We examined the connection between deep con-
vection (DC) and the presence of high CN concentrations by a combination of backtrajectory cal-
culations and the analysis of cloud-top temperatures from GOES-13 weather satellite images,
similar to the approach used in some previous studies (de Reus et al., 2001; Froyd et al., 2009;
Weigelt et al., 2009). We analyzed backtrajectories initialized at the aircraft locations where we
had observed elevated aerosol concentrations, as listed in Table 1. Then we checked for each
hour along the backtrajectories whether the airmass had crossed a region with DC (cloud top
temperatures below -30 ºC). The results show that in almost all cases, the aerosol enriched air-
masses had encountered deep convection within the last 120 hours.
In Fig. 19 we present the results from two flights (AC09 and AC18) as examples. We
find that for all flight segments that showed high aerosol concentrations in the UT (dark shad-
ing), the airmasses had made contact with DC with cloud tops typically reaching about -80 ºC.
Of course, given the abundance of convection over Amazonia, it is to be expected that most air-
masses would have interacted with convection within 120 hours (such as the example shown in
the Supplement Fig. S2). For comparison, over the northeastern United States during summer-
time, Bertram et al. (2007) had found that more than 50% of UT air had encountered DC within
the previous 2 days.
The cumulative plot of the time since the most recent DC contact (Fig. 20a) shows that on
all flights (except AC19, the flight over the Atlantic) almost all aerosol-enhanced air masses had
seen DC within the last 30-40 hours. The cloud tops during these encounters typically reached -





70 to -80 ℃ (Fig. 20b). In many cases, the airmass history analysis shows multiple contacts with deep convection within the preceding 72 hours. It must be noted, however, that the physical interaction between an UT airmass and a specific deep convective event is not represented in the trajectory model and that the trajectory history preceding the most recent such encounter becomes much more uncertain.

In some cases, the airmasses could be tracked back to regions where the cold cloud encountered by the tracked airmass looked more like cirrus than identifiable deep convective outflow. The same favorable conditions for nucleation (low temperature, low pre-existing aerosol surface) as in the outflow regions prevail also in native cirrus, and Lee et al. (2004) had reported NPF in cirrus without immediate connection to DC. This might also have occurred in our campaign, but it is usually difficult to distinguish cirrus and very aged outflow.

More specific information about the time required for particle production and the evolution of the aerosol populations in the UT can be derived from a close examination of the trajectories for individual flight segments. Flight AC18 provides some illustrative examples. The trajectories of the first particle plumes encountered (A1 and A2, Table 1) had passed close to areas of intense deep convection (-30 to -60 ℃) about 17-21 hours before sampling. Because it is likely that the aerosol precursor substances are formed by photochemical reactions, we also looked at the amount of time that the airmass was exposed to sunlight (Lee et al., 2003). Since the convective encounters occurred between 16LT and 00LT and the measurements were taken at about 11LT, the airmass had only about 5-7 h of sun exposure. Assuming that the formation of the particles required photochemical processes, this implies that about 5-7 h were sufficient to produce particle concentrations above 20,000 cm$^{-3}$ with sizes >20 nm. The enrichment in this case occured only in the particles sizes <90 nm, with a UFF of about 0.98, while $N_{acc}$ remained at the same levels as in the surrounding background FT. Segment F, near the end of the flight, was sampling a similar region as A1, with a similar airmass trajectory. Since this segment was taken near the end of the day, the airmass had experienced about 11 hours of sunlight. There is somewhat of a shift towards larger particles, but this might also be coincidental.

The air in segments B and C had traveled along similar trajectories as A1 and A2, but unfortunately there are no GOES images available for the time when they crossed the convective





region encountered by A1 and A2, and so no conclusions can be drawn for these segments. Seg-
ments D and E1 represent airmasses that had made multiple and extended convection encounters
over the central and western Amazon during the past 3 days. Here, we find only weak enhance-
ments in $N_{CN}$, but significantly elevated $N_{CCN0.5}$ and $N_{acc}$, with a UFF of 0.73 and 0.82, respec-
tively, suggesting that coagulation and growth had taken place over this time period.
Some of the highest $N_{CN}$ (up to ca. 45,000 cm$^{-3}$) and UFF (0.98) were found in Segment
AC18-E2, which was sampling the air just a few hours downwind of a massive convective sys-
tem that reached well above our flight altitude of almost 14 km. The air sampled here had trav-
eled for about one hour after leaving the convective complex before being encountered by
HALO and had been interacting with this complex for up to 5 h, all of them in daylight. As in
A1, A2, and F, there was no detectable enhancement in aerosol mass, as represented by $N_{acc}$ and
$N_{CCN0.5}$. The strongest enhancement in aerosol mass, on the other hand, was seen in the early part
of segment E1, which didn't show a strong increase in number concentration. The air during this
segment had made its last contact with a convective system about 65-72 hours before sampling.
Another illustrative case is flight AC09 over a clean region in the northern Amazon. Seg-
ments A1-A3 sampled clear air that had DC contact about 16 and 60 hours ago and the UFF
around 0.94 indicated a moderately aged aerosol. Segments B1 and B2 were taken in air immedi-
ately surrounding a Cb anvil, with previous DC contacts at about 14, 80, and 120 hours before.
Here, the relatively low UFF of ~0.92 signaled no influence from the freshly outflowing air. Seg-
ments C, D, and E were in air close to a Cb, within its anvil, and in a large anvil/outflow, respec-
tively. Otherwise, they had a DC contact history similar to B. Here also, the UFF remains fairly
low, and there is no evidence of particle production directly in the anvil/outflow.
To summarize, our observations indicate that, while there is no evidence of immediate
production of detectable particles (i.e., >20 nm) in the actual anvil or outflow, a small number of
daylight hours are sufficient to produce very large concentrations of particles with sizes larger
than about 20 nm in the FT. This is consistent with the observations made in the outflow of a
convective complex off Darwin, Australia, where maximum Aitken concentrations were reported
after ca. 3 hours since the outflow (Waddicor et al., 2012). During NPF events in the FT on the
Jungfraujoch, high concentrations of particles >20 nm were observed about 4-6 hours after sun-
rise (Bianchi et al., 2016). In the FT over other regions, growth may be considerably slower; for



example the measurements over oceanic regions by Weigelt et al. (2009) showed that it took
about 12 hours for particles >12 nm to reach their maximum concentrations.

Considerably longer times (a few days) are required, however, before increases are de-

tectable in the size class >90 nm. The development of significant amounts of particles in the ac-
cumulation mode appears to take two days or more, in agreement with the observations of Froyd
et al. (2009), who had found enhanced aerosol organic mass concentrations over the Caribbean in
UT air originating from Amazonia after 2-4 days in the atmosphere. Since many, if not most of
our trajectories remain over Amazonia for this amount of time, there is enough time available in
the UT over the Amazon Basin to produce CCN-sized aerosols within the region, which can sub-
sequently be transported downward or be exported to other regions.
3.5.3. Aerosol enhancements and chemical tracers

The relationship between new particle production and the input of boundary layer air is

also reflected in a correlation between $N_{CN}$ and CO. When taking all data above 8 km, this corre-
lation is highly significant given the large number of data points (N=68,360) but not very close
($r^2$=0.52) because of the large variability of CO concentrations in the PBL and UT background
between flights (Fig. 21). Closer relationships are obtained when looking at individual flights
and especially at individual profiles within flights.

Weigel et al. (2011) had seen a strong correlation between CO and nucleation mode parti-

cles over West Africa and interpreted it as indication of anthropogenic inputs. In contrast, over
Amazonia we have not seen any evidence that UT aerosol production shows any relationship to
boundary layer pollution, and we interpret the correlation between $N_{CN}$ and CO simply as reflect-
ing the input of air from the PBL, which generally has higher CO concentrations that the UT, by
the cloud outflow. An opposite relationship is generally seen between $N_{CN}$ and $O_3$, which tends
to be lower in the particle-enriched layer. We also see this as an indication of injection of air
from the PBL, which generally has lower $O_3$ concentrations than the UT.

The nitrogen oxides show a complex relationship with the particle enhancements in the

UT, as illustrated at the example of a flight segment from AC07 (Fig. 22). The highest NO con-
centrations are found in the Cb anvils or freshest outflows, as identified by significant concentra-
tions of ice particles (e.g., at 2056, 2119, and 2154 UTC). In these regions, we typically observed
particle minima, as discussed above. In these airmasses, NO has been formed very recently by



lightning, and the NO to $NO_y$ ratios are usually still very high. Here, the particles are still de-
pleted by convection scavenging and there has not been enough time for new particles to form, at
least not in the size range detectable by our instrumentation. On the other hand, there is a strong
positive relationship between $NO_y$ and $N_{CN}$, as seen in Fig. 22 during the entire period from 2051
to 2210 UTC. Regions with high concentrations of new particles generally show elevated $NO_y$,
typically in the range of 1 to 3 ppb, indicating that photochemical reactions have taken place that
both produced new particles and converted NO to $NO_y$.

### 3.6. Flight AC20: A special case with NPF from biomass smoke

On flight AC20, HALO performed detailed sampling of the anvil and outflow of a large
Cb over northern Rondonia, a state with a high incidence of deforestation burning. Numerous
outflow penetrations around this Cb were made, and the ice particles sampled here could be
clearly identified as freshly produced in the Cb top. The CN concentrations in the UT away from
the outflow were unimpressive, typically in the range 2000 to 10,000 $cm^{-3}$. However, in sharp
contrast to the other flights, where the air in the outflow always had been depleted in aerosol par-
ticles, on this flight the outflow often showed much higher CN concentrations, between 10,000
and 20,000 $cm^{-3}$ (Fig. 23a). The concentrations of CCN and nonvolatile CN in the outflow were
either the same as in the surrounding air or slightly higher, also contrasting with the observations
on the other flights, where they had been depleted. Since the $N_{CN}$ in the outflow were also much
higher than in the PBL ($\sim$2000 $cm^{-3}$), entrainment of PBL air cannot explain the CN enrichment.
The mixing ratios of CO, NO, and $NO_y$ were also elevated in the outflow (Fig. 23b),
which in the case of CO and $NO_y$ might be explained by inputs from the PBL, where CO and
$NO_y$ levels were around 120-200 ppb and 2-3 ppb, respectively. The NO values in the PBL, on
the other hand, were only about 0.13 ppb, similar to the UT values, requiring an additional NO
source for the outflow.
The explanation for this unusual behavior may be found in the layer between 11.5 and
12.5 km that was penetrated during both ascent and descent (Fig. 23c). In this layer, $N_{CN}$ reached
30,000 $cm^{-3}$, CO was elevated to $\sim$140 ppb, $N_{acc}$ to 850 $cm^{-3}$, and $NO_y$ to $\sim$1.6 ppb. The data also
suggest a slight enrichment in rBC, but this is close to the limit of detection. These values sug-
gest that this is a detrainment layer polluted with biomass smoke, as we have often seen on previ-
ous campaigns over the burning regions in southern Amazonia (Andreae et al., 2004). An urban



origin of this pollution is unlikely, since the only town in the region, Porto Velho, lies about 50-100 km downwind of the sampling area. The enhancement ratios in this layer, however, differ from fresh biomass smoke. The ratio $\Delta N_{acc}/\Delta CO$ is ~6-12 cm$^{-3}$ ppb$^{-1}$ and the ratio $\Delta CCN/\Delta CO$ about 2.5 cm$^{-3}$ ppb$^{-1}$, much lower than the typical ratios in fresh smoke, which are about 20-40 cm$^{-3}$ ppb$^{-1}$ (Janhäll et al., 2010), indicating removal of CCN-sized particles during the upward transport. In contrast, the ratio $\Delta CN/\Delta CO$ was about 350 cm$^{-3}$ ppb$^{-1}$, almost an order of magnitude above the values typical of fresh smoke. These results suggest that biomass smoke was brought to the UT either from the strongly smoke-polluted PBL in this region or actually by a pyro-Cb over an active fire, and that the concentration of the larger particles was strongly reduced by scavenging, which allowed new particle formation in this smoke layer. The enrichments seen in the outflow penetrations at altitudes above the 12-km layer may be the result of entrainment of air from this layer or of rapid particle formation in situ. While we have this kind of observations from only one flight, which took place over the most polluted region sampled during this campaign, they are suggestive of the potential of rapid particle formation and growth in smoke detrainment layers, an issue that merits further study in future campaigns.

**3.7. Conceptual model and role in aerosol life cycle**

The discussion in the preceding sections can be summarized in a conceptual model of the aerosol life cycle over the Amazon Basin (Fig. 20). In the Amazon PBL, the classical nucleation events characterized by the rapid appearance of large numbers of particles <10 nm and subsequent growth into an Aitken mode (e.g., Kulmala and Kerminen, 2008) has never been reported, in spite of several years of observations by several teams (Martin et al., 2010; Rizzo et al., 2013; Andreae et al., 2015). This has been attributed to the low emissions of gaseous sulfur species in the basin (Andreae and Andreae, 1988; Andreae et al., 1990a), which result in $H_2SO_4$ vapor concentrations that are too low to induce nucleation (Martin et al., 2010). Nucleation of particles from organic vapors alone is not favored in the Amazonian PBL because of high temperatures and humidity as well as the competition by the condensation sink on pre-existing particles, which results in organic coatings on almost all primary and secondary particles in the Amazonian PBL (Pöschl et al., 2010; Pöhlker et al., 2012).

Cloud updrafts in deep convection bring air from the PBL into the middle and upper troposphere, where it is released in the convective outflow (Krejci et al., 2003). During this process,





1077 most pre-existing aerosols are removed by precipitation scavenging, especially the larger parti-

1078 cles that account for most of the condensation sink (Ekman et al., 2006). Most likely, VOCs with

1079 low and very low volatilities are also removed by deposition on hydrometeors, which provide a

1080 considerable amount of surface area inside the clouds (Murphy et al., 2015).

1081  The outflow regions in the UT present an ideal environment for particle nucleation, as

1082 had already been suggested in some earlier studies (Twohy et al., 2002; Lee et al., 2004; Kulmala

1083 et al., 2006; Weigelt et al., 2009). The temperatures are some 60-80 K lower than in the PBL,

1084 which decreases the equilibrium vapor pressure of gaseous species (Murphy et al., 2015) and in-

1085 creases the nucleation rate. Based on classical nucleation theory and molecular dynamics calcu-

1086 lations, Yu et al. (2017) have estimated an increase in nucleation rate by one order of magnitude

1087 per 10 K. Nucleation rate measurements in the CERN CLOUD chamber indicate a similar tem-

1088 perature dependence (Dunne et al., 2016). Because the preexisting aerosol has been depleted dur-

1089 ing the passage through convective clouds before being released into the UT from the cloud out-

1090 flow, the low particle surface area in the UT presents very little competition to nucleation from a

1091 condensation sink (Twohy et al., 2002; Lee et al., 2003; Lee et al., 2004; Young et al., 2007;

1092 Benson et al., 2008).

1093  The rapid transport of PBL air to the UT inside deep convective clouds facilitates lofting

1094 of reactive BVOCs from the Amazon boundary layer (Colomb et al., 2006; Apel et al., 2012).

1095 Here, the initially $O_3$- and $NO_x$-poor boundary layer air is supplied with $O_3$ by mixing with UT

1096 air and addition of NO from lightning, creating a highly reactive chemical environment. This

1097 mixture is exposed to an extremely high actinic flux due to the high altitude and multiple scatter-

1098 ing by ice particles. Because of the low airmass at UT altitudes, the actinic flux is already very

1099 high shortly after sunrise. In this environment, rapid photooxidation of BVOCs and formation of

1100 ELVOCs/HOMs is to be expected. In laboratory studies, HOMs have been shown to be rapidly

1101 produced at fairly high yields both by ozonolysis of terpenes and by reactions with OH radicals

1102 (Ehn et al., 2014; Jokinen et al., 2015; Berndt et al., 2016; Dunne et al., 2016).

1103  In the absence of measurements of the relevant gaseous sulfur species and the composi-

1104 tion of the nucleating clusters, we cannot make firm conclusions about the actual nucleation

1105 mechanism. Over marine regions and polluted continental regions, the particles observed in out-

1106 flows and in the UT were mostly identified as sulfates (Clarke et al., 1999; Twohy et al., 2002;





Kojima et al., 2004; Waddicor et al., 2012), and consequently $H_2SO_4$ has been proposed as the
nucleating species. However, since in some cases this identification was based only on the vola-
tility of the particles, they could have also consisted of organics or mixtures of organics and
$H_2SO_4$. Over the Amazon, nucleation by $H_2SO_4$ cannot be excluded based on our observations,
especially if there was already some $SO_2$ or $H_2SO_4$ present in the UT before the injection of the
organic-rich PBL air. However, since the Amazonian BL contains very little $SO_2$, the sulfur spe-
cies would have had to come from outside the region and thus they would have had the oppor-
tunity to be oxidized to $H_2SO_4$ and nucleate into particles during its travel in the UT well before
entering Amazonia. It is therefore much more likely that the particles in the Amazon UT formed
by homogeneous nucleation of organics, as has been suggested by several authors (Kulmala et
al., 2006; Ekman et al., 2008; Murphy et al., 2015). Nucleation by formation of clusters contain-
ing both $H_2SO_4$ and oxidized organic molecules is of course also a possibility that we cannot ex-
clude (Metzger et al., 2010; Riccobono et al., 2014). However, recent studies have shown that
HOM compounds can nucleate to form particles even in the absence of $H_2SO_4$, especially in the
UT (Bianchi et al., 2016; Kirkby et al., 2016), and nucleation of HOMs without involvement of
$H_2SO_4$ has been suggested to be the dominant mode of new particle formation in the pre-indus-
trial atmosphere by the modeling study of Gordon et al. (2016). The importance of ions produced
from cosmic radiation in this nucleation process is still controversial (Lee et al., 2003; Yu et al.,
2008; Bianchi et al., 2016; Kirkby et al., 2016).
Regardless of the actual nucleating species, $H_2SO_4$ or HOMs/ELVOCs, the growth of the
particles observed in our campaign must have been dominated by organics, as shown by the
composition of the aerosol measured by the AMS. The dominance of organics in the growth of
aerosols in pristine environments has also been suggested on the basis of modeling studies, both
for the lower troposphere (Laaksonen et al., 2008; Riipinen et al., 2012; Öström et al., 2017) and
the UT (Ekman et al., 2008; Murphy et al., 2015). In particular, isoprene-derived SOA has been
suggested to be important in the growth of sub-CCN-size particles to CCN (Ekman et al., 2008;
Jokinen et al., 2015), which would be consistent with the prevalence of isoprene in the Amazo-
nian PBL and our observations of IEPOX-SOA in the UT aerosol. As the particles grow, the de-
crease of the Kelvin (curvature) effect with increasing size of the growing particles implies that
subsequently relatively more volatile organics can condense (Tröstl et al., 2016), in agreement
with the observed high volatile fraction we observed in the upper tropospheric CN.



While in general the volatile fraction of the particles in the UT was very high, there were
also regions with a significant fraction of particles that did not evaporate at 250 ℃ (see section
3.4.3). These were dominated by relatively aged organics, which, based on the absence on de-
tectable rBC, must also be of secondary origin. Such thermally refractory organics may explain
the presence of non-volatile particles in the tropical UTLS, which had been observed in previous
campaigns especially in the region above 360 K (Borrmann et al., 2010).
Once particles have nucleated in the UT and grown into the Aitken mode and in some
cases even into the accumulation mode size ranges, they can be transported downward towards
the lower troposphere both by general subsidence under the prevailing high pressure system over
Amazonia and by downdrafts associated with deep convective activity. Large-scale entrainment
of UT and MT air into the boundary layer has been suggested as the major source of new parti-
cles in marine regions (Raes, 1995; Katoshevski et al., 1999; Clarke et al., 2013). Over Amazo-
nia with its high degree of convective activity, downdrafts are likely to play a more important
role. Downward transport of UT air by downdrafts associated with deep convective activity has
been shown to inject air with lower moisture content, lower equivalent potential temperature, and
elevated $O_3$ into the PBL (Zipser, 1977; Betts et al., 2002; Sahu and Lal, 2006; Grant et al., 2008;
Hu et al., 2010; Gerken et al., 2016). It would follow that the same mechanism also brings down
aerosol-rich air from the UT into the PBL. Indeed, in a recent aircraft study over the central Am-
azon, this mechanism was shown to be an important source of atmospheric aerosols, predomi-
nantly in the Aitken mode, to the Amazonian PBL (Wang et al., 2016a). Here, they can continue
to grow by condensation of BVOC-derived organics into the accumulation mode and become
available as CCN, closing the aerosol cycle over Amazonia.
**4. Summary and Conclusions**
As part of the ACRIDICON-CHUVA 2014 aircraft campaign, we investigated the char-
acteristics and sources of aerosols in the upper troposphere over the Amazon Basin. We observed
regions with high number concentrations of aerosol particles (tens of thousands per $cm^3$ STP) in
the UT on all flights that reached above 8 km. The aerosol enhancements were commonly in the
form of distinct layers with thicknesses of a few hundreds to a few thousands of meters. Such
layer structures are a common feature of the free troposphere and have been related to detrain-
ment from deep convection and large-scale subsidence (Newell et al., 1999).





In other regions, upward transport of aerosols from the PBL had been suggested to be an
important source of UT aerosols, based on the abundance of low-volatility particles (Clarke and
Kapustin, 2010), TEM analysis of individual particles (Kojima et al., 2004), or modeling of
cloud processes (Yin et al., 2005). Over Amazonia, however, the UT aerosol was fundamentally
different from the aerosol in the LT, indicating that upward transport of PBL aerosols, especially
combustion aerosols from BB, is not an important source of aerosols to the Amazonian UT.
The number concentrations of particles in the UT were often by several orders of magni-
tude higher than in the LT, and their size distribution was dominated by the Aitken rather than
the accumulation mode. In contrast to the LT, the particles in the UT were predominantly vola-
tile at 250 ºC and had much higher organics and nitrate contents. The extremely low concentra-
tions of rBC in the MT and UT show that the aerosols above the LT are not combustion-derived
and indicate that the low-volatility fraction must be representing secondary organics of extremely
low volatility (ELVOCs/HOMs). Regarding the size class large enough to act as CCN (larger
than 60-80 nm), we can conclude based on the absence of rBC and the lack of BB indicators in
the AMS measurements that the enhanced CCN in the UT are not related to upward transport of
combustion products, in contrast to most previous studies (e.g., Krejci et al., 2003; Engström et
al., 2008; Clarke et al., 2013).
By analyzing the history of the particle-enriched airmasses and comparing the transport
paths to GOES infrared imagery, we could show in almost all cases that these airmasses had
been in contact with deep convective outflow. Measurements inside the cloud tops and the out-
flow anvils close to the clouds showed that the pre-existing aerosols in the ascending air had
been almost completely scavenged by in-cloud processes, making the clouds initially a net aero-
sol sink. The near-complete scavenging is consistent with the hypothesized large water vapor su-
persaturation in pristine tropical deep convective clouds, which can nucleate particles that are
much smaller than the commonly defined CCN (Khain et al., 2012).
Based on our measurements, we propose that BVOCs in the cloud outflow are rapidly ox-
idized to HOMs/ELVOCs, which because of the low temperatures and low condensation sink
can readily nucleate new particles and grow to sizes ≥20 nm within a few hours, making deep
convective clouds an indirect aerosol source. This had also been concluded based on a large sta-
tistical sampling of UT air in the Northern Hemisphere by the CARIBIC aircraft measurement



program (Weigelt et al., 2009). The importance of NPF in the UT for the budget of CN and CCN
had been proposed previously on the basis of modeling studies (Yu et al., 2008; Merikanto et al.,
2009; Carslaw et al., 2017), and is evident in the global enhancement of CN in the UT, especially
in tropical regions, seen in compilations of data from numerous aircraft campaigns (Yu et al.,
2008; Reddington et al., 2016). In this way, aerosol production by BVOC oxidation in the UT
can provide the "missing source" of FT organic aerosol, which had been deduced from a mis-
match between models and observations (Heald et al., 2005).

The high aerosol concentrations in the UT provide a reservoir of particles that are availa-

ble for downward transport into the PBL both by large-scale downward motion and by convec-
tive downdrafts. In a recent study, we have shown that transport of aerosols by downdrafts from
the free troposphere is an important, if not the dominant, source of particles to the lower tropo-
sphere (LT) over the Amazon (Wang et al., 2016a). The particles that are produced by this mech-
anism in the UT over the Amazon (and probably other tropical continents as well) can be trans-
ported globally due to their long lifetime in the UT (Williams et al., 2002; Clarke et al., 2013)
and affect the microstructure of low-level clouds after they eventually descend into the PBL,
possibly at very large distances from the source areas of their precursors.

Our study and the results of some previous studies (Lee et al., 2003; Froyd et al., 2009)

suggest that UT aerosol production is especially important in the tropics because of the high rate
of BVOC production and the abundance of deep convection, but its relevance may also extend to
temperate and boreal regions. Our measurements both in the Amazon and at a remote site in cen-
tral Siberia, distant from $SO_2$ emission sources and thus experiencing very low $H_2SO_4$ concentra-
tions, show that classical nucleation events are very rare to absent at such sites and may not pro-
vide a strong source of new particles (Heintzenberg et al., 2011; Andreae et al., 2015;
Wiedensohler et al., 2017). Consequently, the UT may be an important, possibly even the domi-
nant source of tropospheric aerosol particles in regions that are not strongly affected by anthro-
pogenic aerosols. This would assign clouds a central role in the aerosol life cycle, controlling
both source and sink of aerosol particles, at least in regions of low anthropogenic pollution. Fur-
thermore, the relevance of UT aerosol production may not be limited to the troposphere, because
the UT and the TTL are also important reservoirs for the transport of particles into the lower
stratosphere (Fueglistaler et al., 2009; Borrmann et al., 2010; Randel and Jensen, 2013). Organic



1228 aerosols in the lower stratosphere have been shown to have significant radiative effects (Yu et

1229 al., 2016).

1230   The conceptual model proposed here implies a profound difference between the present-

1231 day polluted atmosphere and the pristine pre-industrial situation, especially over the continents.

1232 In the pristine atmosphere, the gradient of particle number concentrations may have been from

1233 high values in the UT to low values in the PBL, as we have found in Amazonia. In polluted con-

1234 tinental regions, on the other hand, nucleation and NPF occur predominantly in the lower tropo-

1235 sphere, which thus has become the dominant source of atmospheric aerosols in today's atmos-

1236 phere over much of the world. Consequently, in the anthropocene the aerosol concentration pro-

1237 file has been turned upside down in polluted regions, since now the highest concentrations are

1238 found in the PBL.

1239   This has important consequences for the Earth's climate system. The aerosol concentra-

1240 tions in the PBL influence cloud microphysical properties and radiative energy fluxes, which af-

1241 fect the characteristics of convection and thereby influence cloud radiative forcing, atmospheric

1242 stability, precipitation, and atmospheric dynamics at all scales (Jiang et al., 2008; Koren et al.,

1243 2008; Rosenfeld et al., 2008; Koren et al., 2010; Fan et al., 2012; Rosenfeld et al., 2014;

1244 Gonçalves et al., 2015; Stolz et al., 2015; Dagan et al., 2016; Braga et al., 2017). By their radia-

1245 tive and microphysical effects on convection dynamics, aerosols are also able to increase upper

1246 tropospheric humidity, which plays an important role in the Earth's radiation budget (Sherwood,

1247 2002; Kottayil and Satheesan, 2015; Riuttanen et al., 2016) and may also affect the potential for

1248 aerosol nucleation in the UT, thus providing an additional feedback.

1250 **5. Acknowledgments**

1251   We thank the entire ACRIDICON-CHUVA team for the great cooperation that made this

1252 study possible. Our thanks go especially to the HALO pilots, Steffen Gemsa, Michael Gross-

1253 rubatscher, and Stefan Grillenbeck, who always worked hard to put the aircraft at the right place

1254 for our measurements, even under sometimes difficult conditions. We appreciate the support of

1255 the colleagues from enviscope GmbH for their valuable help in certifying and installing the nu-

1256 merous instruments for HALO and thank the HALO team of the DLR for their cooperation. We

1257 acknowledge the generous support of the ACRIDICON-CHUVA campaign by the Max Planck



Society, the German Aerospace Center (DLR), FAPESP (São Paulo Research Foundation), and
the German Science Foundation (Deutsche Forschungsgemeinschaft, DFG) within the DFG Pri-
ority Program (SPP 1294) "Atmospheric and Earth System Research with the Research Aircraft
HALO (High Altitude and Long Range Research Aircraft)". This study was also supported by
EU Project HAIC under FP7-AAT-2012-3.5.1-1 and by the German Science Foundation within
DFG SPP 1294 HALO by contract no VO1504/4-1 and contract no JU 3059/1-1. C. Voigt
acknowledges financing by the Helmholtz Association under contract no. W2/W3-60. M. A.
Cecchini was funded by FAPESP grants number 2014/08615-7 and 2014/21189-7. The participa-
tion of D. Rosenfeld was supported by project BACCHUS, European Commission FP7-603445.
This study was also supported by the German Federal Ministry of Education and Research
(BMBF, grant No. 01LG1205E), and by the German Science Foundation within DFG SPP 1294
HALO by contract no VO1504/4-1, SCHN1138/1-2, and contract no JU 3059/1-1.





## 6. Figure Captions

Figure 1: Tracks of the flights on which measurements at high altitude were made during ACRIDICON-CHUVA. The flight segments at altitudes >8 km are shown as heavier lines.

Figure 2: Columnar precipitable water anomaly for September 2014 (based on the 1981-2010 average NCEP/NCAR Reanalysis).

Figure 3: Total rainfall (mm per month, 1° resolution) for September 2014. Data from the Global Precipitation Climatology Centre (GPCC).

Figure 4: Mean wind speeds during September 2014 at a) 850 hPa and b) 200 hPa (Data from NCEP/NCAR).

Figure 5: Vertical profiles of potential temperature, static air temperature, and relative humidity measured on HALO during the ACRIDICON-CHUVA flights over the Amazon Basin.

Figure 6: Trajectory statistics based on (a) 72-hour and (b) 120-hour backtrajectory calculations for September 2014, initialized at Manaus at an elevation of 12 km.

Figure 7: Vertical profiles of CN concentrations, $N_{CN}$; a) overall statistics from all flights, b) examples from individual profiles on flight AC07 (segment G) and AC09 (segments A1 and A2).

Figure 8: Vertical profiles of accumulation mode particle concentrations, $N_{acc}$; a) 1-min averaged data from all flights, b) $N_{acc}$ profile from flight AC19 together with the profile of $N_{CN}$ from the same flight (1-sec data).

Figure 9: Size spectra: The black line shows the mean boundary layer DMPS size spectrum from a segment in the PBL on flight AC13 (16:55 to 17:18 UTC). The square black symbols represent the mean, the grey shaded area the standard deviation of the measurements. The line is a logarithmic fit with modal diameters of 74 and 175 nm. The colored lines represent size distributions from 0.65 to 5.8 km from a G1 flight during GoAmazon (Wang et al., 2016a).

Figure 10: Vertical profiles of the ultrafine fraction (UFF); a) overall statistics from all flights, b) examples from individual profiles on flight AC18.

Figure 11: Vertical profiles of CCN concentrations at 0.52% supersaturation; a) overall statistics from all flights (1-min averages), b) examples from individual profiles on flights AC09 (green)





and AC12+13 (red). Flights AC12 and AC13 were conducted over the same region on successive
days.
Figure 12: a) CCN fraction ($N_{CCN0.5}/N_{CN}$) vs altitude, all data. b) CCN fraction vs. CN concentra-
tion for specific segments from flight AC18 (see text for discussion).
Figure 13: a) CCN fractions ($N_{CCN0.5}/N_{CN}$) and b) CCN concentrations ($N_{CCN0.5}$) vs. supersatura-
tion from selected legs from flights AC09, AC10, and AC18; c,d) data from flights AC12 and
AC13 for the LT, MT, and UT.
Figure 14: Volatile fraction. a) statistics from all flights; b) individual segments from flight
AC18 (see text for discussion).
Figure 15: Refractory black carbon vs altitude, all flights, 30-second averages.
Figure 16: Aerosol chemical composition as determined by AMS and SP2 measurements in the
lower, middle and upper troposphere.
Figure 17: Plot of the AMS factors $f_{44}$ vs. $f_{43}$, indicating the median values for the LT and UT
and values for some UT flight segments with elevated aerosol concentrations. With increasing
degree of oxidation, the measurements move to the upper left of the triangle
Figure 18: Measurements during passages through cumulonimbus cloud tops and outflow anvils:
a) Several cloud top penetrations at 10.7 to 12 km altitude on flight AC18 showing reduced $N_{CN}$
and $N_{CCN0.5}$ inside the cloud top; b) Outflow from a large active cumulonimbus, showing strong
aerosol depletion and NO production by lightning.
Fig. 19: Airmass contacts with deep convection. The colors indicate the cloud top temperature of
the convective system with which the trajectory had the most recent contact. The aircraft altitude
at which the airmass was sampled is indicated by the red line. The colored dots are plotted at the
altitude at which the airmass crossed the grid cell with the convective system. The dots are only
plotted if this altitude is greater than 6 km and if it encountered a DC region (i.e., $T_b < -30$ °C).
The shaded areas correspond to the flight segments with elevated CN concentrations. a) flight
AC09, b) flight AC18.
Figure 20: a) Number of hours since last contact with deep convection for flight segments with
elevated aerosol concentrations (cumulative frequency, all flights); b) frequency distribution of



minimum GOES brightness temperature ($T_b$) for selected flights legs (within 5-day backward tra-
jectories).
Figure 21: CN vs CO concentrations in the upper troposphere above 8 km (15-second averages).
Figure 22: CN, NO, and $NO_y$ concentrations in a flight segment in the upper troposphere on
flight AC07.
Figure 23: a) Measurements of $N_{CCN0.5}$, $N_{CN}$, $N_{nonvol}$, and ice particles during cloud top penetra-
tions on flight AC20. b) Concentrations of CO, NO, and $NO_y$ on the same flight segments. c)
Measurements of $N_{acc}$, $N_{CN}$, rBC, CO, and $O_3$ during the climb from 11.0 to 13.5 km.
Figure 24: Conceptual model of the aerosol life cycle over the Amazon Basin.





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



Table 1: Properties of the flight legs on which elevated aerosol concentrations were measured during ACRIDICON-CHUVA.

| Flight | Leg | Start UTC | End UTC | Altitude range (m) | $N_{CN}$ max. (cm$^{-3}$) | $N_{CN}$ mean (cm$^{-3}$) | $N_{CCN0.5}$ mean (cm$^{-3}$) | $N_{acc}$ mean (cm$^{-3}$) | Ultrafine fraction | Trajectory type | Min $T_b$ [min,max][a] (°C) | Time since last DC [min,max][b] (hours) | Time in DC [min,max][c] (hours) | Sampling environment |
|---|---|---|---|---|---|---|---|---|---|---|---|---|---|---|
| AC07 | A1 | 1622 | 1626 | 8300-9200 | 17200 | 9360 | 657 | 696 | 0.93 | A | [-76,-65] | [0,0] | [21,27] | in and near outflows |
| AC07 | AA1 | 1626 | 1627 | 9140 | 36100 | 19230 | 775 | 588 | 0.97 | A | - - - | - - - | | in and near outflows |
| AC07 | A2 | 1627 | 1633 | 8100-9100 | 38400 | 24250 | 471 | 499 | 0.98 | A | [-77,-76] | [0,0] | [19,26] | clear air |
| AC07 | AA2 | 1633 | 1637 | 6700-8200 | 26700 | 6450 | 708 | 565 | 0.91 | A | - - - | - - - | | clear air |
| AC07 | B | 1714 | 1717 | 7000-8400 | 15900 | 7140 | 214 | 270 | 0.96 | A | [-75,-68] | [0,0] | [13,28] | clear air |
| AC07 | C | 1923 | 1929 | 9000 | 22600 | 16480 | 272 | 389 | 0.98 | A | [-78,-74] | [0,0] | [27,40] | clear air |
| AC07 | D1 | 2024 | 2027 | 8500-10500 | 23200 | 14270 | - - - | 146 | 0.99 | A | [-74,-68] | [0,0] | [29,40] | clear air near outflow |
| AC07 | D2 | 2028 | 2112 | 11000 | 28200 | 15160 | - - - | 76 | 0.99 | A | [-76,-68] | [0,0] | [12,28] | outflow, mixed with cirrus |
| AC07 | E | 2126 | 2129 | 13100 | 33500 | 15140 | - - - | - - - | - - - | A | [-72,-67] | [0,0] | [21,28] | pristine ice cirrus |
| AC07 | F | 2130 | 2147 | 13200 | 25300 | 12030 | 13 | - - - | - - - | A | [-72,-69] | [0,5] | [24,32] | clear air |
| AC07 | G | 2205 | 2211 | 13000-10000 | 20500 | 15470 | 284 | - - - | - - - | A | [-76,-51] | [0,0] | [24,31] | cirrus |
| AC07 | GG | 2210 | 2212 | 10200-9500 | 19500 | 16840 | - | 869 | 0.95 | A | | | | cirrus |
| AC08 | | | | No useful high alt CN data. CCN moderately elevated at ca. 10 and 13 km, ca. 1200 /cc | | | | | | | | | | |
| AC09 | A1 | 1453 | 1455 | 11400 | 24100 | 10370 | 901 | 572 | 0.94 | B | [-74,-71] | [16,16] | [22,41] | clear air |
| AC09 | A2 | 1455 | 1458 | 11900 | 27600 | 12970 | 1103 | 808 | 0.94 | B | [-76,-72] | [16,17] | [34,41] | clear air |
| AC09 | A3 | 1501 | 1503 | 11000 | 35100 | 14470 | 629 | 697 | 0.95 | B | [-72,-70] | [17,17] | [38,40] | clear air |
| AC09 | B1 | 1815 | 1820 | 11000 | 19100 | 10540 | 1393 | 954 | 0.91 | B | [-76,-74] | [14,14] | [49,54] | around Cb anvil |
| AC09 | B2 | 1821 | 1827 | 11300-11600 | 28300 | 15370 | 1414 | 1012 | 0.93 | B | [-78,-73] | [14,14] | [47,57] | around Cb anvil |
| AC09 | C | 1830 | 1838 | 11600 | 31700 | 9130 | 1490 | 1127 | 0.88 | B | [-79,-76] | [1,19] | [45,56] | clear air |
| AC09 | D | 1838 | 1923 | 11300-11900 | 13000 | 5690 | 1012 | 869 | 0.85 | B, C | [-80,-74] | [1,1] | [34,57] | outflow region |
| AC09 | E | 1929 | 1957 | 11300 | 24200 | 12790 | 891 | 856 | 0.93 | B, C | [-76,-70] | [2,21] | [24,48] | outflow region |
| AC10 | A | 1709 | 1714 | 6700-8600 | 27400 | 13040 | 355 | 389 | 0.94 | C | [-66,-54] | [6,7] | [9,32] | clear air |
| AC10 | B | 1721 | 1728 | 9200 | 32500 | 12480 | 850 | 861 | 0.91 | D | [-78,-72] | [4,10] | [34,56] | clear air |
| AC10 | C | 1800 | 1808 | 9200 | 26000 | 13100 | 1020 | 937 | 0.91 | B | [-79,-71] | [7,10] | [33,56] | clear air |
| AC10 | D | 1811 | 1815 | 9200-10100 | 33000 | 20180 | 1130 | 684 | 0.95 | B | [-77,-71] | [5,5] | [23,51] | clear air |
| AC10 | E | 1817 | 1833 | 10800-13600 | 33400 | 22210 | 712 | 289 | 0.98 | E | [-84,-72] | [0,12] | [42,76] | thin cirrus |
| AC10 | F | 1835 | 1906 | 13800 | 34700 | 16540 | 464 | - - - | - - - | E | [-80,-68] | [0,0] | [33,54] | cirrus layer |
| AC10 | G | 1912 | 1919 | 10600-7500 | 24200 | 10220 | 1230 | 1160 | 0.83 | B | [-80,-58] | [0,14] | [11,60] | clear air |



| | | | | | | | | | | | | | | |
|---|---|---|---|---|---|---|---|---|---|---|---|---|---|---|
| AC11 | A | 1603 | 1605 | 8700-9700 | 47400 | 26280 | 572 | 323 | 0.98 | E | [-54,-32] | [3,44] | [1,18] | clear air |
| AC11 | B | 1613 | 1630 | 11800 | 4700 | 3850 | 1390 | 763 | 0.80 | E, D | [-76,-58] | [0,6] | [14,41] | clear air |
| AC11 | C | 1633 | 1642 | 11800-10800 | 31700 | 6080 | 1436 | 937 | 0.78 | D | [-80,-77] | [0,0] | [30,46] | around anvil |
| AC11 | D | 1831 | 1850 | 5200-6700 | 25000 | 14380 | --- | 187 | 0.98 | C | [-79,-79] | [0,0] | [18,19] | outflow region |
| AC11 | E | 1907 | 1930 | 9900-12200 | 36100 | 29280 | --- | 330 | 0.99 | D | [-85,-74] | [0,0] | [26,82] | outflow region |
| AC11 | F1 | 1940 | 1942 | 12200 | 54900 | 22060 | --- | 674 | 0.95 | E, D | [-84,-84] | [0,0] | [55,55] | outflow region |
| AC11 | F2 | 1942 | 1951 | 12200 | 32800 | 20720 | --- | 549 | 0.97 | E, D | [-84,-84] | [0,0] | [55,55] | outflow region |
| AC11 | G | 2005 | 2030 | 13700-14200 | 2830 | 10090 | --- | --- | --- | D | [-84,-84] | [0,0] | [55,55] | outflow region |
| AC11 | H | 2042 | 2057 | 12200-10400 | 47900 | 20240 | --- | 663 | 0.96 | A | [-84,-84] | [0,0] | [55,55] | outflow region |
| AC12 | A | 1512 | 1518 | 9800-11300 | 19300 | 8040 | 1130 | 341 | 0.95 | E | [-79,-74] | [0,0] | [23,37] | clear |
| AC12 | B | 1524 | 1527 | 11300 | 24700 | 9290 | 1120 | 358 | 0.95 | A | [-83,-71] | [0,0] | [26,66] | thin outflow |
| AC12 | C | 1537 | 1541 | 7300-5600 | 26200 | 7760 | 356 | 186 | 0.95 | B | [-78,-57] | [1,1] | [7,16] | clear |
| AC12 | D | 1922 | 1925 | 8000-9700 | 17400 | 11980 | 650 | 132 | 0.99 | B | [-71,-71] | [17,20] | [6,12] | clear |
| AC12 | E | 1928 | 1933 | 10800-12200 | 25300 | 15740 | 423 | 75 | 0.99 | B | [-70,-57] | [20,24] | [8,18] | clear |
| AC12 | F1 | 1936 | 1950 | 12200-13100 | 7020 | 5940 | 2010 | 698 | 0.88 | D | [-80,-67] | [0,38] | [12,40] | clear |
| AC12 | F2 | 1952 | 2015 | 13100 | 7300 | 5950 | 1190 | 594 | 0.90 | B, D | [-82,-74] | [0,21] | [28,77] | aged outflow |
| AC12 | G | 2017 | 2020 | 13200-12800 | 19600 | 10930 | 661 | 422 | 0.96 | E | [-79,-75] | [0,0] | [26,49] | outflow |
| AC12 | H | 2023 | 2027 | 11300-9600 | 23900 | 16930 | 849 | 372 | 0.98 | C | [-80,-77] | [0,0] | [37,59] | mostly clear air |
| AC13 | A | 1520 | 1533 | 11000-11900 | 43500 | 13830 | 1054 | --- | --- | C | [-78,-75] | [0,12] | [27,43] | mostly cirrus and old outflow |
| AC13 | B | 1550 | 1607 | 11900-6900 | 36300 | 11890 | 1012 | 476 | 0.95 | A | [-83,-50] | [1,8] | [11,47] | mostly cirrus and old outflow |
| AC13 | C | 1901 | 1908 | 9500 | 25700 | 17870 | 687 | --- | --- | A | [-72,-66] | [0,0] | [13,24] | clear air around anvils |
| AC13 | D1 | 1909 | 1912 | 10700 | 26200 | 18600 | 910 | --- | --- | A | [-70,-66] | [0,0] | [15,19] | "" |
| AC13 | D2 | 1916 | 1919 | 10700 | 28200 | 19170 | 1017 | --- | --- | A | [-73,-69] | [0,0] | [24,25] | "" |
| AC13 | D3 | 1921 | 1926 | 10700 | 29500 | 19010 | 919 | --- | --- | A | [-69,-68] | [0,0] | [15,26] | "" |
| AC13 | D4 | 1930 | 1933 | 10700 | 21600 | 10890 | 727 | --- | --- | A | [-68,-67] | [0,0] | [14,17] | "" |
| AC13 | E | 1939 | 1942 | 11900 | 22500 | 15100 | 770 | --- | --- | A | [-57,-47] | [10,10] | [5,8] | "" |
| AC13 | F | 2036 | 2043 | 12200 | 18600 | 7840 | 912 | --- | --- | A | [-78,-76] | [0,0] | [34,43] | clear air, some cirrus |
| AC14 | no useable high alt data | | | | | | | | | | | | | |
| AC15 | A | 1415 | 1419 | 10500-11700 | 58500 | 38170 | 687 | 453 | 0.98 | D | [-81,-78] | [0,9] | [63,68] | air around a huge Cb anvil |
| AC15 | B | 1419 | 1424 | 11800-12900 | 67900 | 46970 | 701 | 405 | 0.98 | D | [-81,-81] | [0,0] | [59,66] | mostly cirrus and old outflow |
| AC15 | C | 1431 | 1432 | 13200 | 49500 | 20900 | 1070 | 747 | 0.94 | D | [-84,-84] | [0,0] | [55,55] | "" |
| AC15 | D | 1436 | 1437 | 13200 | 38300 | 15300 | 1009 | 633 | 0.92 | D | [-84,-77] | [0,0] | [50,56] | "" |



| Flight | | | | | | | | | | | | | | |
|---|---|---|---|---|---|---|---|---|---|---|---|---|---|---|
| AC15 | E | 1448 | 1449 | 12500 | 44500 | 29220 | 603 | 718 | 0.97 | D | [-81,-79] | [0,0] | [54,59] | "" |
| AC15 | F | 1452 | 1455 | 12500 | 60500 | 45100 | 672 | 514 | 0.97 | D | [-79,-75] | [0,0] | [52,56] | "" |
| AC15 | G | 1456 | 1500 | 12500-11900 | 59200 | 38070 | 748 | 574 | 0.98 | D | [-82,-72] | [0,0] | [53,62] | "" |
| AC15 | H | 1502 | 1505 | 11900-11600 | 49800 | 16440 | 1114 | 750 | 0.94 | D | [-76,-73] | [0,0] | [62,69] | "" |
| AC15 | I | 1518 | 1519 | 11300 | 46800 | 22000 | 1848 | 931 | 0.93 | D | [-79,-73] | [0,0] | [65,71] | "" |
| AC15 | J | 1526 | 1528 | 10700 | 21700 | 8980 | 1292 | 817 | 0.86 | D | [-76,-75] | [0,0] | [59,65] | "" |
| AC16 | A | 1554 | 1600 | 10700-12200 | 40300 | 21210 | 606 | 223 | 0.98 | B | [-75,-68] | [0,0] | [9,18] | clear air |
| AC16 | B | 1749 | 1757 | 10000-10300 | 28200 | 11350 | 926 | 282 | 0.97 | B | [-68,-57] | [0,0] | [8,10] | air around a large Cb anvil |
| AC16 | C | 1803 | 1815 | 10300-10700 | 27200 | 15180 | 746 | 208 | 0.98 | B | [-75,-60] | [0,0] | [9,12] | air around a large Cb anvil |
| AC16 | D | 1818 | 1820 | 10700-11300 | 23100 | 11540 | 789 | 356 | 0.97 | B | [-75,-67] | [0,0] | [10,17] | air around a large Cb anvil |
| AC16 | E | 1824 | 1826 | 12000 | 26700 | 14070 | 488 | 354 | 0.97 | B | [-75,-75] | [0,0] | [17,19] | air around a large Cb anvil |
| AC16 | F | 1857 | 1911 | 12600-11900 | 19500 | 11210 | 598 | 521 | 0.94 | B | [-73,-66] | [0,0] | [22,28] | air around a large Cb anvil |
| AC16 | G | 1925 | 1935 | 11900 | 22700 | 12880 | 703 | 492 | 0.95 | B | [-73,-70] | [0,0] | [22,30] | air around a large Cb anvil |
| AC16 | H | 1950 | 2000 | 11900-9600 | 27100 | 12670 | 806 | 444 | 0.96 | B | [-75,-65] | [0,0] | [13,29] | air around a large Cb anvil |
| AC17 | no high alt data | | | | | | | | | | | | | |
| AC18 | A1 | 1454 | 1456 | 8300-8600 | 20700 | 10698 | - | 219 | 0.98 | B | [-60,-10] | [14,17] | [2, 5] | clear air |
| AC18 | A2 | 1520 | 1522 | 12900-8400 | 22500 | 14538 | 479 | 400 | 0.97 | C | [-58,-38] | [14,18] | [1, 5] | clear air |
| AC18 | B | 1753 | 1801 | 7100 | 10040 | 6255 | 400 | 312 | 0.95 | C | [-30, -0] | [0,0] | [1, 2] | clear air around anvils |
| AC18 | C | 1833 | 1834 | 7100-7400 | 14200 | 10713 | 404 | 280 | 0.97 | C | [-52,-28] | [22,22] | [1, 1] | clear air around anvils |
| AC18 | D | 1913 | 2005 | 11300-12000 | 4000 | 2367 | 916 | 640 | 0.73 | A, D | [-75,-37] | [0,16] | [3,46] | clear air around anvils |
| AC18 | E1 | 2017 | 2034 | 13000-13700 | 8170 | 4841 | 1481 | 892 | 0.82 | A,D | [-84,-68] | [0,44] | [21,45] | clear air |
| AC18 | E2 | 2040 | 2043 | 13700-13200 | 44700 | 13679 | 469 | 283 | 0.98 | D | [-77,-71] | [0,0] | [28,42] | clear air downwind of large Cb |
| AC18 | F | 2053 | 2057 | 9500-8100 | 15800 | 8778 | 444 | 318 | 0.96 | C, D | [-68,-32] | [1,20] | [1,11] | clear air |
| AC19 | A1 | 1518 | 1519 | 7300-7700 | 30600 | 28480 | 451 | 339 | 0.99 | B | [-82,-65] | [14,43] | [7,14] | clear air |
| AC19 | A2 | 1536 | 1601 | 12600 | 3600 | 2910 | 679 | 268 | 0.91 | E | [-72,-58] | [43,94] | [6,19] | clear air, high alt leg |
| AC19 | E1 | 2009 | 2010 | 8500-8900 | 14700 | 11470 | 642 | 271 | 0.98 | B | [-75,-59] | [16,92] | [8,16] | clear air |
| AC19 | E2 | 2023 | 2100 | 13800 | 3900 | 2690 | 1024 | 498 | 0.81 | A | [-76,-29] | [0,105] | [1,22] | clear air |
| AC19 | E3 | 2106 | 2119 | 13800 | 10200 | 2770 | 1073 | 950 | 0.65 | B | [-73,-57] | [0,1] | [6,25] | outflow |
| AC19 | E4 | 2127 | 2128 | 7500-6600 | 66000 | 16210 | 440 | 414 | 0.96 | D | [-60,-59] | [3,22] | [4,7] | clear air |
| AC20 | A | 1654 | 1658 | 11700-12500 | 30300 | 21540 | 881 | 616 | 0.97 | A, D | [-77,-53] | [1,1] | [7,28] | NPF at top of smoke layer |
| AC20 | B | 1901 | 1905 | 12300 | 21300 | 9340 | 614 | 381 | 0.95 | A, D | [-78,-70] | [0,0] | [14,42] | NPF at top of smoke layer |





Atmospheric Chemistry and Physics Discussions — Open Access — EGU

[a] Minimum and maximum temperature at top of most recent deep convection in grid boxes through which the trajectories for the flight leg had passed.

[a] Trajectories were calculated for each minute of the leg, and for each trajectory the time between sampling and the most recent encounter with DC was determined. Given are the shortest and the longest of these time intervals.

[a] Minimum and maximum length of time that the trajectories from each leg had spent in grid boxes with DC.



Table 2: Composition of UT aerosols based on AMS and SP2 measurements

| Flight | Time | $N_{CN}$ | $N_{CCN0.5}$ | $N_{acc}$ | OA | NO3 | SO4 | NH4 | rBC | OA/SO4 | NO3/SO4 | Ultrafine fraction | CO |
|---|---|---|---|---|---|---|---|---|---|---|---|---|---|
| | UT | cm$^{-3}$ | cm$^{-3}$ | cm$^{-3}$ | µg m$^{-3}$ | µg m$^{-3}$ | µg m$^{-3}$ | µg m$^{-3}$ | µg m$^{-3}$ | | | | ppb |
| **AC07** | | | | | | | | | | | | | |
| <4 km | - | 1620±680 | 1070±410 | 1363±651 | 1.15±0.82 | 0.057±0.031 | 0.14±0.07 | 0.21±0.16 | 0.40±0.21 | 8.1±5.8 | 0.40±0.29 | 0.19±0.16 | - |
| >7 km | - | 9300±7420 | 300±210 | 278±232 | 0.43±0.36 | 0.052±0.036 | 0.038±0.032 | 0.07±0.47 | 0.003±0.007 | 11.3±13.5 | 1.4±1.5 | 0.92±0.008 | - |
| AA1 | 16:24-16:29 | 19200 | 650 | 588 | 1.03 | 0.097 | <0.005 | - | 0.002 | >200 | >20 | 0.97 | - |
| AA2 | 16:33-16:37 | 6450 | 710 | 565 | 0.90 | 0.086 | 0.011 | - | 0.002 | 82 | 7.8 | 0.89 | - |
| GG | 22:09-22:11 | 16800 | - | 921 | 1.72 | 0.143 | <0.005 | - | 0.002 | >350 | >30 | - | - |
| **AC09** | | | | | | | | | | | | | |
| <5 km | - | 920±490 | 290±95 | 395±189 | 0.42±0.29 | 0.020±0.027 | 0.26±0.12 | 0.02±0.13 | 0.085±0.095 | 2.2±1.8 | 0.14±0.13 | 0.51±0.26 | - |
| >9 km | - | 8020±5180 | 1090±430 | 861±338 | 2.53±0.60 | 0.31±0.17 | 0.24±0.19 | 0.02±0.17 | 0.001±0.003 | 13.4±6.3 | 1.6±1.1 | 0.86±0.07 | - |
| AA | 14:48-15:08 | 2280 | 1050 | 754 | 2.23 | 0.30 | 0.14 | 0.013 | 0.001 | 24.0 | 3.0 | 0.54 | - |
| BB | 18:18-19:23 | 8060 | 1200 | 922 | 2.63 | 0.32 | 0.27 | 0.023 | 0.001 | 10.7 | 1.3 | 0.85 | - |
| EE | 19:28-19:58 | 12000 | 950 | 892 | 2.75 | 0.31 | 0.23 | 0.018 | 0.001 | 12.8 | 1.4 | 0.92 | - |
| A1±A2 | 14:53-14:58 | 12100 | 1040 | 724 | 2.50 | 0.36 | 0.15 | 0.039 | <0.001 | 23.9 | 3.4 | 0.91 | - |
| **AC18** | | | | | | | | | | | | | |
| <5 km | - | 740±220 | 350±100 | 473±212 | 1.61±1.26 | 0.070±0.054 | 0.92±0.47 | 0.17±0.16 | 0.15±0.15 | 1.6±0.8 | 0.078±0.055 | 0.51±0.26 | - |
| >10 km | - | 2950±2640 | 920±310 | 560±145 | 2.66±0.98 | 0.32±0.15 | 0.40±0.12 | <0.05 | 0.002±0.005 | 7.0±3.0 | 0.85±0.38 | 0.86±0.07 | - |
| AA | 15:06-15:16 | (1740) | 870 | 545 | 2.20 | 0.28 | 0.35 | <0.05 | 0.001 | 6.9 | 0.89 | 1.50 | - |
| DD | 19:21-20:05 | 2360 | 910 | 639 | 2.75 | 0.31 | 0.39 | <0.05 | 0.002 | 7.8 | 0.88 | 0.61 | - |
| A1 | 14:54-14:56 | 87000 | - | 203 | 0.52 | 0.099 | 0.27 | <0.05 | 0.002 | 2.5 | 0.44 | - | - |
| A2 | 15:20-15:22 | 17400 | 500 | 433 | 1.36 | 0.157 | 0.27 | <0.05 | 0.002 | 5.1 | 0.62 | 0.97 | - |
| E2 | 20:40-20:43 | 15900 | 360 | - | 1.28 | 0.157 | 0.35 | <0.05 | - | 3.9 | 0.44 | 0.98 | - |
| F | 20:54-20:56 | 11600 | 460 | 361 | 1.37 | 0.164 | 0.42 | <0.05 | 0.002 | 3.3 | 0.39 | 0.96 | - |
| UT | 9-15 km | 7700±8000 | 840±440 | 568±313 | 2.57±1.12 | 0.273±0.165 | 0.32±0.23 | 0.21±0.22 | 0.003±0.003 | 8.1±6.7 | 0.86±0.78 | 0.86±0.11 | 116±39 |
| MT | 5-8 km | 2130±3100 | 410±150 | 284±169 | 1.07±0.80 | 0.075±0.103 | 0.35±0.22 | 0.22±0.16 | 0.007±0.015 | 3.0±3.0 | 0.21±0.32 | 0.79±0.15 | 97±22 |
| PBL | 0-4 km | 1650±980 | 950±700 | 1261±876 | 4.71±3.65 | 0.189±0.212 | 0.82±0.61 | 0.43±0.59 | 0.39±0.26 | 5.8±6.2 | 0.23±0.31 | 0.28±0.23 | 157±54 |




Figure 1: Tracks of the flights on which measurements at high altitude were made during ACRIDICON-CHUVA. The flight segments at altitudes >8 km are shown as heavier lines.

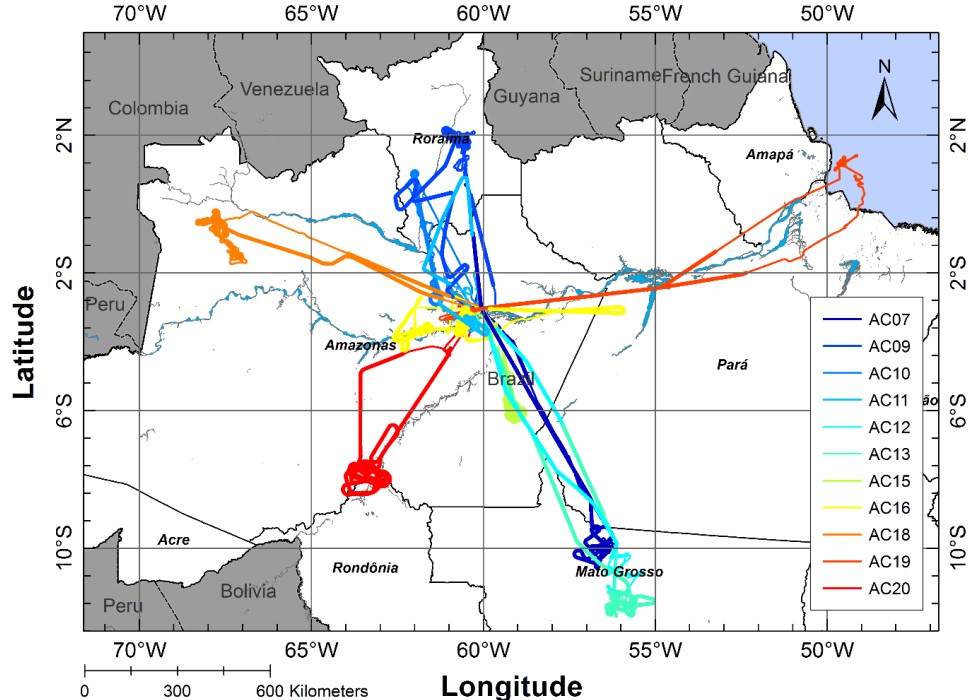



Figure 2: Columnar precipitable water anomaly for September 2014 (based on the 1981-2010 average NCEP/NCAR Reanalysis).

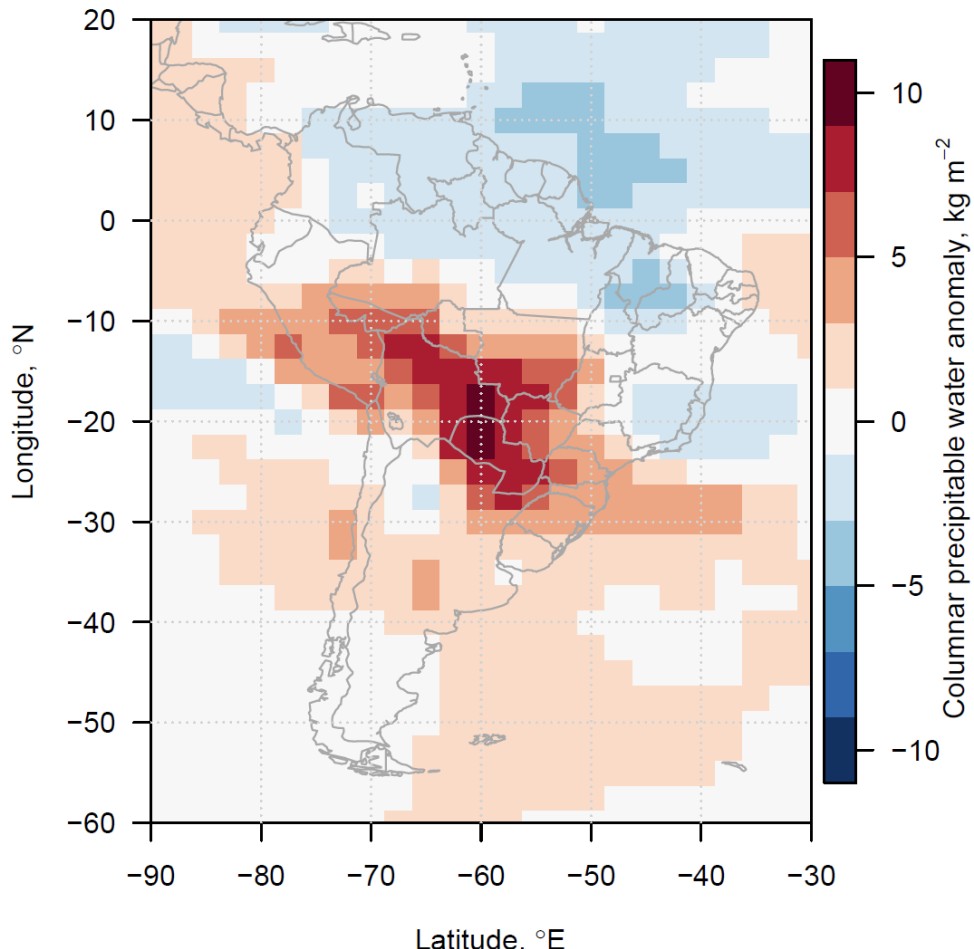



Figure 3: Total rainfall (mm per month, 1° resolution) for September 2014. Data from Global Precipitation Climatology Centre (GPCC).

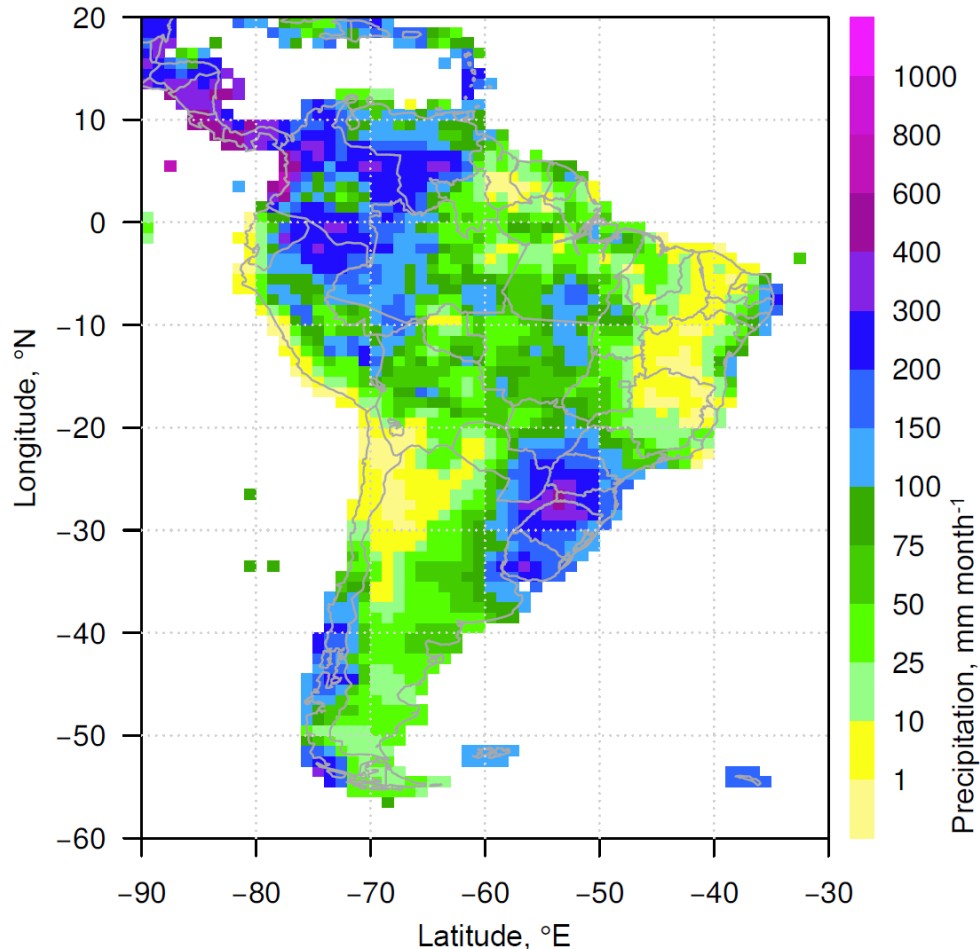




Figure 4: Mean wind speeds during September 2014 at a) 850 hPa and b) 200 hPa (Data from NCEP/NCAR).

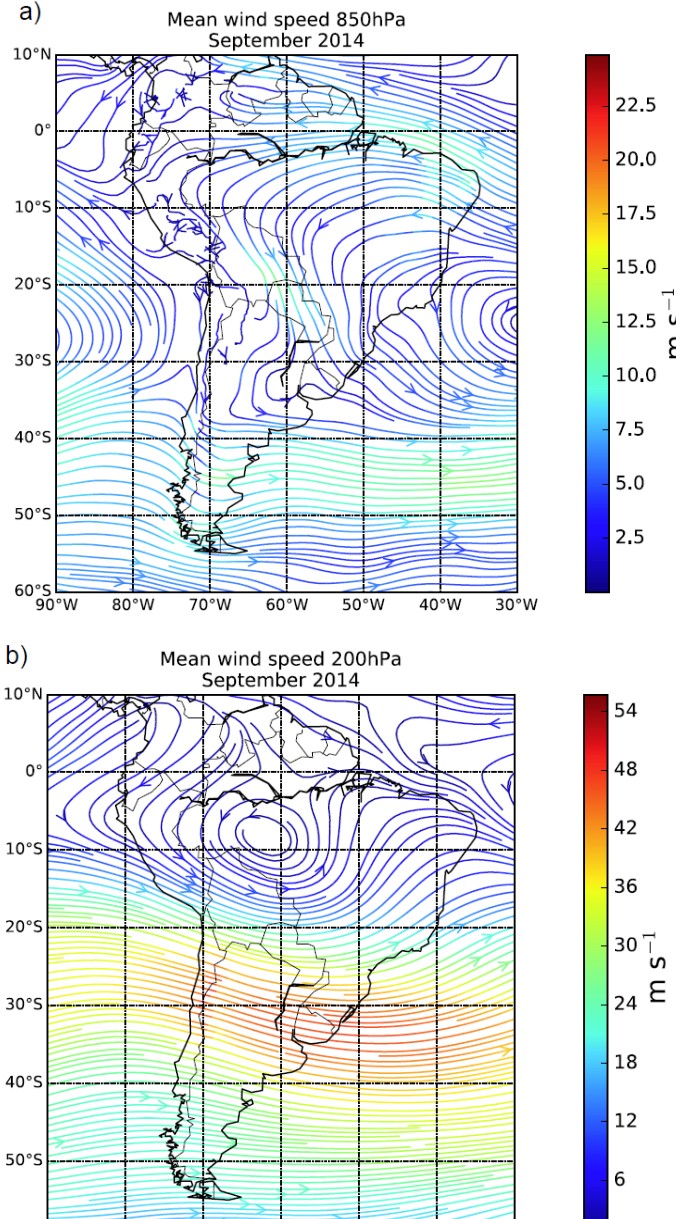



Figure 5: Vertical profiles of potential temperature, static air temperature and relative humidity measured on HALO during the ACRIDICON-CHUVA flights over the Amazon Basin.

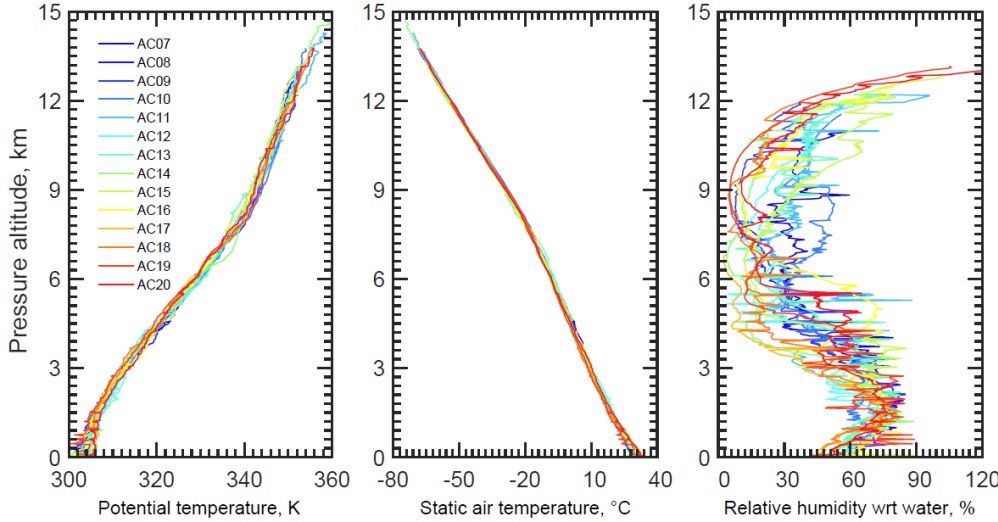



Figure 6: Trajectory statistics based on (a) 72-hour and (b) 120-hour backtrajectory calculations for September 2014, initialized at Manaus at an elevation of 12 km.



Figure 7: Vertical profiles of CN concentrations, $N_{CN}$; a) overall statistics from all flights, b) examples from individual profiles on flight AC07 (segment G) and AC09 (segments A1 and A2).

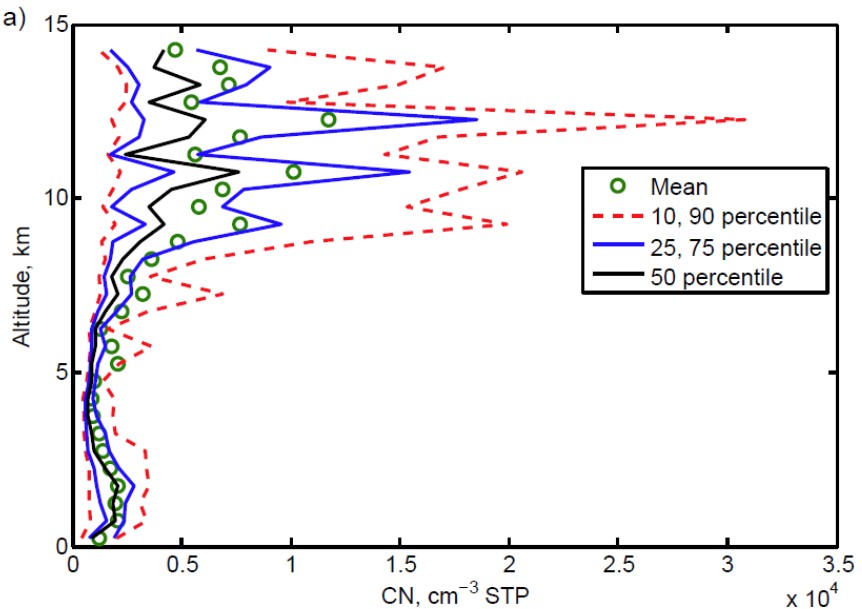

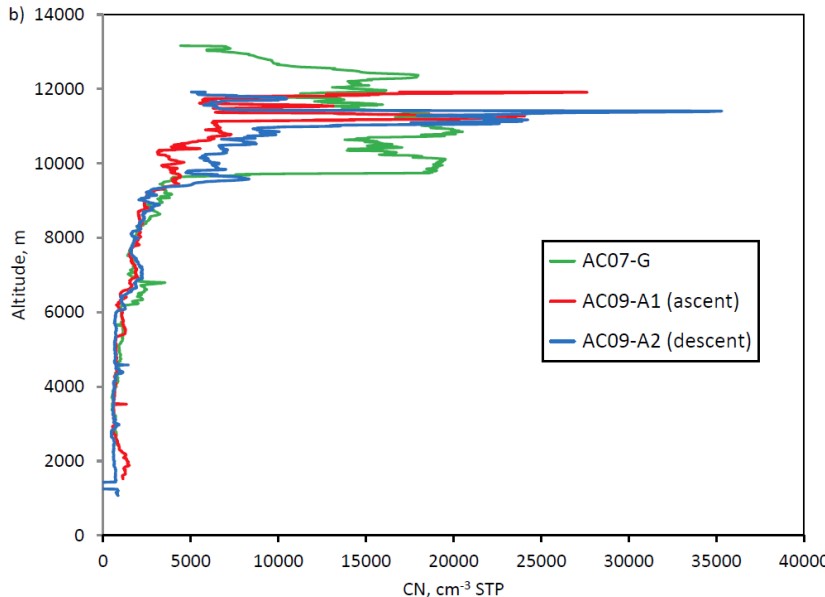





Figure 8: Vertical profiles of accumulation mode particle concentrations, $N_{acc}$; a) 1-min averaged data from all flights, b) $N_{acc}$ profile from flight AC19 together with the profile of $N_{CN}$ from the same flight (1-sec data).

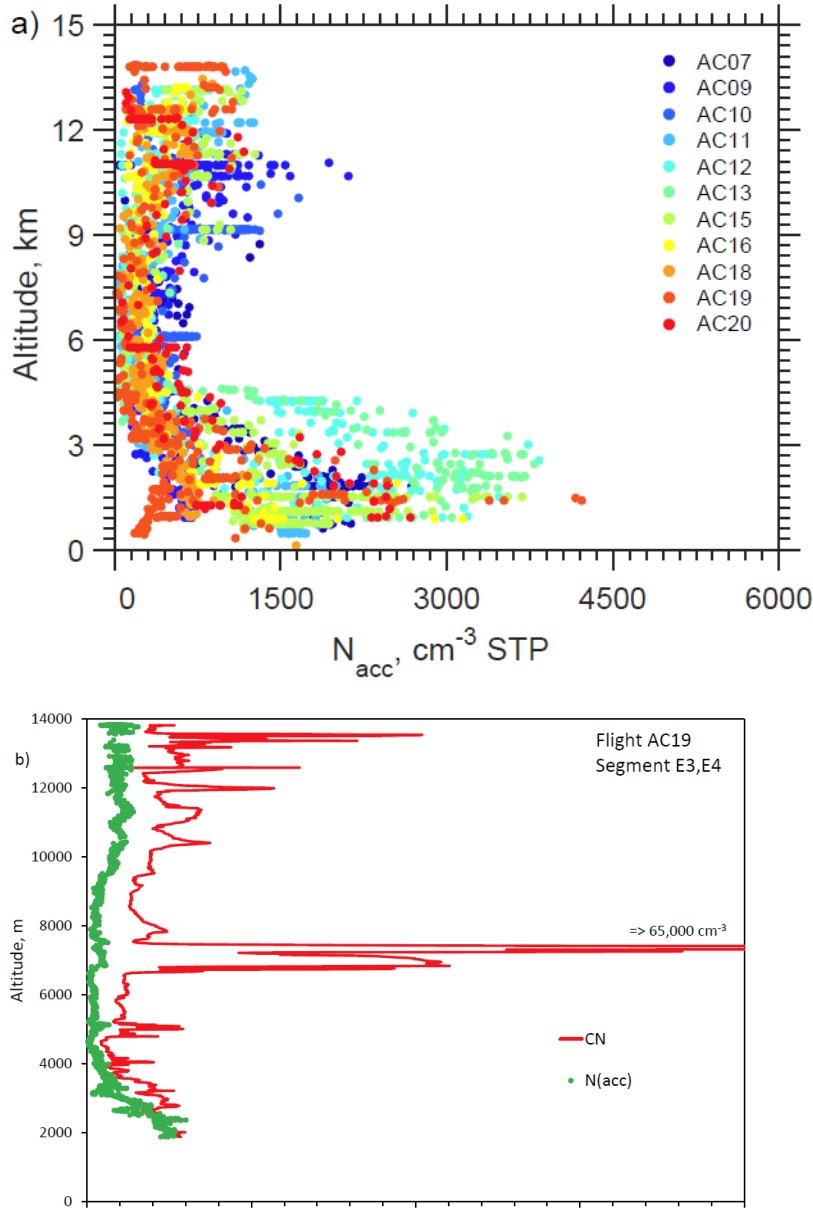




Figure 9: Size spectra: The black line shows the mean boundary layer DMPS size spectrum from a segment in the PBL on flight AC13 (16:55 to 17:18UT). The square black symbols represent the mean, the grey shaded area the standard deviation of the measurements. The line is a logarithmic fit with modal diameters of 74 and 175 nm. The colored lines represent size distributions from 0.65 to 5.8 km from a G1 flight during GoAmazon (Wang et al., 2016a).

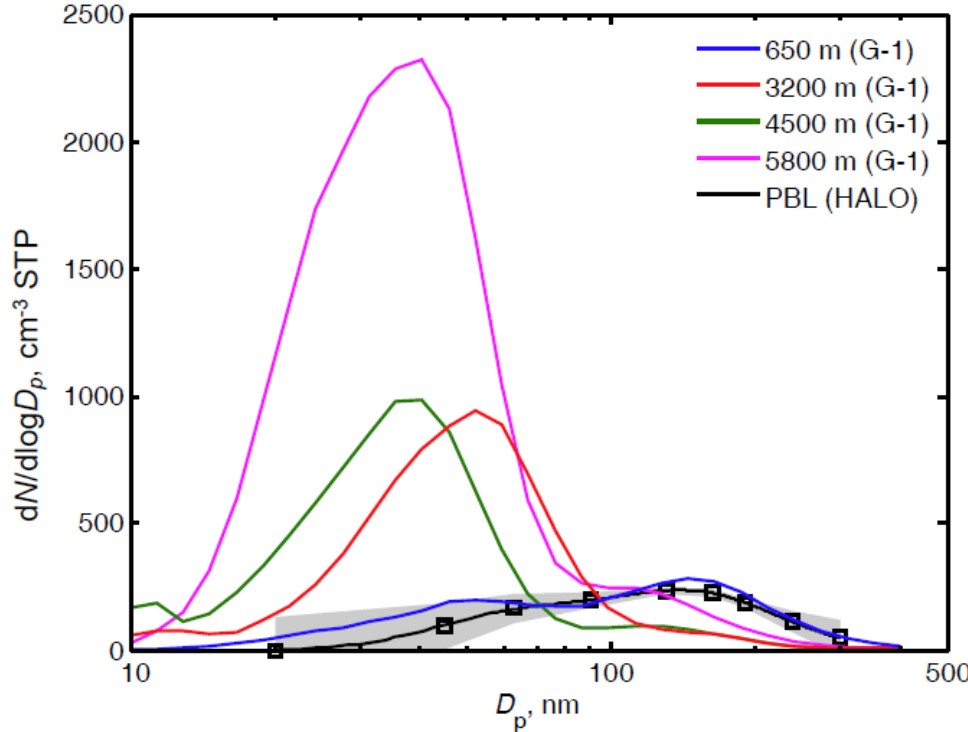


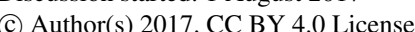


Figure 10: Vertical profiles of the ultrafine fraction (UFF); a) overall statistics from all flights, b) examples from individual profiles on flight AC18.

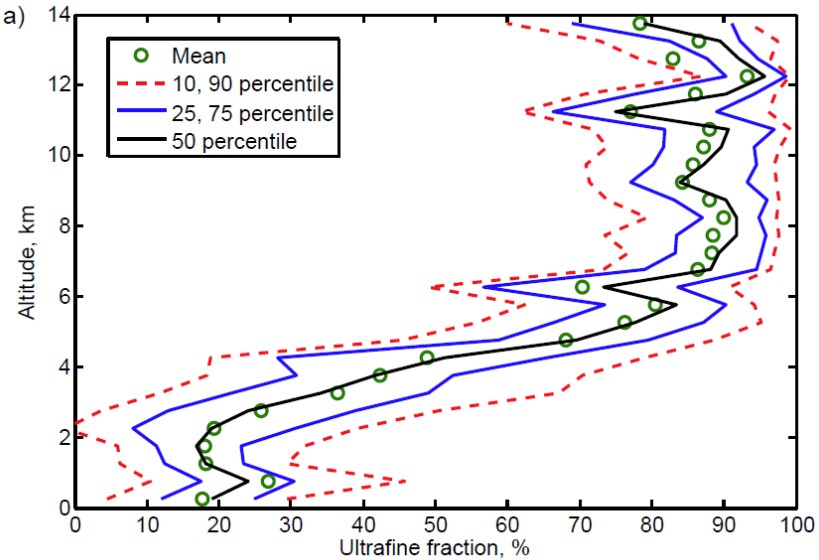

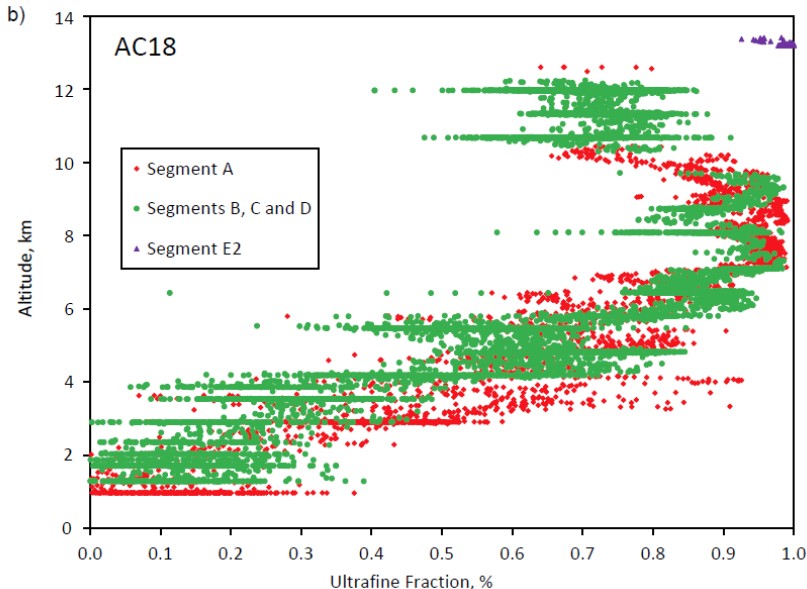


Figure 11: Vertical profiles of CCN concentrations at 0.52% supersaturation; a) overall
statistics from all flights (1-min averages), b) examples from individual profiles on flights
AC09 (green) and AC12+13 (red). Flights AC12 and AC13 were conducted over the same
region on successive days.

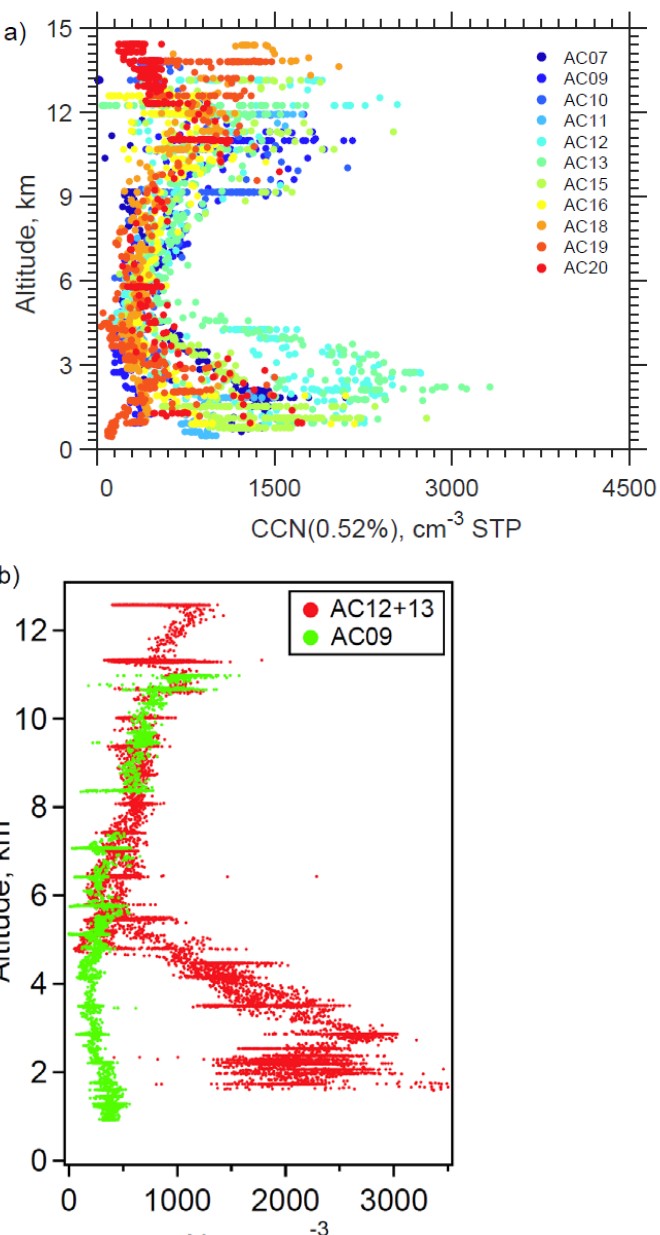



Figure 12: a) CCN fraction vs altitude, all data. b) CCN fraction vs. CN concentration for specific segments from flight AC18 (see text).

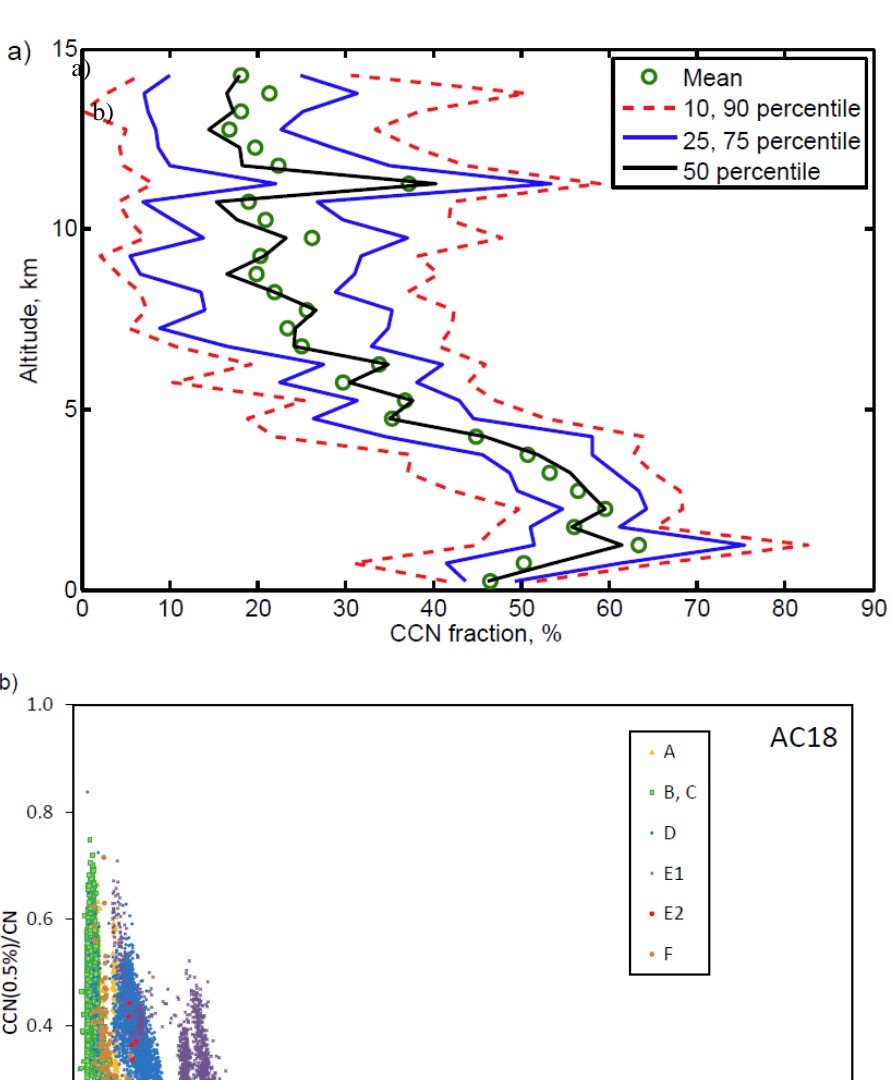



Figure 13: a) CCN fractions ($N_{CCN0.5}/N_{CN}$) and b) CCN concentrations ($N_{CCN0.5}$) vs. supersaturation from selected legs from flights AC09, AC10, and AC18; c,d) data from flights AC12 and AC13 for the LT, MT, and UT.





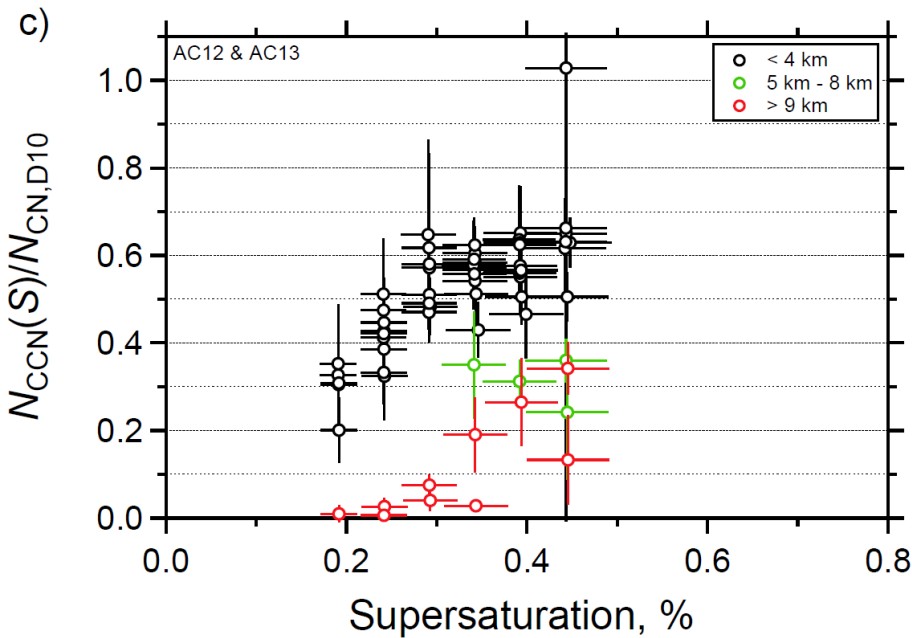

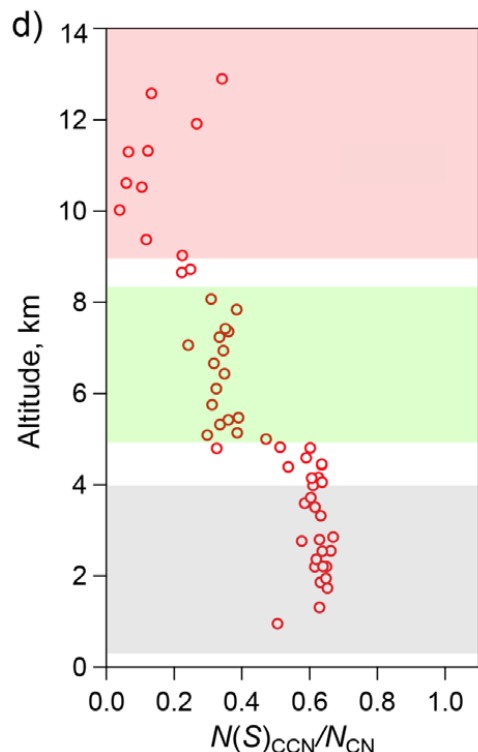



Figure 14: Volatile fraction. a) statistics from all flights; b) individual segments from flight AC18 (see text)

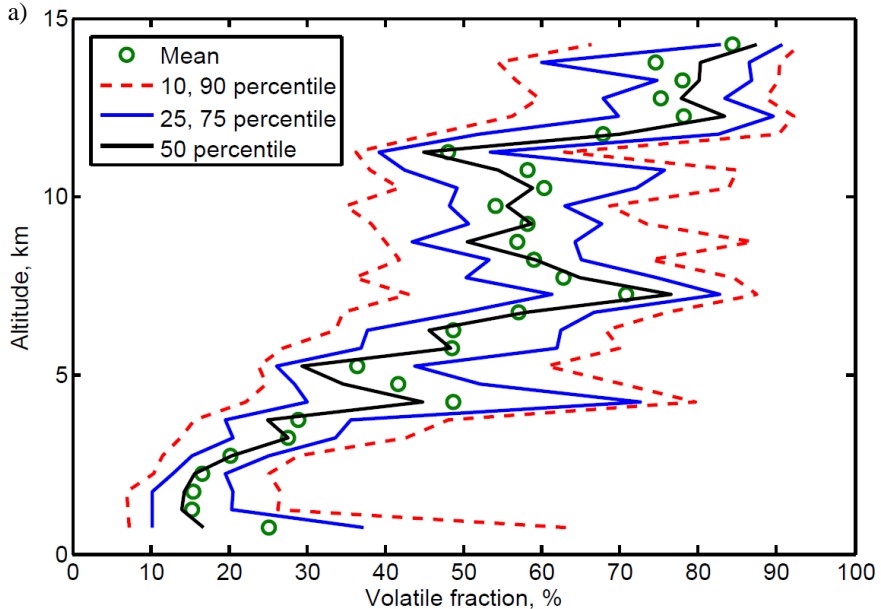

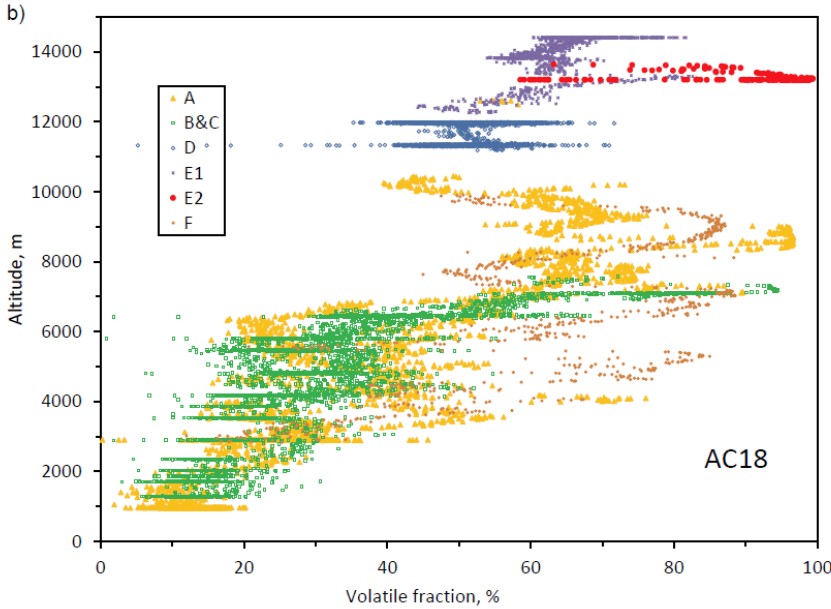

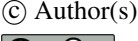



Figure 15: Refractory black carbon vs altitude, all flights, 30-second averages.

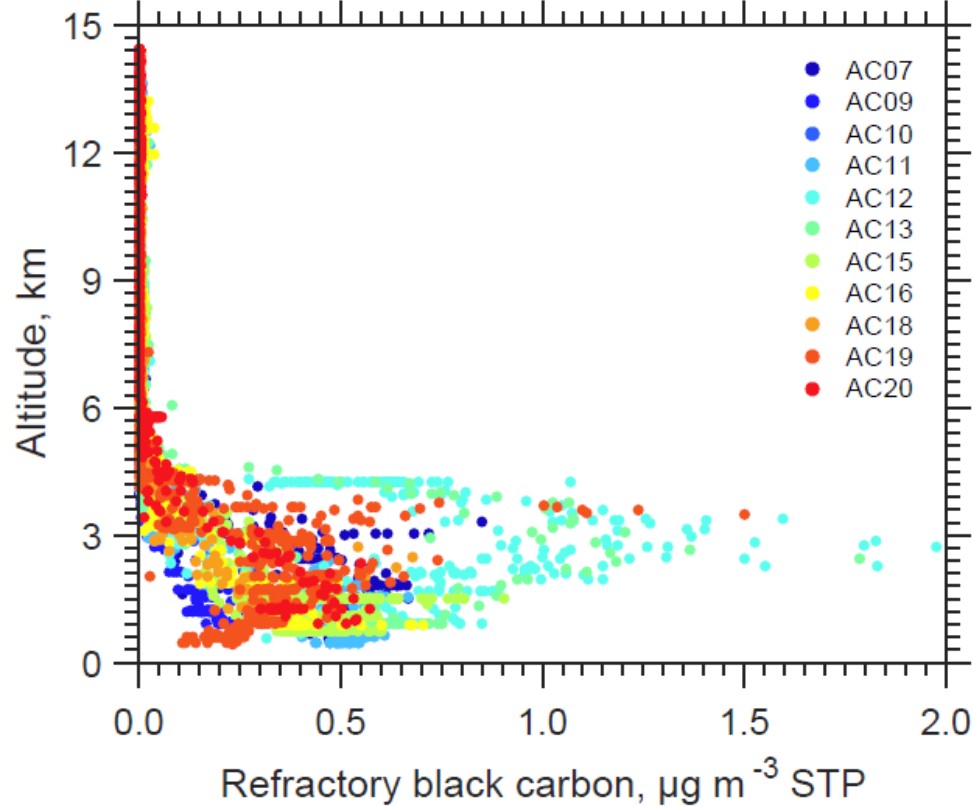




Figure 16: Aerosol chemical composition as determined by AMS and SP2 measurements in the lower, middle and upper troposphere.

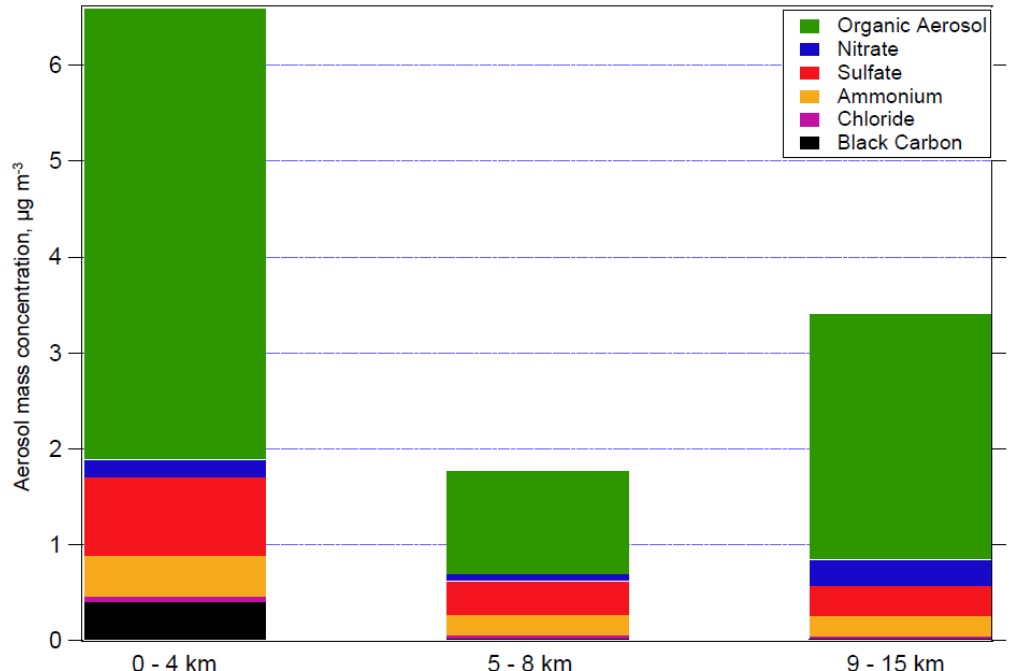




Figure 17: Plot of the AMS factors $f_{44}$ vs. $f_{43}$, indicating the median values for the LT and UT and values for some UT flight segments with elevated aerosol concentrations. With increasing degree of oxidation, the measurements move to the upper left of the triangle

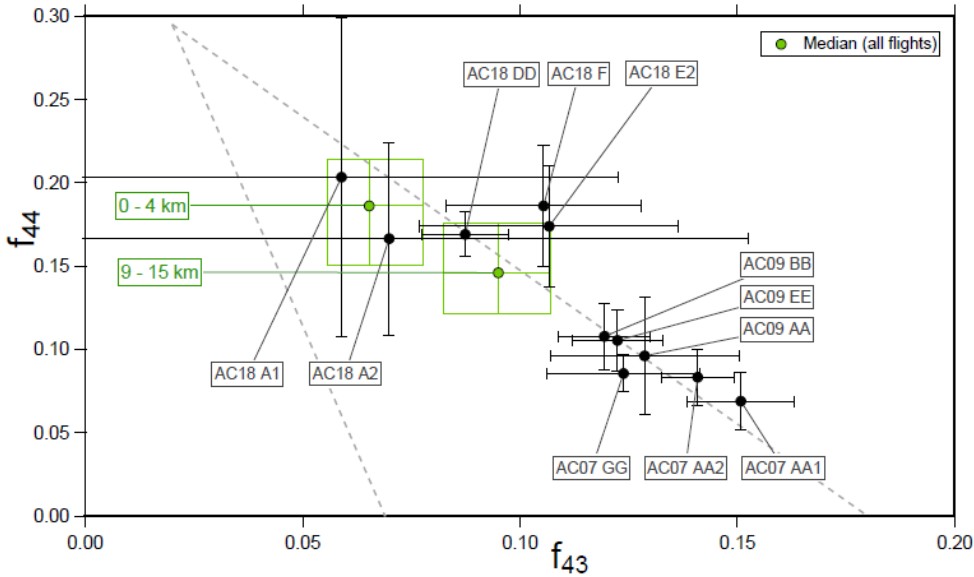



Figure 18: Measurements during passages through cumulonimbus cloud tops and outflow anvils: a) Several cloud top penetrations at 10.7 to 12 km altitude on flight AC18 showing reduced $N_{CN}$ and $N_{CCN0.5}$ inside the cloud top; b) Outflow from a large active cumulonimbus, showing strong aerosol depletion and NO production by lightning.

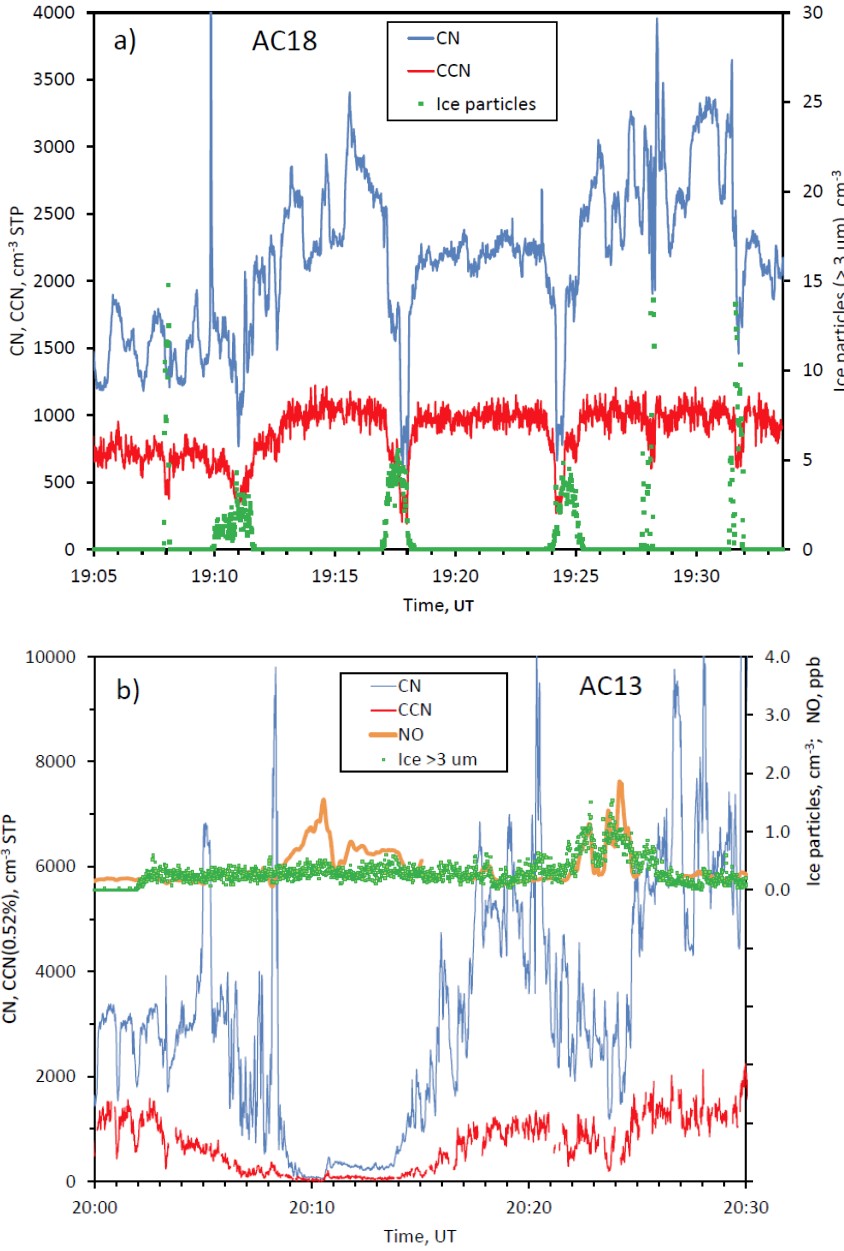

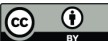


Fig. 19: Airmass contacts with deep convection. The colors indicate the cloud top temperature of the convective system with which the trajectory had the most recent contact. The aircraft altitude at which the airmass was sampled is indicated by the red line. The colored dots are plotted at the altitude at which the airmass crossed the grid cell with the convective system. The dots are only plotted if this altitude is greater than 6 km and if it encountered a DC (i.e., $T_b < -30$ ºC). The shaded areas correspond to the flight segments with elevated CN concentrations. a) flight AC09, b) flight AC18.

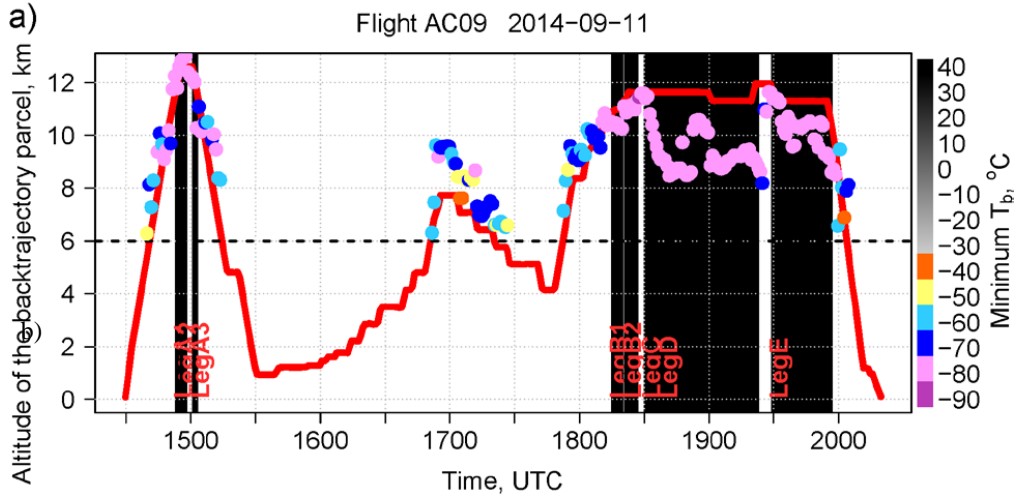

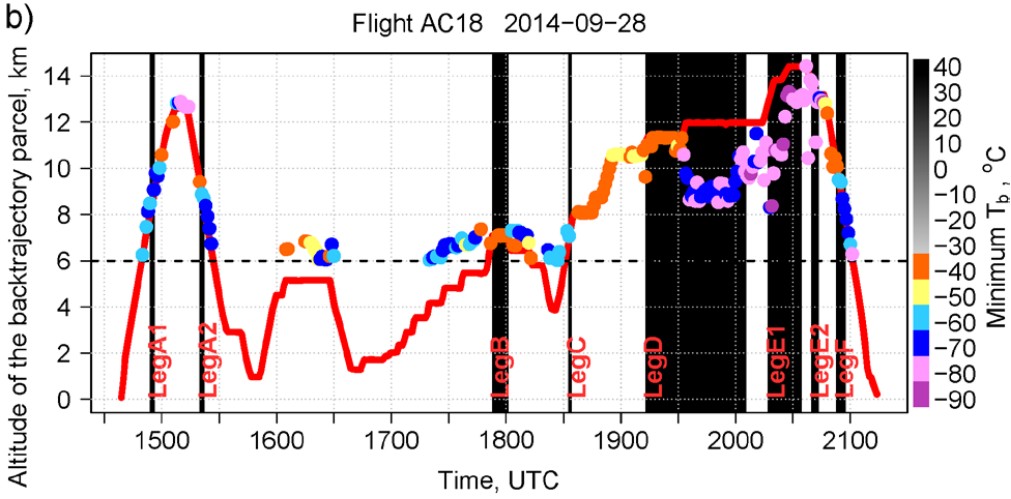



Figure 20: a) Number of hours since last contact with deep convection for flight segments with elevated aerosol concentrations (cumulative frequency, all flights); b) frequency distribution of minimum GOES brightness temperature ($T_b$) for selected flights legs (within −5 days backward trajectories).

a)

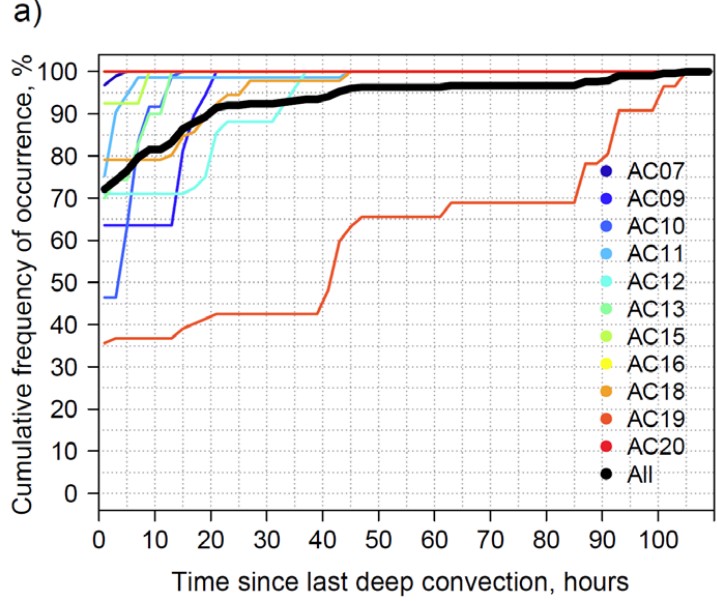

b)

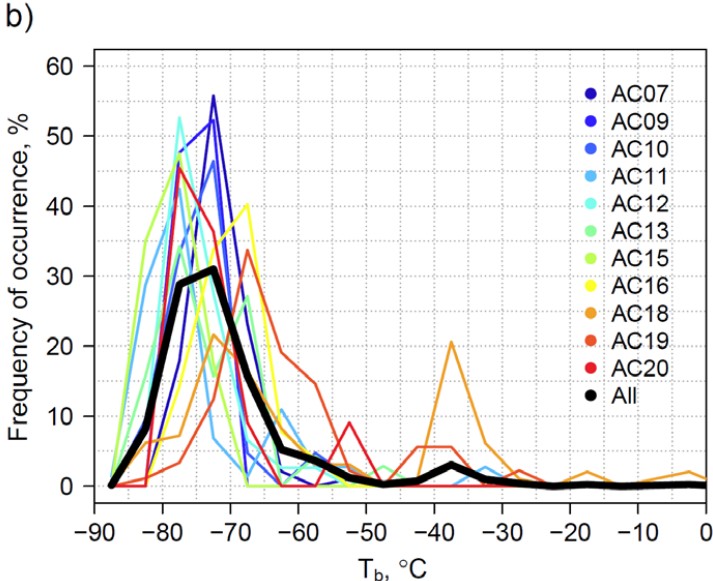





Figure 21: CN vs CO in the upper troposphere above 8 km (15-second averages).

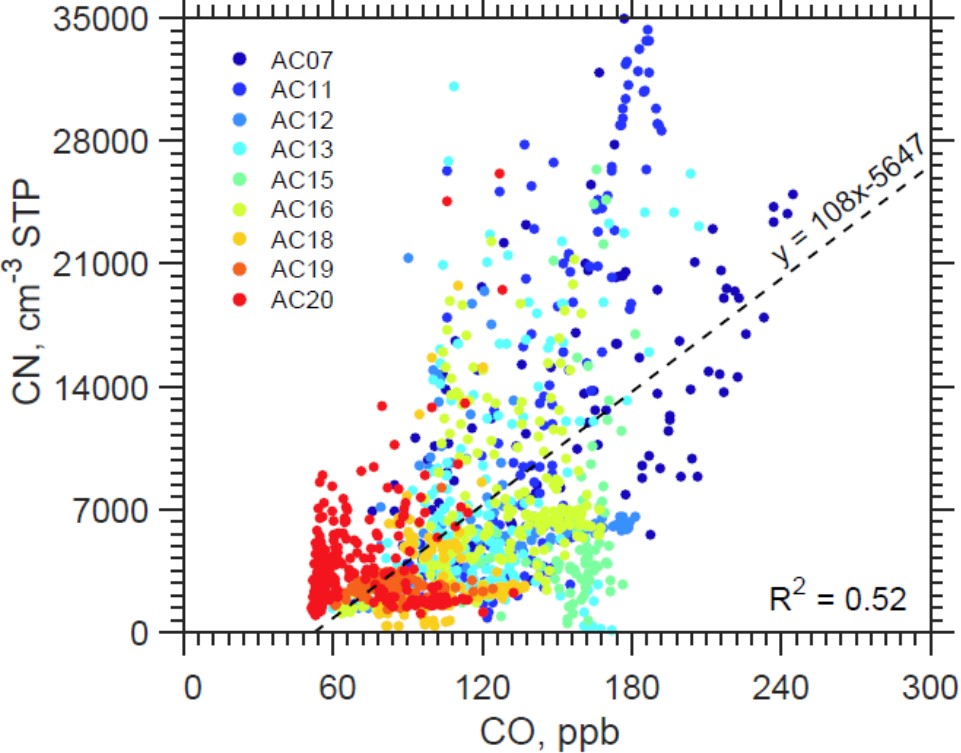

Figure 22: CN, NO and NO$_y$ in a flight segment in the upper troposphere on flight AC07.

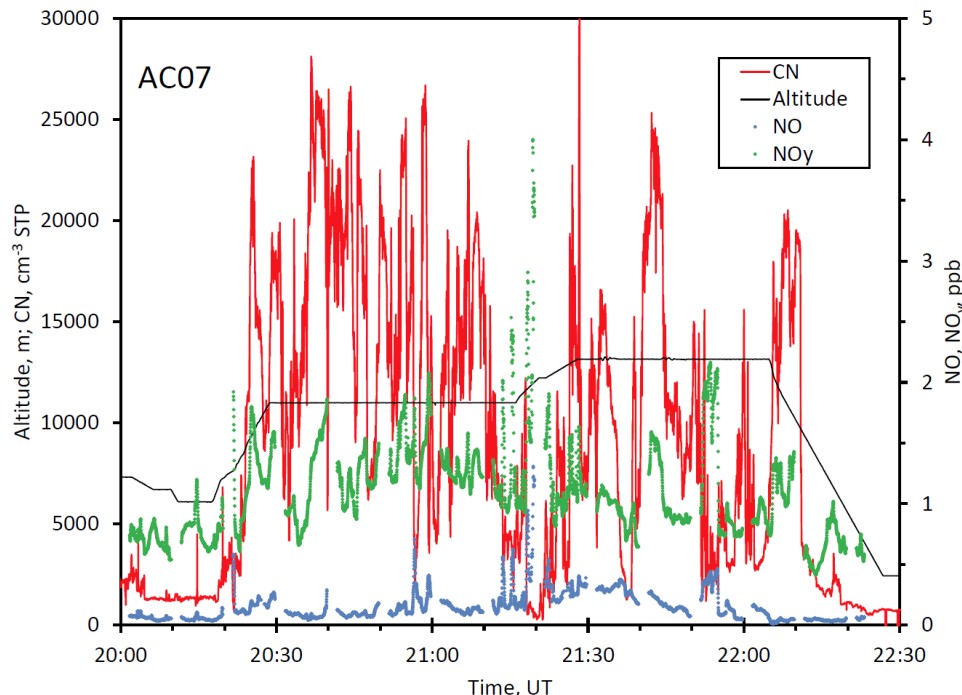



Figure 23: a) Measurements of $N_{CCN0.5}$, $N_{CN}$, $N_{nonvol}$, and ice particles during cloud top penetrations on flight AC20. b) Concentrations of CO, NO, and $NO_y$ on the same flight segments. c) Measurements of $N_{acc}$, $N_{CN}$, rBC, CO, and $O_3$ during the climb from 11.0 to 13.5 km.

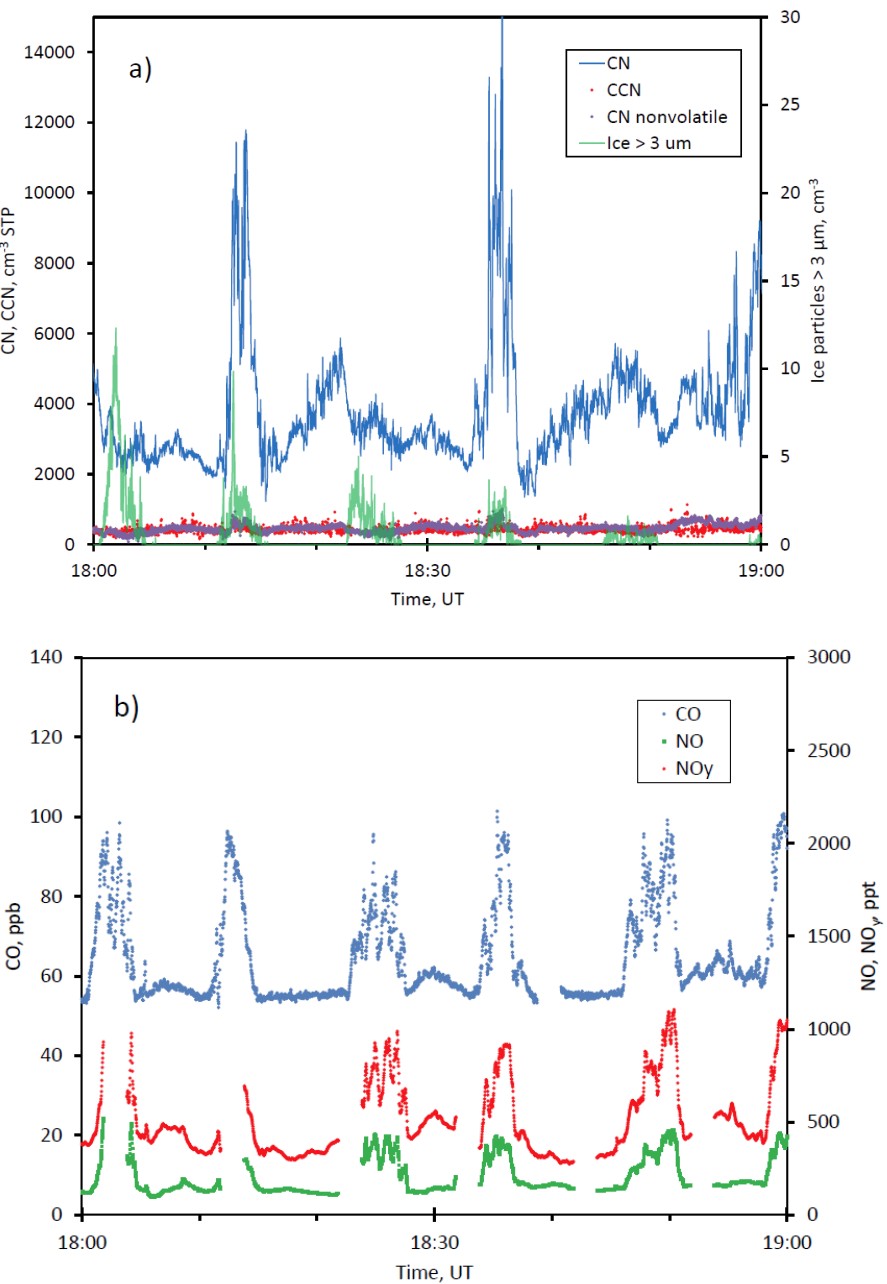





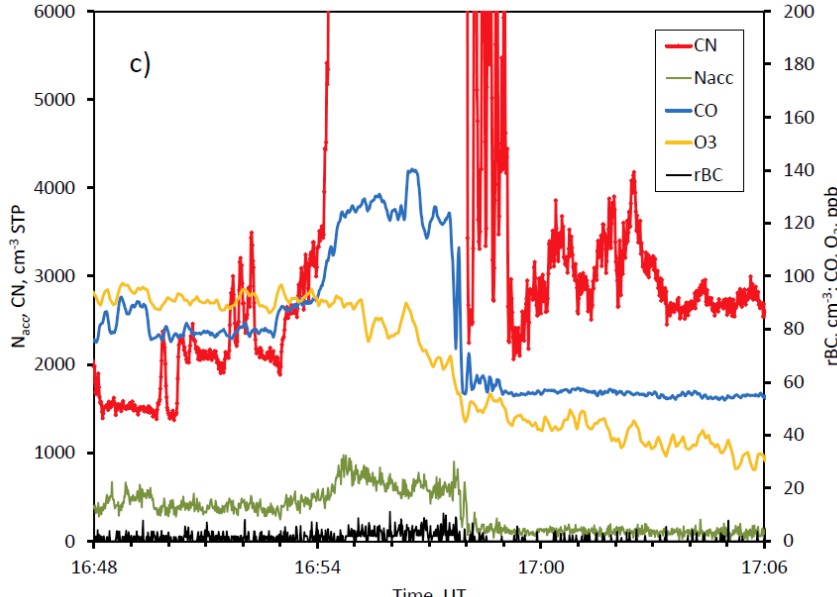



Figure 24: Conceptual model of the aerosol life cycle over the Amazon Basin

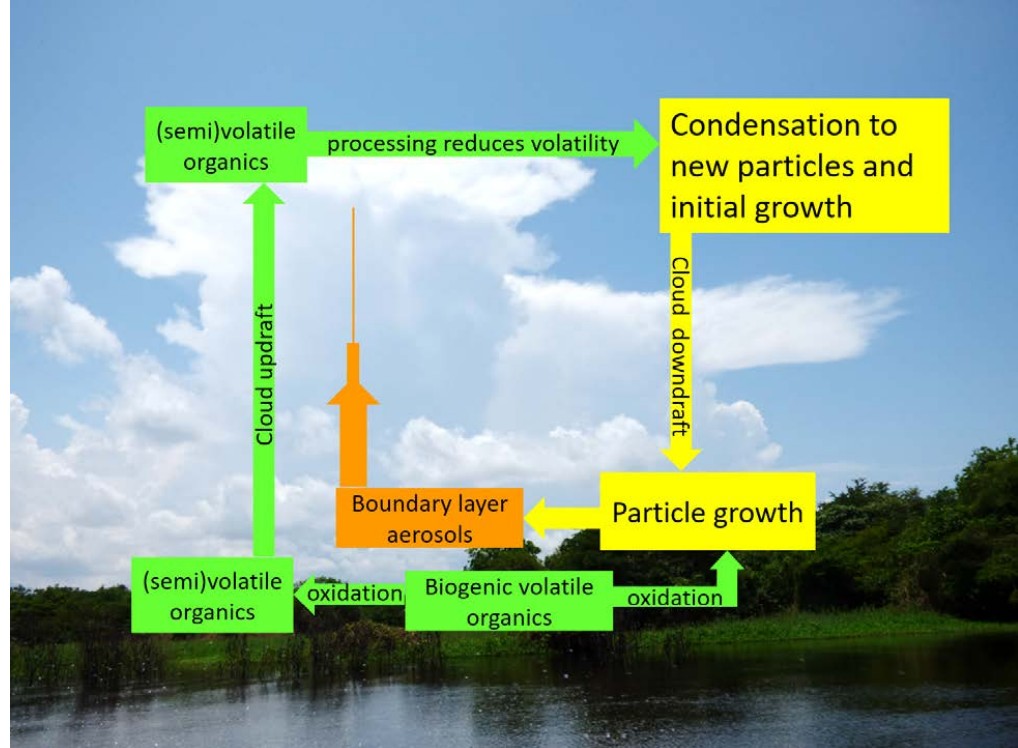