# Peer review of "Aerosol characteristics and particle production in the upper troposphere"

_Atmospheric Chemistry and Physics, 2017_

## Referee Comment (RC1) · H. Gordon (Referee) · 22 Aug 2017

**Review of Aerosol characteristics and particle production in the upper troposphere over the Amazon Basin**

**Summary**

This manuscript reports on new particle formation at high altitude in the Amazon region. I believe it is an important study and it will surely be highly cited. Addressing my "major" comments should not require substantial revisions to the manuscript.

**Major comments**

**Introduction**

Given the relatively short length of the introduction the authors do an admirable job of reviewing the relevant literature. However, I think it is necessary to highlight a couple of key papers, which otherwise are a bit lost in the long lists of citations. I didn't read all the references, but from a random selection the Twohy (2002) and Weigel (2011) papers deserve a dedicated couple of summary sentences each somewhere in the introduction to compare them with the current work.

**Methods:**

Section 2.10 outlines a sophisticated and valuable treatment of the back trajectories. Some minor clarifications on how the analysis was done, perhaps in the supplementary material, would be useful. Specifics:

1. I think it may be helpful to show trajectories in longitude-altitude or (better) time-altitude space (e.g. for Figure S1). Would this shed any light on what the model is doing in areas of deep convection? The online HYSPLIT version gives these plots by default.
2. Please can the authors expand on the footnotes in Table 1? Are the maxima and minima that are given the maximum and minimum out of the five trajectories of the five cluster centres they obtained from FLEXPART? Was the procedure explained in Figure S2 simply repeated for trajectories of each five possible cluster centres each time?
3. After the first contact with deep convection, (though not with the outflow of deep convection) presumably the five cluster centres diverge radically in horizontal and vertical positions

as the air mass is vertically redistributed. Could the authors put the trajectories of the other four cluster centres on Figure S2 (or perhaps a copy of Figure S2, to help avoid confusion) as an example? Otherwise it is hard to see where the ranges in Table 1 for the time in gridboxes with deep convection are coming from. Ideally, it would be great to see how these clusters are transported in time-altitude space, as well.

I note that Stohl et al (2002), where the clustering is introduced, does not report any validation of the algorithm in regions where deep convective clouds are present. Has this been done elsewhere? Are the five clusters really representative of the underlying distribution and does this affect the ranges for time spent in gridboxes with deep convection in Table 1? Given the huge vertical difference in winds (Figure 4 and line 449) one might speculate that the trajectories can be all over the place after contact with deep convection (though maybe not after contact only with an outflowing air mass). The authors do acknowledge this briefly (line 929) and it may not be very important if one contact with outflow is usually enough to produce NPF. However, I think these uncertainties merit a bit more discussion in the text, some kind of demonstration in a supplementary figure as I suggest above, and a brief comment in the caption of Table 1.

4. The 10-14km altitude range (e.g. line 463) seems quite high compared to many of the NPF bursts observed  -one of the examples is at 7km. Some words on what happens at slightly lower altitudes would be useful, if this can be provided without huge extra effort.
5. Is there a dependence of the NPF characteristics on trajectory type (A-E in Figure 1)? Is it possible to draw general conclusions in addition to the discussion of specific flights and the statement that only a few daylight hours are needed for the NPF, in Section 3.5?

**Results:**

1. I'm reluctant to suggest additions to an already long and comprehensive study. However, I do feel information is lacking on the air masses in the UT in which particle concentrations were low (except for the immediate cloud outflow region, which is already

described). Clearly from Fig. 7a quite a few segments with fewer than 2000 particles/cm$^3$ were seen. At line 661 the authors could remind readers of this by changing "two distinct aerosol populations" as "two types of elevated aerosol population".

If studying the air masses with very low particle concentration shows significant differences in their interaction with deep convection compared to the air masses with high particle concentrations, the authors' conceptual model may become more powerful: it may be possible to suggest contact with deep convection is a necessary condition for particle production in these situations. If no significant differences are found, this would also be interesting, though it would certainly not invalidate the conceptual model, as there are many possible explanations for the absence of NPF.

Thus, could the authors consider either adding another (shorter!) Table 1, where at least some of the flight legs where aerosol concentrations in the UT were below 2000cm$^{-3}$ are listed? Is there any systematic difference in the timings at which the air masses with few particles first made contact with deep convection, and at which the air masses with many particles made contact? I appreciate that the authors may prefer to leave this for further work if the analysis has not already been done.

2. From Figure 5, the relative humidity at 7-10km altitude is very low – apparently unusually low (line 414 ish). It may be interesting to look for evidence of the RH enhancing or suppressing the particle number concentrations- if there is any effect of RH visible, this might suggest that the new particle formation is not at the kinetic limit for the vapours involved (or that water is important for the chemistry leading to the NPF). However, again I appreciate that this kind of investigation may be more appropriate for future studies with instrumentation better able to measure organic gas-phase chemistry.

3. Related to comment #1, can the authors suggest some possible explanations for why the areas of extremely high particle concentration (suggestive of very recent new particle formation) are usually organised in thin layers?

**Conceptual model:**

In general I find the arguments in this section compelling and I have only minor comments, see below.

**Conclusions:**

At lines 1230-1238, the authors point out that in pre-industrial times, the mechanism they propose would operate unchanged, while sources of low-altitude particles would be diminished, meaning that upper-troposphere new particle formation may in some cases become the dominant source of CCN in the boundary layer. They further propose that the aerosol profile in polluted continental regions may be flipped in the pre-industrial compared to the present day.

The authors do make it clear that these statements are speculative, and I appreciate the need to be concise. However, at lines 1223-1224 I think they should additionally point out that the pre-industrial atmosphere may not have been particularly pristine in many places, with large marine, volcanic and fire emissions leading to uncertain but possibly high concentrations of boundary layer particles. It would be enough to modify "strongly affected by anthropogenic aerosols" to "strongly affected by anthropogenic or natural primary aerosols".

Furthermore, to justify the arguments in the paragraph "The conceptual model proposed here implies…" the authors need to show evidence that in present-day *polluted* areas, concentrations of particles greater than say 3nm in diameter are usually lower at high altitude than they are at low altitude. A very brief look at flight data from INTEX over the eastern USA suggested to me that there is still plenty of particle production in the upper troposphere in polluted regions (in these areas, of course there are more particles in the BL, but also more SO2 making particles in the UT). There is a modification to the gradient of the aerosol profile over the industrial period (modelling studies suggest this is true even as a global average, see for example Fig. 1a of http://onlinelibrary.wiley.com/doi/10.1002/2017JD026844/abstract) but to say "turned upside down" seems a bit strong.

**Minor comments**

The text is well written and logically structured, but as it is long, the introduction of more cross-referencing between sections to relate different parts of the text together would be very helpful.

For example, it would be helpful to reference Figures 4 and 6 at the appropriate places in the paragraph starting on line 471. Also at line 662 it would be helpful to remind the reader that the two aerosol populations were already introduced at line 547, to confirm the distinctions drawn are the same in the two cases.

Structurally, the one concern I have is that Section 3.4 and Section 3.5 start with essentially the same question, then Section 3.4 deals with one part of it and then 3.5 introduces another possible source (immediate outflows) and most of the section is then spent dealing with this new issue that was not previously introduced. Can the authors think about whether it is possible to organise these sections more rigidly and flag up the most important messages more strongly? The discussion of the trajectory results (3.5.2,3.5.3) probably merits a new section 3.6.

Line 93: the authors might cite here only the papers which really focus on UT NPF: the Carslaw (2017) citation seems out of place in this paragraph.

Line 197 or 218: please state approximate distance between inlet and instrument, to put these flow rates and efficiencies in context. Also for the UHSAS and CCNC.

The authors convincingly demonstrate NPF is the only possible source of the particles. However, they should emphasise the sentence at line 843-845 more, where the key reason for why the particles cannot come from long range transport is explained (even though it is fairly obvious). This could be done by forward referencing Section 3.5 from line 553, or restructuring slightly as suggested above.

Line 806: please label the citation to Schulz as 'submitted', or 'in preparation', here. I couldn't find the paper.

Line 1087 The authors should specify that the CERN CLOUD chamber studies so far published only provide the temperature dependence of inorganic NPF. NPF involving organic molecules may behave quite differently, though NPF is still obviously expected to increase at lower temperatures (all other things being

equal). Similarly, the Yu (2017) study does not fully account for the gas-phase chemistry (as this chemistry is not fully characterised the authors had little choice), so it treats NPF of organics rather similarly to that for H2SO4.

Line 1123 The Gordon (2016) modelling study didn't quite suggest "dominant mode of new particle formation in the pre-industrial atmosphere", perhaps replace by "in large parts of the pre-industrial atmosphere".

On page 68, the footnote labels to Table 1 all read "a".

Fig S1 caption: aren't the parcels zoomed in approximately a 6x6 degree box, not 3x3? Despite the valuable efforts of the authors to make things clear with the colour scale of the trajectories and marking the GOES time on the figure, I found the way this was phrased in the caption a little confusing. If I understand, the snapshots are zoomed in a box centred at the parcel location at the time shown on the top **of** the snapshots, **in parentheses backwards from the parcel start**. Perhaps the authors could add something like the italicised words/phrases to the caption?

**Checklist**

1. Does the paper address relevant scientific questions within the scope of ACP?

   Yes

2. Does the paper present novel concepts, ideas, tools, or data?

   Yes

3. Are substantial conclusions reached?

   Yes

4. Are the scientific methods and assumptions valid and clearly outlined?

   Yes

5. Are the results sufficient to support the interpretations and conclusions?

   Yes, if a couple of sentences in the conclusions can be toned down slightly.

6. Is the description of experiments and calculations sufficiently complete and precise to allow their reproduction by fellow scientists (traceability of results)?

   Yes

7. Do the authors give proper credit to related work and clearly indicate their own new/original contribution?

   Yes

8. Does the title clearly reflect the contents of the paper?

   Yes

9. Does the abstract provide a concise and complete summary?

   Yes

10. Is the overall presentation well structured and clear?

    Yes

11. Is the language fluent and precise?

    Yes

12. Are mathematical formulae, symbols, abbreviations, and units correctly defined and used?

    Yes

13. Should any parts of the paper (text, formulae, figures, tables) be clarified, reduced, combined, or eliminated?

    No (or no substantial part of the text)

14. Are the number and quality of references appropriate?

Yes

15.Is the amount and quality of supplementary material appropriate?

Yes.

---

## Referee Comment (RC2) · Anonymous Referee #2 · 23 Aug 2017

**Review of "Aerosol characteristics and particle production in the upper troposphere over the Amazon basin"**

**General comments**

In this study characteristics of aerosol particles over the Amazon basin are investigated using aircraft measurements. The study focuses on the layers of enhanced particle concentrations observed in the upper troposphere. The particles in these layers were found to differ from particles in the lower troposphere with respect of their concentration, size, and chemical composition. Authors show that in most cases air masses with high particle concentrations have previously been in contact with deep convective outflow. Therefore, they suggest that particles are formed in the upper troposphere from precursors vapors brought up by deep convection.

The study is of good scientific quality and certainly worth publishing in the ACP after some minor revisions. First of all, when reading the manuscript, one gets an impression that this is the first time when the conceptual model with the production of particles in the upper troposphere from material brought up by deep convection and the transport of particles back to the boundary layer is suggested (e.g. P2, L52–60). However, as the authors discuss in Sections 1 and 3.7, this is not an entirely novel idea. Therefore, the authors should make it clearer, what is new in their conceptual model, and what has been suggested before. More specific comments are presented below.

**Specific comments**

P4, L113–115: The use of terms is slightly unclear here. The current convention is to use HOMs to generally refer to highly oxygenated organic compounds, while ELVOCs are only those HOMs that have extremely low volatility. In some earlier articles all HOMs were called ELVOCs but this is not preferable.

P18, L523: Are these values means for different flights?

P18, L535: It would be good if authors presented typical ratios between concentrations in the upper troposphere and lower troposphere for different size ranges.

P19, L546: The enhancement of accumulation mode particle concentration as well as high total particle concentrations would be easier to see if particle concentrations were plotted using a logarithmic scale (this also applies to some other plots).

P21, L626: Could higher concentrations of CCN compared to accumulation mode particles be also caused by underestimation of accumulation mode particle concentration due to high losses?

P22, L646–647: Why there is a peak in CCN fraction at ~11 km?

P23, L664: Should UFF be low (instead of high) when discussing these more aged particles?

P23, L668: In Fig. 13 there seems to be AC10-F instead of AC07-F.

P23, L683: For me it is not obvious where this region with high CCN concentrations is in Fig. 11b. In any case, this region could be mentioned already when discussing the vertical distribution of CCN.

P24, L717: It is told here that the average rBC concentration below 5 km is $0.31 \pm 0.29$ g m$^{-3}$. It would be good to clarify what 0.29 g m$^{-3}$ means here (and elsewhere in this section); is it an uncertainty for the average?

P31, L915: Please report how large the fraction of the cases where these air masses had encountered deep convection is. Also, would it be possible to perform more statistical analysis of the connection between enhanced particle concentrations and deep convection, for example by studying correlation between time since contact and particle concentration?

P31, L926: Why the flight AC19 was different?

P34, L1009: Please report the correlation coefficient obtained for $N_{CN}$ and $O_3$.

P35, L1019: Please report the correlation coefficient. Also, adding a plot of $NO_y$ vs $N_{CN}$ could be useful.

P37, L1078: Check the terminology as VOCs (volatile organic compounds) cannot have low/very low volatilities by definition. Moreover, if low volatile vapors are removed in the cloud outflow, how can there be enough low-volatile vapors to form particles?

P37, L1100 & P38, L1128 & P40, L1194: Instead of "ELVOCs/HOMs" I would suggest using only "HOMs". See also the comment above.

P38, L115: Stating that pure organic nucleation is "much more likely" than nucleation including both organic and sulfuric acid appears to be a too strong statement, especially when the authors do not have data on the vapor concentrations. In the summary section, the authors also write that "we propose that BVOCs in the cloud outflow are rapidly oxidized to HOMs/ELVOCs, which because of the low temperatures and low condensation sink can readily nucleate new particles and grow to sizes ≥20 nm within a few hours". I would suggest modifying this to something like "… oxidized to HOMs, which because of the low temperature and low condensation sink can form new particles, possibly together with sulfuric acid, and condense on particles growing them to sizes >20 nm"

P39, L1160: The "Summary and conclusions" section is very long and partly seems to repeat some things discussed in the previous section. Therefore, I would suggest making the summary section shorter, especially the end of the section (starting from the line 1205). If needed, some of the text could also be moved to the previous section.

**Technical corrections**

P1, L38: Change "September/October" to "September–October"

P2, L47: Change "depleted in aerosol particles" to "depleted of aerosol particles"

P2, L49: Please change hyphen in "5-72" to en dash (–). This should be changed everywhere in the manuscript where ranges of numbers are shown.

P2, L56: Change "biogenic volatile organic carbon" to "biogenic volatile organic compounds".

P3, L74: Change "are" to "they are"

P3, L81: Rephrase this sentence so that it does not begin with "where".

P3, L82: Check the use of verb tenses in the whole manuscript. For example here "was" should be changed to "has been".

P4, L109: Please rephrase the sentence.

P7, L212: Modify the reference to follow the journal's guidelines.

P8, L244: Change to "The DMPS data *were* then analyzed *by* taking into..."

P10, L278: Change "on the S" to "on S".

P10, L284: Change "by M. Pöhlker et al." to "by Pöhlker et al."

P15, L422: Please check that the reference style follows the journal's guidelines.

P20, L591–593: The description of ultrafine fraction should be presented in a clearer way.

P21, L610: Remove "M." and add this also to the reference list.

P23, L662: Please change "at one extreme are" to "at one extreme there are". Also, change "at the other extreme are" to "at the other extreme there are".

P23, L689: The description of volatile fraction is not clear here; it is not explained what $N_{nonvol}$ stands for.

P25, L722. Change "June/July" to "June–July"

P26, L751: Please use subscripts for chemical compositions (e.g. $SO_4$, $NH_4$…)

P26, L752: When using abbreviation "BB" for the first time, please write the whole word.

P31, L903: Change "can this be reversed" to "this can be reversed"

P34, L1000: Change "close" to for example "strong"

P34, L1015: Change "2056" to "20:56" etc.

P36, L1064: "Fig. 20" should be "Fig. 24"

P36, L1064: I would suggest using some other term than "classical nucleation events", as a reader may confuse it with the classical nucleation theory. The term is used also elsewhere in the manuscript.

P37, 1090–1091: Rephrase the sentence "the low particle surface area in the UT presents very little competition to nucleation from a condensation sink", as it is slightly unclear.

P40, L1171: Please make it clear that "UT aerosol was fundamentally different from the aerosol in the LT" is the result of this study. Also, I would suggest combining this and the previous paragraph.

Table 2: Please state in the table caption what the numbers reported in the table are: means with their uncertainty ranges?

Figure 1: It is difficult to see the difference between normal and "heavier" lines, so I would recommend using some other way to distinguish them.

Figures 2–4: As the manuscript includes so many figures, I would consider moving these figures (or at least some of them) to the supplementary material.

Figure 7b: In many of the figures (especially the lower panels) font size and line thickness/dot size should be increased.

Figure 10b: The values in the figure seem to be fractions, not percentage values as indicated by the figure label.

Figure 19a: There seems to be something wrong with the y-axis label.

---

## Referee Comment (RC3) · Anonymous Referee #3 · 6 Sep 2017

**Summary**

The manuscript presents the results of the ACRIDICON-CHUVA aircraft campaign over the Amazon basin in which the characteristics of aerosol particles were determined in 11 flights that reached the upper troposphere (altitude > 8 km). Comprehensive instrumentation on board of the aircraft collected data of aerosol particle size, number, and composition, in addition to gas phase composition and cloud microphysical properties and meteorological data. The authors analyze and interpret the measurements with respect to the formation mechanism of aerosol in the upper troposphere and with regard to the upper troposphere as an aerosol source for the Amazon basin.

[Figure]

The introduction provides the background and references on upper tropospheric aerosol measurements, discusses the upper troposphere as an aerosol source, and gives a brief overview of aerosol nucleation mechanisms. The "Methods" section documents and explains the instrumentation used in the aircraft campaign, the data collection and analysis procedures, and the analysis of air mass history using back-trajectory modeling. The "Results and Discussion" section guides the reader systematically through the observations and the insights they provide. First the synoptic meteorological situation and air mass history in the Amazon region during the campaign are discussed, followed by a characterization of the atmospheric chemical composition. The authors then present and discuss the vertical distribution of aerosol particle concentration and the differences between upper- and lower-tropospheric aerosol in terms of different particle properties. Very high number concentrations and a composition dominated by organics of upper tropospheric aerosol particles stand out among the numerous findings. The authors proceed to characterize the relationship of deep convection, high aerosol number concentrations, and new particle formation in the upper troposphere. A special case of a deep convective cloud that interacts with a (fresh) biomass burning plume with new particle formation in the cloud outflow is discussed separately. Finally, a conceptual model for the aerosol life cycle in the Amazon basin is proposed. In the conceptual model, deep convection lifts boundary layer air into the middle and upper troposphere, where it is released. Preexisting aerosol particles are removed by cloud scavenging, while aerosol nucleation precursors molecules, likely biogenic volatile organic compounds, are released by the convective outflows. These nucleation precursors are oxidized to molecules that drive aerosol nucleation, and account for the very high concentrations of aerosol particles observed in the middle and upper troposphere. Large scale subsidence and downdrafts surrounding deep convection would then bring these newly formed aerosol particles into the boundary layer. The upper troposphere in the Amazon basin would therefore, in particular in pristine conditions, serve as an important and possibly dominant source of aerosol particles.

**Comments**

The present work gives a comprehensive account of aerosol properties in the atmospheric column from the boundary layer to the upper troposphere over the Amazon basin, facilitated by a well-instrumented aircraft with high-altitude and long distance capability. The results support the transport of boundary layer air containing aerosol nucleation precursors by deep convection into the upper troposphere, with subsequent new particle formation in convective outflow and growth involving organics as a key aerosol source in the region. The work is an important contribution to the understanding of the aerosol life cycle over tropical landmasses.

The text is written in an accessible and organized style with an appropriate amount of detail. A thorough description of the campaign, the instruments, measurements, procedures, and results is given. Limitations of the measurements (such as the lower aerosol size measurement cutoff of 20 nm, which hampers identification of new particle formation) and associated uncertainties in the interpretation of the results are well accounted for in the discussion. Figures are informative and integrate themselves seamlessly into the narrative.

The manuscript is nearly ready for publication, except for several points that I would like the authors to address.

1) The conceptual aerosol life cycle model in which convection lifts boundary layer air with nucleation precursor molecules into the upper troposphere, where nucleation takes place in the detrainment zone, followed by aerosol growth and descent through the troposphere into the boundary layer, has been to the best of my knowledge first formulated by A. D. Clarke (1992) based on observations and supported by subsequent investigations (e.g. Clarke, 1993; Clarke et al., 1998). These measurements were carried out over the oceans and implied sulfuric acid, likely from dimethyl sulfide and sulfur dioxide oxidation, as the molecule driving aerosol nucleation. Clarke and Kapustin (2002) wrote that "the tropics commonly have low aerosol mass but very high

number concentrations in the upper free troposphere (FT) that appear to form from sulfuric acid (nucleation) in convective regions and near cloud edges. These age and subside to become effective cloud condensation nuclei (CCN) when mixed into the marine boundary layer." This conceptual model is applied in the present manuscript to a pristine tropical continental region with organic molecules as the likely nucleation precursor. References to works by Clarke et al. and their context do, however, not provide due credit. I would like to ask the authors to add a brief paragraph in which their analysis and findings are placed into the context of this previously developed aerosol life cycle model and which provides credit to A. D. Clarke for its development with the below references.

Clarke, A. D., Atmospheric nuclei in the remote free troposphere, J. Atmos. Chem., 14, 479-488, 1992.

Clarke, A. D., Atmospheric nuclei in the Pacific midtroposphere: Their nature, concentration, and evolution, J. Geophys. Res., 98(D11), 20633-20647, doi:10.1029/93JD00797, 1993.

Clarke, A. D., J. L. Varner, F. Eisele, R. L. Mauldin, D. Tanner, and M. Litchy, Particle production in the remote marine atmosphere: Cloud out-flow and subsidence during ACE 1, J. Geophys. Res., 103, 16,397-16,409, 1998.

Clarke, A. D. and Kapustin, V. N.: A Pacific aerosol survey. Part I: A decade of data on particle production, transport, evolution, and mixing in the troposphere, J. Atmos. Sci., 52, 363-382, doi:10.1175/1520-0469(2002)059<0363:APASPI>2.0.CO;2, 2002.

2) Line 78-79: " ... or upward into the Tropical Transition Layer (TTL) and the lower stratosphere (Weigel et al., 2011; Randel and Jensen, 2013) ..."

Please add a reference to Brock et al. (1995), who identified the role of upper tropospheric aerosol nucleation for stratospheric aerosol concentrations.

C. A. Brock, P. Hamill, J. C. Wilson, H. H. Jonsson, K. R. Chan: Particle Formation

in the Upper Tropical Troposphere: A Source of Nuclei for the Stratospheric Aerosol, Science, 1650-1653, 1995

3) Line 719-723: "Interestingly, these concentrations over the Amazon Basin are only slightly higher than the values measured over the tropical Western Atlantic during the Saharan Aerosol Long-range Transport and Aerosol-Cloud-Interaction Experiment (SALTRACE), June/July 2013: ca. 0.2 ug m-3 in the LT and ca. 0.001 ug m-3 in the FT (Schwarz et al., 2017), which suggests that a significant fraction of the rBC is entering the basin by long-range transport from Africa."

It is not clear that one can make this statement simply by comparing BC mass concentrations from two campaigns that are more than year apart, without analyzing transport and the contribution of local BC sources. Can you add a supporting discussion or evidence that would corroborate the point, or instead, formulate the statement hypothetically?

---

## Author Comment (AC1) · 1 Nov 2017

**Response to Reviewer 3**
We thank the reviewer for his/her careful review and positive and constructive comments. The reviewer comments are in plain font, the responses in *Italics*

The manuscript is nearly ready for publication, except for several points that I would like the authors to address.

1) The conceptual aerosol life cycle model in which convection lifts boundary layer air with nucleation precursor molecules into the upper troposphere, where nucleation takes place in the detrainment zone, followed by aerosol growth and descent through the troposphere into the boundary layer, has been to the best of my knowledge first formulated by A. D. Clarke (1992) based on observations and supported by subsequent investigations (e.g. Clarke, 1993; Clarke et al., 1998). These measurements were carried out over the oceans and implied sulfuric acid, likely from dimethyl sulfide and sulfur dioxide oxidation, as the molecule driving aerosol nucleation. Clarke and Kapustin (2002) wrote that "the tropics commonly have low aerosol mass but very high number concentrations in the upper free troposphere (FT) that appear to form from sulfuric acid (nucleation) in convective regions and near cloud edges. These age and subside to become effective cloud condensation nuclei (CCN) when mixed into the marine boundary layer." This conceptual model is applied in the present manuscript to a pristine tropical continental region with organic molecules as the likely nucleation precursor. References to works by Clarke et al. and their context do, however, not provide due credit. I would like to ask the authors to add a brief paragraph in which their analysis and findings are placed into the context of this previously developed aerosol life cycle model and which provides credit to A. D. Clarke for its development with the below references.

Clarke, A. D., Atmospheric nuclei in the remote free troposphere, J. Atmos. Chem., 14, 479-488, 1992.

Clarke, A. D., Atmospheric nuclei in the Pacific midtroposphere: Their nature, concentration, and evolution, J. Geophys. Res., 98(D11), 20633-20647, doi:10.1029/93JD00797, 1993.

Clarke, A. D., J. L. Varner, F. Eisele, R. L. Mauldin, D. Tanner, and M. Litchy, Particle production in the remote marine atmosphere: Cloud out-flow and subsidence during ACE 1, J. Geophys. Res., 103, 16,397-16,409, 1998.

Clarke, A. D. and Kapustin, V. N.: A Pacific aerosol survey. Part I: A decade of data on particle production, transport, evolution, and mixing in the troposphere, J. Atmos. Sci., 52, 363-382, doi:10.1175/1520-0469(2002)059<0363:APASPI>2.0.CO;2, 2002.

*As submitted, the paper contained 15 references to the work of Clarke and coworkers. In accordance with the reviewer's suggestion, we have added the suggested four new references to Clarke's work and included the following paragraph in the introduction: "Based on observations over the remote Pacific and supported by extensive subsequent investigations, Clarke and coworkers proposed an aerosol life cycle model in which convection lifts boundary layer air with nucleation precursor molecules into the upper troposphere, where nucleation takes place in the detrainment zone, followed by aerosol growth and descent through the troposphere into the boundary layer (Clarke, 1992; Clarke, 1993; Clarke et al., 1998). These measurements were carried out over the oceans and implied sulfuric acid, likely from dimethyl*

*sulfide and sulfur dioxide oxidation, as the molecule driving aerosol nucleation. Clarke and Kapustin (2002) wrote that "the tropics commonly have low aerosol mass but very high number concentrations in the upper free troposphere (FT) that appear to form from sulfuric acid (nucleation) in convective regions and near cloud edges. These age and subside to become effective cloud condensation nuclei (CCN) when mixed into the marine boundary layer.""*
*In section 3.7, we are contrasting our model to that of Clark and other workers in several important aspects, e.g., the role of organics vs sulfates and the mechanism of downward transport. See also our response to the first comment by Reviewer 2.*

2) Line 78-79: " ... or upward into the Tropical Transition Layer (TTL) and the lower stratosphere (Weigel et al., 2011; Randel and Jensen, 2013) ..." Please add a reference to Brock et al. (1995), who identified the role of upper tropospheric aerosol nucleation for stratospheric aerosol concentrations. C. A. Brock, P. Hamill, J. C. Wilson, H. H. Jonsson, K. R. Chan: Particle Formation in the Upper Tropical Troposphere: A Source of Nuclei for the Stratospheric Aerosol, Science, 1650-1653, 1995

*Done.*

3) Line 719-723: "Interestingly, these concentrations over the Amazon Basin are only slightly higher than the values measured over the tropical Western Atlantic during the Saharan Aerosol Long-range Transport and Aerosol-Cloud-Interaction Experiment (SALTRACE), June/July 2013: ca. 0.2 ug m-3 in the LT and ca. 0.001 ug m-3 in the FT (Schwarz et al., 2017), which suggests that a significant fraction of the rBC is entering the basin by long-range transport from Africa." It is not clear that one can make this statement simply by comparing BC mass concentrations from two campaigns that are more than year apart, without analyzing transport and the contribution of local BC sources. Can you add a supporting discussion or evidence that would corroborate the point, or instead, formulate the statement hypothetically?

*We have a considerable amount of evidence for the transport of BC and other aerosol constituents from Africa to the Amazon Basin from several campaigns. Recently, we have published a modeling study on this topic (Wang et al., 2016). We are currently preparing a paper in which we are documenting the transport of biomass smoke from Southern Africa to the Amazon during ACRIDICON-CHUVA. This has also been observed in previous campaigns, e.g., Andreae et al. (1994). We have added the following text to section 3.4.4:*

*"Transport of biomass smoke containing BC and other constituents from Africa to South America has been documented previously, e.g., from Northern Africa during the wet season (Talbot et al., 1990; Wang et al., 2016) and from Southern Africa during the dry season (Andreae et al., 1994). A detailed study on the transport of Southern African aerosols to the Amazon during ACRIDICON-CHUVA is in preparation and will be published elsewhere."*

Clarke, A. D., Atmospheric nuclei in the remote free troposphere: J. Atmos. Chem., 14, 479-488, doi:10.1007/bf00115252, 1992.

Clarke, A. D., Atmospheric nuclei in the Pacific midtroposphere - their nature, concentration, and evolution: J. Geophys. Res., 98, 20,633-20,647, doi:10.1029/93jd00797, 1993.

Clarke, A. D., Varner, J. L., Eisele, F., Mauldin, R. L., Tanner, D., and Litchy, M., Particle production in the remote marine atmosphere: Cloud outflow and subsidence during ACE 1: J. Geophys. Res., 103, 16,397-16,409, doi:10.1029/97jd02987, 1998.

Clarke, A. D., and Kapustin, V. N., A Pacific aerosol survey. Part I: A decade of data on particle production, transport, evolution, and mixing in the troposphere: J. Atmos. Sci., 59, 363-382, 2002.

Wang, Q., Saturno, J., Chi, X., Walter, D., Lavric, J. V., Moran-Zuloaga, D., Ditas, F., Pöhlker, C., Brito, J., Carbone, S., Artaxo, P., and Andreae, M. O., Modeling investigation of light-absorbing aerosols in the Amazon Basin during the wet season: Atmos. Chem. Phys., 16, 14,775-14,794, doi:10.5194/acp-16-14775-2016, 2016.

Andreae, M. O., Anderson, B. E., Blake, D. R., Bradshaw, J. D., Collins, J. E., Gregory, G. L., Sachse, G. W., and Shipham, M. C., Influence of plumes from biomass burning on atmospheric chemistry over the equatorial Atlantic during CITE-3: J. Geophys. Res., 99, 12,793-12,808, 1994.

---

## Author Comment (AC2) · 2 Nov 2017

**Response to Reviewer 2**
We thank the reviewer for his/her positive and constructive comments and for his/her thorough review. The reviewer comments are in plain font, the responses in *Italics*.

**General comments**
In this study characteristics of aerosol particles over the Amazon basin are investigated using aircraft measurements. The study focuses on the layers of enhanced particle concentrations observed in the upper troposphere. The particles in these layers were found to differ from particles in the lower troposphere with respect of their concentration, size, and chemical composition. Authors show that in most cases air masses with high particle concentrations have previously been in contact with deep convective outflow. Therefore, they suggest that particles are formed in the upper troposphere from precursors vapors brought up by deep convection. The study is of good scientific quality and certainly worth publishing in the ACP after some minor revisions. First of all, when reading the manuscript, one gets an impression that this is the first time when the conceptual model with the production of particles in the upper troposphere from material brought up by deep convection and the transport of particles back to the boundary layer is suggested (e.g. P2, L52– 60). However, as the authors discuss in Sections 1 and 3.7, this is not an entirely novel idea. Therefore, the authors should make it clearer, what is new in their conceptual model, and what has been suggested before. More specific comments are presented below.

*In the introduction, we now write "…where production of new aerosol particles takes place in the UT from **biogenic volatile organic material** brought up by deep convection…" to highlight the fact that our model is based on BVOC, whereas previous authors have mostly considered sulfur compounds or organics from pollution, including biomass burning. We have also added a paragraph to the introduction, making special reference to the work of Clarke and coworkers. See also our response to the first comment by Reviewer 3.*

*In section 3.7, we refer extensively to previous work:*

*"The outflow regions in the UT present an ideal environment for particle nucleation, as had already been suggested in some earlier studies (Twohy et al., 2002; Lee et al., 2004; Kulmala et al., 2006; Weigelt et al., 2009)."*

*"Over marine regions and polluted continental regions, the particles observed in outflows and in the UT were mostly identified as sulfates (Clarke et al., 1999; Twohy et al., 2002; Kojima et al., 2004; Waddicor et al., 2012), and consequently $H_2SO_4$ has been proposed as the nucleating species."*

*We then go on to propose that, in contrast to these studies, organics may be the nucleating species, although a final proof still has to await our next campaign.*

*We also highlight the difference in the proposed mechanism of downward transport: "Large-scale entrainment of UT and MT air into the boundary layer has been suggested as the major source of new particles in marine regions (Raes, 1995; Katoshevski et al., 1999; Clarke et al., 2013). Over Amazonia with its high degree of convective activity, downdrafts are likely to play a more important role."*

*We never make the claim that "…this is the first time when the conceptual model with the production of particles in the upper troposphere from material brought up by deep convection and the transport of particles back to the boundary layer is suggested…". In fact, the reviewer*

*says so him/herself: "… as the authors discuss in Sections 1 and 3.7, this is not an entirely novel idea…".*

*Many more examples could be given. We feel that we have discussed previous work extensively in the introduction, in section 3.7, and in the conclusions. We find it difficult so see what more we could do to put our study in the context of previous work without repeating ourselves.*

**Specific comments**
P4, L113–115: The use of terms is slightly unclear here. The current convention is to use HOMs to generally refer to highly oxygenated organic compounds, while ELVOCs are only those HOMs that have extremely low volatility. In some earlier articles all HOMs were called ELVOCs but this is not preferable.

*In the community working on HOMs and ELVOCs there is currently no commonly accepted convention on terminology. Some authors suggest abandoning ELVOCs altogether and calling everything HOMs, while others are not using HOMs at all. In the Introduction we state "Extremely low volatility organic compounds (ELVOCs, which may be at least in part identical to HOMs)…". In section 3.7 and the conclusions, we either use "ELVOCs/HOMs" or use the term that the authors of the papers use in the work that we are referencing.*

P18, L523: Are these values means for different flights?

*The values are meant to reflect the range of quartiles above 8 km. This has been clarified in the text.*

P18, L535: It would be good if authors presented typical ratios between concentrations in the upper troposphere and lower troposphere for different size ranges.

*There is an entire section devoted to this issue, section 3.4.1, which discusses the ratio between ultrafine and accumulation mode particles (expressed as ultrafine fraction, UFF). Averages for the particle concentrations in the different size classes are given in Table 2, to which we now refer to in the first paragraph of section 3.3 by "…, and average concentrations for the particle concentrations in the different size classes and altitude regions are given in Table 2". We have also added the magnitude of the ratio in the text: "On average, $N_{CN}$ in the UT were almost five times as high as in the LT." and "On average, $N_{acc}$ in the UT was only about half the concentration measured in the LT."*

P19, L546: The enhancement of accumulation mode particle concentration as well as high total particle concentrations would be easier to see if particle concentrations were plotted using a logarithmic scale (this also applies to some other plots).

*We disagree. We started with log plots and switched to linear ones because they showed the differences much more clearly.*

P21, L626: Could higher concentrations of CCN compared to accumulation mode particles be also caused by underestimation of accumulation mode particle concentration due to high losses?

*The accumulation mode particles were measured by a UHSAS in a wing pod. There is no evidence for particle losses with this setup, which has been tested thoroughly, see also the paper by Walser et al. (2017) referenced in section 2.4.*

P22, L646–647: Why there is a peak in CCN fraction at ~11 km?

*The high values of the CCN fraction at this altitude are caused by the inclusion of a large number of measurements from flight AC20 on a horizontal leg at 11 km. This layer has only modest CN concentrations (around 1700 $cm^{-3}$), but elevated CCN, $NO_y$, CO, and aerosol nitrate and organics, with similar values to the biomass-burning-polluted boundary layer below. This flight was exceptional in that it was the only flight during the campaign on which we had evidence for transport of biomass smoke to the UT (see section 3.6). We included a short explanation and a forward reference to section 3.6 in the caption to Fig. 12a. We also added the following sentence in section 3.6:* "Further evidence for the upward transport of biomass smoke was found in measurements on a horizontal leg at 11 km, which had only modest CN concentrations (around 1700 $cm^{-3}$), but elevated CCN, $NO_y$, CO, and aerosol nitrate and organics, with similar vales to the biomass-burning-polluted boundary layer below."

P23, L664: Should UFF be low (instead of high) when discussing these more aged particles?

*Corrected.*

P23, L668: In Fig. 13 there seems to be AC10-F instead of AC07-F.

*We corrected the label in Fig. 13.*

P23, L683: For me it is not obvious where this region with high CCN concentrations is in Fig. 11b. In any case, this region could be mentioned already when discussing the vertical distribution of CCN.

*The concentrations in this region were not dramatically elevated, only up to about 1500 $cm^{-3}$. We changed the text to make this clearer. It would not have been appropriate to mention this region earlier, since it is specific to flight AC13, which is discussed in this paragraph as an illustrative example.*

P24, L717: It is told here that the average rBC concentration below 5 km is 0.31±0.29 g m-3. It would be good to clarify what 0.29 g m-3 means here (and elsewhere in this section); is it an uncertainty for the average?

*Here, and everywhere else, we give averages and standard deviations, unless stated otherwise. To make this clear, we have added this definition of our notation in section 3.2.1, where it is used first.*

P31, L915: Please report how large the fraction of the cases where these air masses had encountered deep convection is. Also, would it be possible to perform more statistical analysis of

the connection between enhanced particle concentrations and deep convection, for example by studying correlation between time since contact and particle concentration?

*Actually, the fraction was 100%! The "almost" was left over from when we had not yet done the analysis for all cases, and has now been removed. We looked for such correlations, but could not find anything obvious. Unfortunately, since the mission objectives had been focused on aerosol/cloud-microphysics interactions, the flights were not designed to look into this issue. We plan to conduct a dedicated campaign in the future.*

P31, L926: Why the flight AC19 was different?

*Most of this flight took place outside of the Amazon basin, off the east coast of South America over the Atlantic.*

P34, L1009: Please report the correlation coefficient obtained for $N_{CN}$ and $O_3$.

*Because of the great variability in the $O_3$ concentrations in the UT, there is no general correlation for the entire mission ($r^2=0.02$). For individual flights, modest but significant correlations emerge, which are still affected by the high variability of both variables. We added the following text:*
"Because of the great variability in the $O_3$ concentrations in the UT, there is no general correlation between $N_{CN}$ and $O_3$ for the entire mission ($r^2=0.02$). For individual flights, modest, but statistically significant, negative correlations can be found, e.g., an $r^2$ value of 0.13 (N=8509) in the UT on flight AC09. The scatter plot in Fig. S08 shows that high $O_3$ concentrations were always associated with low $N_{CN}$, but that there were low-$O_3$ regions in the UT both with and without enhanced particle concentrations."

P35, L1019: Please report the correlation coefficient. Also, adding a plot of NOy vs $N_{CN}$ could be useful.

*Again, there is no significant overall correlation. As pointed out in the text, the relationships are very complex because the transformation of NO and the formation of particles both occur on short timescales that cannot be resolved by a general correlation analysis. In the paper, we provide some examples of these interactions, but a full analysis if the nitrogen oxide chemistry and its role in aerosol formation in the UT must await a dedicated mission.*

P37, L1078: Check the terminology as VOCs (volatile organic compounds) cannot have low/very low volatilities by definition. Moreover, if low volatile vapors are removed in the cloud outflow, how can there be enough low-volatile vapors to form particles?

*We replaced "VOCs" by "organic compounds". However, we remind the reviewer that the terms LVOCs and ELVOCs are very commonly used in the literature. The low-volatile vapors that form the new particles are produced by the oxidation of volatile vapors by photochemistry in the UT, as discussed in the subsequent paragraphs. For a better flow of the discussion we have moved the paragraph with this discussion up, to follow directly after the statement referred to by the reviewer.*

P37, L1100 & P38, L1128 & P40, L1194: Instead of "ELVOCs/HOMs" I would suggest using only "HOMs". See also the comment above.

*We responded to this suggestion already above.*

P38, L115: Stating that pure organic nucleation is "much more likely" than nucleation including both organic and sulfuric acid appears to be a too strong statement, especially when the authors do not have data on the vapor concentrations. In the summary section, the authors also write that "we propose that BVOCs in the cloud outflow are rapidly oxidized to HOMs/ELVOCs, which because of the low temperatures and low condensation sink can readily nucleate new particles and grow to sizes ≥20 nm within a few hours". I would suggest modifying this to something like "… oxidized to HOMs, which because of the low temperature and low condensation sink can form new particles, possibly together with sulfuric acid, and condense on particles growing them to sizes >20 nm"

*We changed "much more likely" to "likely" and changed the sentence in the summary to include the possible role of $H_2SO_4$, as suggested by the reviewer.*

P39, L1160: The "Summary and conclusions" section is very long and partly seems to repeat some things discussed in the previous section. Therefore, I would suggest making the summary section shorter, especially the end of the section (starting from the line 1205). If needed, some of the text could also be moved to the previous section.

*We disagree. This paper describes a very complex data set with a large range of information from many different instruments, from atmospheric transport models, and remote sensing. In the Summary and Conclusions we have tried to pull this information together in a concise way. The summary part must necessarily repeat, to a certain extent, things that have been said before. The end of the section is very important, since it puts the results into a "big picture" perspective.*

**Technical corrections**
P1, L38: Change "September/October" to "September–October"
*Done*
P2, L47: Change "depleted in aerosol particles" to "depleted of aerosol particles"
*Done*
P2, L49: Please change hyphen in "5-72" to en dash (–). This should be changed everywhere in the manuscript where ranges of numbers are shown.
*Done*
P2, L56: Change "biogenic volatile organic carbon" to "biogenic volatile organic compounds".
*Done*
P3, L74: Change "are" to "they are"
*Done*
P3, L81: Rephrase this sentence so that it does not begin with "where".
*Done*
P3, L82: Check the use of verb tenses in the whole manuscript. For example, here "was" should be changed to "has been".

*"was" is correct here.*
P4, L109: Please rephrase the sentence.
*Done*
P7, L212: Modify the reference to follow the journal's guidelines.
*This reference is a place holder and will be updated when the final files are prepared.*
P8, L244: Change to "The DMPS data *were* then analyzed *by* taking into..."
*Done*
P10, L278: Change "on the S" to "on S".
*Done*
P10, L284: Change "by M. Pöhlker et al." to "by Pöhlker et al."
*Since there are references to two different Pöhlkers as first authors, we use the initial to differentiate them. If the journal does not like this, the copyeditor is free to change it.*
P15, L422: Please check that the reference style follows the journal's guidelines.
*This can be checked by the copyeditor.*
P20, L591–593: The description of ultrafine fraction should be presented in a clearer way.
*We can't think of a clearer way. The definition equation is clear and unambiguous. If the editor disagrees, we would appreciate a suggestion for a better expression.*
P21, L610: Remove "M." and add this also to the reference list.
*Reference deleted.*
P23, L662: Please change "at one extreme are" to "at one extreme there are". Also, change "at the other extreme are" to "at the other extreme there are".
*Done*
P23, L689: The description of volatile fraction is not clear here; it is not explained what Nnonvol stands for.
*Definition added.*
P25, L722. Change "June/July" to "June–July"
*Done*
P26, L751: Please use subscripts for chemical compositions (e.g. SO4, NH4…)
*These are in fact not chemical formulae, in which case they would have to be written also with the ionic charges, but abbreviations that are commonly used in the AMS literature.*
P26, L752: When using abbreviation "BB" for the first time, please write the whole word.
*Done*
P31, L903: Change "can this be reversed" to "this can be reversed"
*Retained as is. Can be changed by copyeditor if necessary.*
P34, L1000: Change "close" to for example "strong"
*"close correlation" is very common usage. We don't think "strong" would be better.*
P34, L1015: Change "2056" to "20:56" etc.
*This notation is very common in the meteorological literature.*
P36, L1064: "Fig. 20" should be "Fig. 24"
*Corrected*
P36, L1064: I would suggest using some other term than "classical nucleation events", as a reader may confuse it with the classical nucleation theory. The term is used also elsewhere in the manuscript.
*We put "classical" in quotes to distinguish it from other usages.*
P37, 1090–1091: Rephrase the sentence "the low particle surface area in the UT presents very little competition to nucleation from a condensation sink", as it is slightly unclear.

*Done*

P40, L1171: Please make it clear that "UT aerosol was fundamentally different from the aerosol in the LT" is the result of this study.

*Done*

Table 2: Please state in the table caption what the numbers reported in the table are: means with their uncertainty ranges?

*Done*

Figure 1: It is difficult to see the difference between normal and "heavier" lines, so I would recommend using some other way to distinguish them.

*We increased the thickness contrast between the lines.*

Figures 2–4: As the manuscript includes so many figures, I would consider moving these figures (or at least some of them) to the supplementary material.

*We disagree. This meteorological information is essential to understand the context of the mission.*

Figure 7b: In many of the figures (especially the lower panels) font size and line thickness/dot size should be increased.

*The figures were somewhat preliminary. They will be updated for the final submitted files.*

Figure 10b: The values in the figure seem to be fractions, not percentage values as indicated by the figure label.

*Percentages are one way to express a fraction.*

Figure 19a: There seems to be something wrong with the y-axis label.

*Fixed.*

---

## Author Comment (AC3) · 2 Nov 2017

**Response to Reviewer 1  (H. Gordon)**
The reviewer comments are in Arial, the responses in Times New Roman

**Summary**
This manuscript reports on new particle formation at high altitude in the Amazon region. I believe it is an important study and it will surely be highly cited. Addressing my "major" comments should not require substantial revisions to the manuscript.

We thank the reviewer for his/her positive statements and substantive comments and suggestions.

**Major comments**
**Introduction**
Given the relatively short length of the introduction the authors do an admirable job of reviewing the relevant literature. However, I think it is necessary to highlight a couple of key papers, which otherwise are a bit lost in the long lists of citations. I didn't read all the references, but from a random selection the Twohy (2002) and Weigel (2011) papers deserve a dedicated couple of summary sentences each somewhere in the introduction to compare them with the current work.

We have added the following sentences: "Twohy et al. (2002) observed particle concentrations up to 45,000 cm$^{-3}$ over North America and suggested that they had been formed in situ from gas-phase precursors brought up by deep convection. Weigel et al. (2011) found similar concentrations in the UT over tropical America, Africa, and Australia, which they attributed to new particle formation from sulfuric acid and possibly organics." The Twohy et al. paper is cited three times in the introduction and four times in the discussion. The Weigel et al. paper is cited six times in the introduction and three times in the discussion. The results from both papers are compared to ours in the discussion.

**Methods:**
Section 2.10 outlines a sophisticated and valuable treatment of the back trajectories. Some minor clarifications on how the analysis was done, perhaps in the supplementary material, would be useful.

Specifics:
1. I think it may be helpful to show trajectories in longitude-altitude or (better) time-altitude space (e.g. for Figure S1). Would this shed any light on what the model is doing in areas of deep convection? The online HYSPLIT version gives these plots by default.

We have added a longitude-altitude plot to Figure S1. Like the vast majority of the UT trajectories, this one remains in the UT over the time frame considered. The trajectory model does not resolve individual convective elements, but only incorporates a general parameterization of vertical movement. See also the response to comment 3 below.

2. Please can the authors expand on the footnotes in Table 1? Are the maxima and minima that are given the maximum and minimum out of the five trajectories of the five

cluster centres they obtained from FLEXPART? Was the procedure explained in Figure S2 simply repeated for trajectories of each five possible cluster centres each time?

For simplicity, out of the five clusters, we consider only the center cluster given by FLEXPART. Therefore, the minima and maxima values of Table1 correspond only the values of center clusters trajectories within the flight leg time frame traced backwards up to 120 hours. This is now explained in the text. Doing the analysis for all five clusters would require an extraordinary amount of work and is not likely to give any other results, given the high abundance of deep convection in the basin. We have added the following sentence to the text: "For simplicity, out of the five clusters, we consider only the center cluster given by FLEXPART. Therefore, all trajectories mentioned hereafter refer to the center trajectory."

3. After the first contact with deep convection, (though not with the outflow of deep convection) presumably the five cluster centres diverge radically in horizontal and vertical positions as the air mass is vertically redistributed. Could the authors put the trajectories of the other four cluster centres on Figure S2 (or perhaps a copy of Figure S2, to help avoid confusion) as an example? Otherwise it is hard to see where the ranges in Table 1 for the time in gridboxes with deep convection are coming from. Ideally, it would be great to see how these clusters are transported in time-altitude space, as well. I note that Stohl et al (2002), where the clustering is introduced, does not report any validation of the algorithm in regions where deep convective clouds are present. Has this been done elsewhere? Are the five clusters really representative of the underlying distribution and does this affect the ranges for time spent in gridboxes with deep convection in Table 1? Given the huge vertical difference in winds (Figure 4 and line 449) one might speculate that the trajectories can be all over the place after contact with deep convection (though maybe not after contact only with an outflowing air mass).

The reviewer here points to a major problem with this and all other trajectory models. Fundamentally, they rely on the meteorological data from weather models which do not resolve individual convective elements. Convection is only represented in a parameterized way and therefore reflects the general vertical movement of an airmass, but not an individual parcel subject to a convective event. Thus, they cannot trace a parcel backwards through a convective event. The best they can do is show that a parcel came into the vicinity of a convective event, und thus was likely to be affected by the outflow. Coming close to a convective event does not make the parcels diverge, because the trajectory model actually does not see the event. Fundamentally, this is correct behavior, because the air in the outflow joins the general flow in the upper troposphere, and only those subparcels that actually came up through the cloud "should" have backtrajectories that go down through the cloud. Thus, if a back-tracked air parcel is not an outflow parcel, it should track backwards with the mean flow as represented by the model. It is thus legitimate to keep following it backward to perhaps encountering another region of convective outflow. The actual processes can only be resolved by a dedicated mission looking at the development of an individual outflow in a Lagrangian sense, which we hope to do in the future.

The authors do acknowledge this briefly (line 929) and it may not be very important if one contact with outflow is usually enough to produce NPF. However, I think these uncertainties merit a bit more discussion in the text, some kind of demonstration in a supplementary figure as I suggest above, and a brief comment in the caption of Table 1.

We've attempted to clarify this situation as concisely as possible by modifying the text at line 929 (old) by writing:
"Because the model does not "see" the individual convective event that brings up an outflow, it cannot trace a parcel back into this outflow and back down to the boundary layer. On the other hand, an air parcel that passed through the vicinity of the outflow, but is not part of the actual outflow, will keep moving backward along the mean flow in the UT and may then encounter another outflow. Obviously, however, the uncertainty in the trajectory position increases with time going backwards, and is probably enhanced by passage near a region of active convection." Given that our analysis shows that, in view of the frequency of convection over Amazonia and the generally long residence time of air parcels in the anticyclonic movement over the basin, almost all air parcels will pass near convection over a 72-hour time frame, it does not seem worthwhile to go much further in this analysis. See also our comment below in our response to remark 1 in the results section.

4. The 10-14km altitude range (e.g. line 463) seems quite high compared to many of the NPF bursts observed -one of the examples is at 7km. Some words on what happens at slightly lower altitudes would be useful, if this can be provided without huge extra effort.

Actually, the statement in line 463 was incorrect and, as can be seen in Table 1, the analysis was done for all enriched layers, including those at 7 km.

5. Is there a dependence of the NPF characteristics on trajectory type (A-E in Figure 1)?

We could not identify any obvious relationship.

Is it possible to draw general conclusions in addition to the discussion of specific flights and the statement that only a few daylight hours are needed for the NPF, in Section 3.5?

We don't feel that we can draw further generalizations based on the kind of data we have from this mission. To go further, different flight strategies and instrumentation would be required, which we plan to deploy on a future mission.

**Results:**
1. I'm reluctant to suggest additions to an already long and comprehensive study. However, I do feel information is lacking on the air masses in the UT in which particle concentrations were low (except for the immediate cloud outflow region, which is already described). Clearly from Fig. 7a quite a few segments with fewer than 2000 particles/cm3 were seen. At line 661 the authors could remind readers of this by changing "two distinct aerosol populations" as "two types of elevated aerosol population".

Done.

If studying the air masses with very low particle concentration shows significant differences in their interaction with deep convection compared to the air masses with high particle concentrations, the authors' conceptual model may become more powerful: it may be possible to suggest contact with deep convection is a necessary condition for particle production in these situations. If no significant differences are found, this would also be interesting, though it would certainly not invalidate the conceptual model, as there are many possible explanations for the absence of NPF. Thus, could the authors consider either adding another (shorter!) Table 1, where at least some of the flight legs where aerosol concentrations in the UT were below 2000 cm-3 are listed?

We felt this was a very valuable suggestion by the reviewer and examined our data for such legs. To our disappointment it was almost impossible to find such segments. Because of the high variability of the CN concentrations in the UT, the times when $N_{CN}$ was below 2000 cm$^{-3}$ were in almost all cases very short, and would not lend themselves to a meaningful analysis of airmass history. To illustrate this, we show a full time series plot of the measurements from Flight AC09 in the supplement:

[Figure]

The only exception to this were segments that were within a Cb outflow.

We were able to find only six segments, where $N_{CN}$ was consistently below 3000 cm$^{-3}$, and which were not identifiably part of an outflow. These are listed in Table S1 in the

supplement. The segments from flights AC16 and AC18 were well away from clouds, whereas those from AC19 and 20 were in the vicinity of Cbs, but not clearly in an outflow. The segment L from AC19 is low in CN, but actually has a relatively high $N_{CCN0.5}$, and may not really be significantly different from the aged enriched segment E2, which follows immediately after it. The airmass trajectory types in these segments do not contain type D, i.e., recirculation within the Amazon basin. Notably, the air in the segments from AC20, which had the lowest particle concentrations, had come in straight from the Pacific within the last 48 hours. We added the following text to section 3.5.2:

"To test whether there was a difference in the airmass histories between segments with high and low $N_{CN}$, we searched our data for suitable segments with low $N_{CN}$. However, because of the high variability of the CN concentrations in the UT, the times when $N_{CN}$ was below 3000 cm$^{-3}$ were in almost all cases very short, and would not lend themselves to a meaningful analysis of airmass history. To illustrate this, we show a full time series plot of the measurements from Flight AC09 in the supplement (Fig. S7).

We could find only six segments, where $N_{CN}$ was consistently below 3000 cm$^{-3}$, and which were not identifiably part of an outflow. These are listed in Table S1 in the supplement. The segments from flights AC16 and AC18 were well away from clouds, whereas those from AC19 and 20 were in the vicinity of Cbs, but not clearly in an outflow. The segment L from AC19 is low in CN, but actually has a relatively high $N_{CCN0.5}$, and may not really be significantly different from the aged enriched segment E2, which follows immediately after it. Consequently, we don't have a data set that would allow a representative analysis of the history of airmasses with low particle concentrations. Notably, however, the airmass trajectory types in these segments do not contain type D, i.e., recirculation within the Amazon basin. The air in the segments from AC20, which had the lowest particle concentrations, had come in straight from the Pacific within the last 48 hours, but may also contain some outflow air."

Is there any systematic difference in the timings at which the air masses with few particles first made contact with deep convection, and at which the air masses with many particles made contact? I appreciate that the authors may prefer to leave this for further work if the analysis has not already been done.

Again, we feel that in view of the complexity of the airmass histories, dedicated campaigns are needed to resolve this question.

2. From Figure 5, the relative humidity at 7-10km altitude is very low – apparently unusually low (line 414 ish). It may be interesting to look for evidence of the RH enhancing or suppressing the particle number concentrations- if there is any effect of RH visible, this might suggest that the new particle formation is not at the kinetic limit for the vapours involved (or that water is important for the chemistry leading to the NPF). However, again I appreciate that this kind of investigation may be more appropriate for future studies with instrumentation better able to measure organic gasphase chemistry.

The discussion in line 414ff (old) refers to the column moisture content and precipitable water, not to the relative humidity in the upper troposphere. However, to follow up on the reviewer's

suggestion, we examined several flights (AC07, AC09, AC13, and AC18) for relationships between RH and $N_{CN}$. We found a tendency for the layers with high $N_{CN}$ to be associated with moister layers (RH>50%), but also many exceptions. This relationship may simply have to do with the fact that moisture was brought up with the convective clouds, or there may be a relationship with the actual particle formation process, but at this point we have no way to answer these questions. We added a couple of sentences on this in section 3.5.3. We are planning a future campaign dedicated to process-level studies of NPF in the UT.

3. Related to comment #1, can the authors suggest some possible explanations for why the areas of extremely high particle concentration (suggestive of very recent new particle formation) are usually organised in thin layers?

The outflow from convective clouds tends to become stretched into relatively thin layers due to velocity shear and subsidence, especially when transported over considerable distances (for a discussion, see Eastham and Jacob, 2017, and references therein).

**Conceptual model:**
In general, I find the arguments in this section compelling and I have only minor comments, see below.
**Conclusions:**
At lines 1230-1238, the authors point out that in pre-industrial times, the mechanism they propose would operate unchanged, while sources of low-altitude particles would be diminished, meaning that upper-troposphere new particle formation may in some cases become the dominant source of CCN in the boundary layer. They further propose that the aerosol profile in polluted continental regions may be flipped in the pre-industrial compared to the present day.
The authors do make it clear that these statements are speculative, and I appreciate the need to be concise. However, at lines 1223-1224 I think they should additionally point out that the pre-industrial atmosphere may not have been particularly pristine in many places, with large marine, volcanic and fire emissions leading to uncertain but possibly high concentrations of boundary layer particles. It would be enough to modify "strongly affected by anthropogenic aerosols" to "strongly affected by anthropogenic or natural primary aerosols".

Done.

Furthermore, to justify the arguments in the paragraph "The conceptual model proposed here implies…" the authors need to show evidence that in present-day *polluted* areas, concentrations of particles greater than say 3nm in diameter are usually lower at high altitude than they are at low altitude. A very brief look at flight data from INTEX over the eastern USA suggested to me that there is still plenty of particle production in the upper troposphere in polluted regions (in these areas, of course there are more particles in the BL, but also more SO2 making particles in the UT). There is a modification to the gradient of the aerosol profile over the industrial period (modelling studies suggest this is true even as a global average, see for example Fig. 1a of

http://onlinelibrary.wiley.com/doi/10.1002/2017JD026844/abstract) but to say "turned upside down" seems a bit strong.

A climatology of aerosol concentrations in the UT is available from the CARIBIC project. This shows median particle concentrations (> 12 nm) in the region 200-300 hPa to be ~3500 cm$^{-3}$ over North America, ~2500 cm$^{-3}$ over Europe, and ~3000 cm$^{-3}$ over India (Ekman et al., 2012). Of course, there are elevated values at particular place and times, such as those the reviewer refers to, but they appear to be more the exception than the rule. In contrast, the averages measured at ground level at polluted continental sites worldwide range between 3400 and 19,000 cm$^{-3}$ in the compilation by Andreae (2009). This is quite close to being the exact opposite of the distribution measured during ACRIDICON-CHUVA, where the averages (±std.dev.) were 7700±7970 cm$^{-3}$ in the UT and 1650±980 cm$^{-3}$ in the LT. This information has been added into the Conclusions text. But, so as not to over-generalize, we have modified the statement to "… has been turned upside down, at least in many polluted regions".

**Minor comments**
The text is well written and logically structured, but as it is long, the introduction of more cross-referencing between sections to relate different parts of the text together would be very helpful. For example, it would be helpful to reference Figures 4 and 6 at the appropriate places in the paragraph starting on line 471.

Done.

Also at line 662 it would be helpful to remind the reader that the two aerosol populations were already introduced at line 547, to confirm the distinctions drawn are the same in the two cases.

Done.

Structurally, the one concern I have is that Section 3.4 and Section 3.5 start with essentially the same question, then Section 3.4 deals with one part of it and then 3.5 introduces another possible source (immediate outflows) and most of the section is then spent dealing with this new issue that was not previously introduced. Can the authors think about whether it is possible to organise these sections more rigidly and flag up the most important messages more strongly?

We have added some introductory sentences at the beginning of section 3.4 that inform the reader what to expect in sections 3.4 and 3.5.

The discussion of the trajectory results (3.5.2,3.5.3) probably merits a new section 3.6.

We prefer to retain the current structure, as we think it is appropriate to the discussion.

Line 93: the authors might cite here only the papers which really focus on UT NPF: the Carslaw (2017) citation seems out of place in this paragraph.

The reference has been deleted.

Line 197 or 218: please state approximate distance between inlet and instrument, to put these flow rates and efficiencies in context. Also for the UHSAS and CCNC.

The length of the line to the CPC was about 2 m, to the CCN about 1.8 m. The flow in the inlets was increased by using a variable flow bypass to reduce particle losses. The UHSAS is mounted in a wing-pod and has no inlet line.

The authors convincingly demonstrate NPF is the only possible source of the particles. However, they should emphasise the sentence at line 843-845 more, where the key reason for why the particles cannot come from long range transport is explained (even though it is fairly obvious). This could be done by forward referencing Section 3.5 from line 553, or restructuring slightly as suggested above.

Done, by the new introductory sentences at the beginning of section 3.4.

Line 806: please label the citation to Schulz as 'submitted', or 'in preparation', here. I couldn't find the paper.

Done.

Line 1087 The authors should specify that the CERN CLOUD chamber studies so far published only provide the temperature dependence of inorganic NPF. NPF involving organic molecules may behave quite differently, though NPF is still obviously expected to increase at lower temperatures (all other things being equal). Similarly, the Yu (2017) study does not fully account for the gas-phase chemistry (as this chemistry is not fully characterised the authors had little choice), so it treats NPF of organics rather similarly to that for H2SO4.

Cautionary sentence added: "Note, however, that these temperature dependencies are based on measurements for inorganic NPF, and that while the trends for organics are expected to be similar, the magnitude of the increase in nucleation rates for organics may be quite different."

Line 1123 The Gordon (2016) modelling study didn't quite suggest "dominant mode of new particle formation in the pre-industrial atmosphere", perhaps replace by "in large parts of the preindustrial atmosphere".

Done.

On page 68, the footnote labels to Table 1 all read "a".

Corrected.

Fig S1 caption: aren't the parcels zoomed in approximately a 6x6 degree box, not 3x3? Despite the valuable efforts of the authors to make things clear with the colour scale of

the trajectories and marking the GOES time on the figure, I found the way this was phrased in the caption a little confusing. If I understand, the snapshots are zoomed in a box centred at the parcel location at the time shown on the top **of** the snapshots, **in parentheses backwards from the parcel start**. Perhaps the authors could add something like the italicised words/phrases to the caption?

The reviewer must be referring to Figure S2 (not S1). Yes, the boxes are 6x6 degrees and we corrected that in the caption. We added the wording on the number of hours in parentheses.

Andreae, M. O., Correlation between cloud condensation nuclei concentration and aerosol optical thickness in remote and polluted regions: Atmos. Chem. Phys., 9, 543–556, 2009.

Eastham, S. D., and Jacob, D. J., Limits on the ability of global Eulerian models to resolve intercontinental transport of chemical plumes: Atmos. Chem. Phys., 17, 2543-2553, doi:10.5194/acp-17-2543-2017, 2017.

Ekman, A. M. L., Hermann, M., Gross, P., Heintzenberg, J., Kim, D., and Wang, C., Sub-micrometer aerosol particles in the upper troposphere/lowermost stratosphere as measured by CARIBIC and modeled using the MIT-CAM3 global climate model: J. Geophys. Res., 117, D11202, doi:10.1029/2011jd016777, 2012.

Twohy, C. H., Clement, C. F., Gandrud, B. W., Weinheimer, A. J., Campos, T. L., Baumgardner, D., Brune, W. H., Faloona, I., Sachse, G. W., Vay, S. A., and Tan, D., Deep convection as a source of new particles in the midlatitude upper troposphere: J. Geophys. Res., 107, 4560, doi:10.1029/2001JD000323, 2002.

Weigel, R., Borrmann, S., Kazil, J., Minikin, A., Stohl, A., Wilson, J. C., Reeves, J. M., Kunkel, D., de Reus, M., Frey, W., Lovejoy, E. R., Volk, C. M., Viciani, S., D'Amato, F., Schiller, C., Peter, T., Schlager, H., Cairo, F., Law, K. S., Shur, G. N., Belyaev, G. V., and Curtius, J., In situ observations of new particle formation in the tropical upper troposphere: the role of clouds and the nucleation mechanism: Atmos. Chem. Phys., 11, 9983-10,010, doi:10.5194/acp-11-9983-2011, 2011.